



# Leaf-scale quantification of the effect of photosynthetic gas exchange on $\Delta^{17}O$ of atmospheric CO₂

Getachew Agmuas Adnew[1], Thijs L. Pons[2], Gerbrand Koren[3], Wouter Peters[3,4], Thomas Röckmann[1]

[1]Institute for Marine and Atmospheric research Utrecht (IMAU), Utrecht University, The Netherlands
[2]Institute of Environmental Biology, Utrecht University, The Netherlands
[3]Department of Meteorology and Air Quality, Wageningen University, The Netherlands
[4]Centre for Isotope Research, University of Groningen, The Netherlands

*Correspondence to*: Getachew Agmuas Adnew (getachewagmuas@gmail.com)

## Abstract

Understanding the processes that affect the triple oxygen isotope composition of atmospheric CO₂ during gas exchange can help constrain the interaction and fluxes between the atmosphere and the biosphere. We conducted leaf cuvette experiments under controlled conditions, using three plant species. The experiments were conducted at two different light intensities and using CO₂ with different $^{17}O$-excess. The oxygen isotope composition of CO₂ was used to estimate $c_m$, the mole fraction of CO₂ at the CO₂-H₂O exchange site. Our results demonstrate that two key factors determine the effect of gas exchange on the $\Delta^{17}O$ of atmospheric CO₂. The relative difference between $\Delta^{17}O$ of the CO₂ entering the leaf and the CO₂ in equilibrium with leaf water, and the back-diffusion flux of CO₂ from the leaf to the atmosphere, which can be quantified by the $c_m/c_a$ ratio where $c_a$ is the CO₂ mole fraction in the surrounding air. At low $c_m/c_a$ ratio the discrimination is governed mainly by diffusion into the leaf, and at high $c_m/c_a$ ratio by back-diffusion of CO₂ that has equilibrated with the leaf water. Plants with a higher $c_m/c_a$ ratio modify the $\Delta^{17}O$ of atmospheric CO₂ more strongly than plants with a lower $c_m/c_a$ ratio. Based on the leaf cuvette experiments, the global value for discrimination against $\Delta^{17}O$ of atmospheric CO₂ during the photosynthetic gas exchange is estimated to be -0.57±0.14‰ using $c_m/c_a$ values of 0.3 and 0.7 for C₄ and C₃ plants, respectively. The main uncertainties in this global estimate arise from variation in $c_m/c_a$ ratios among plants and growth conditions.

## 1. Introduction

Stable isotope measurements of CO₂ provide important information on the magnitude of the CO₂ fluxes between atmosphere and biosphere, which are the largest components of the global carbon cycle (Farquhar et al., 1989a;1993;Ciais et al., 1997a;1997b;Flanagan and Ehleringer, 1998;Yakir and Sternberg, 2000;Gillon and Yakir, 2001;Cuntz et al., 2003a;2003b). A better understanding of the terrestrial carbon cycle is essential for predicting future climate and atmospheric CO₂ mole fractions (Booth et al., 2012). Gross primary productivity (GPP), the total carbon dioxide uptake by vegetation during photosynthesis, can only be determined indirectly and remains poorly constrained (Cuntz, 2011;Welp et al., 2011). For example, Beer et al. (2010) estimated global GPP to be 102-135 PgC yr⁻¹





(85% confidence interval, CI) using machine learning techniques by extrapolating from a database of eddy-covariance measurements of $CO_2$ fluxes. This estimate has since then been widely used as target for terrestrial vegetation models (Sitch et al., 2015), and replicated using extensions on the technique (Jung et al., 2019). As an alternative, (Welp et al., 2011) estimated global GPP to be 150-175 PgC yr$^{-1}$ using variations in $\delta^{18}O$ of atmospheric $CO_2$ after El Nino events.

The concept behind the latter study was that atmospheric $CO_2$ exchanges oxygen isotopes with leaf and soil water, and this isotope exchange mostly determines the observed variations in $\delta^{18}O$ of $CO_2$ (Francey and Tans, 1987;Yakir, 1998); see below for definition of the delta value. Following the 97/98 ENSO event, the anomalous $\delta^{18}O$ signature imposed on tropical leaf and soil waters was transferred to atmospheric $CO_2$, before slowly disappearing as a function of the lifetime of atmospheric $CO_2$. This in
turn is governed by the land vegetation uptake of $CO_2$ during photosynthesis, as well as soil invasion of $CO_2$ (Miller et al., 1999;Wingate et al., 2009). In addition to the latter term, the equilibration of $CO_2$ with water is an uncertain parameter in this calculation, partly because the $\delta^{18}O$ of water at the site of isotope exchange in the leaf is not well defined. Importantly, a significant but variable $\delta^{18}O$ gradient can occur in leaves due to the preferential evaporation of $H_2^{16}O$ relative to $H_2^{18}O$ (Gan et al., 2002;Farquhar and Gan,
2003;Gan et al., 2003;Cernusak et al., 2016), which induces a considerable uncertainty in estimating $\delta^{18}O$ of the isotopically exchanged $CO_2$. Similar considerations for the transfer of the $\delta^{18}O$ signature of precipitation into the soils, and then up through the roots, stems, and leaves make $^{18}O$ of $CO_2$ a challenging measurement to interpret (Cuntz et al., 2003a;2003b;Peylin et al., 1999).

The $^{17}O$-excess of $CO_2$ ($\Delta^{17}O$, involving the triple oxygen isotope composition of $CO_2$, see equation 5) has been suggested as additional independent tracer for constraining global GPP (Hoag et al., 2005;Thiemens et al., 2013;Hofmann et al., 2017;Liang et al., 2017b;Koren et al., 2019). Because $\Delta^{17}O$ is not or only slightly sensitive to mass-dependent fractionation processes acting on $CO_2$ and $H_2O$, its interpretation may be less sensitive to the effects mentioned above. In the stratosphere, $CO_2$ obtains a
considerable $^{17}O$-excess due to the transfer of oxygen atoms from $^{17}O$-enriched ozone to $CO_2$ via photochemical isotope exchange (Thiemens et al., 1991;1995;Lyons, 2001;Lämmerzahl et al., 2002;Thiemens, 2006;Kawagucci et al., 2008). Once this anomalous signature has been created in the stratosphere, the only process that removes the anomaly is isotope exchange with leaf water, soil water and ocean water at the Earth's surface, after $CO_2$ has re-entered the troposphere (Boering, 2004;Thiemens
et al., 2014;Liang and Mahata, 2015;Hofmann et al., 2017). Isotope exchange with water in clouds and rain droplets is negligible due to the absence of carbonic anhydrase (Francey and Tans, 1987).

Isotope exchange with leaf water is more efficient relative to ocean water due to the presence of the enzyme carbonic anhydrase (CA), which effectively catalyzes the conversion of $CO_2$ and $H_2O$ to $HCO_3^-$
and $H^+$ and vice versa (Francey and Tans, 1987;Friedli et al., 1987;Badger and Price, 1994;Gillon and Yakir, 2001). Therefore, $CO_2$ quickly equilibrates its isotopic composition with that of leaf water with a well-established temperature dependent fractionation factor (Brenninkmeijer et al., 1983;Barkan and Luz, 2012). The $^{17}O$-excess of $CO_2$ ($\Delta^{17}O$) (equation 4) at the $CO_2$-$H_2O$ exchange site in the leaf will vary much less than $\delta^{18}O$ because the transfer of water from the precipitation to the leaves, as well as



evaporation, are mass dependent processes with a well-known three isotope slope (Barkan and Luz, 2005;Landais et al., 2006). Therefore, $\Delta^{17}O$ may be a more robust tracer for GPP than $\delta^{18}O$ (Hoag et al., 2005;Hofmann et al., 2017;Koren et al., 2019).

Several measurements of $\Delta^{17}O$ of $CO_2$ in atmospheric samples from different locations have been performed to use it as a tracer for GPP (Liang et al., 2006;Barkan and Luz, 2012;Thiemens et al., 2014;Liang and Mahata, 2015;Laskar et al., 2016;Hofmann et al., 2017;Liang et al., 2017b). A significant limitation of such studies is that the triple oxygen isotope signatures associated with the large $CO_2$ exchange fluxes (photosynthesis, respiration, soil invasion) are not well established, and in many cases are based on assumptions, but not confirmed by measurement (Koren et al., 2019). Further interpretation
of $\Delta^{17}O$ measurements depends on our ability to understand and untangle the individual processes.

The effect of vegetation on the isotope composition of atmospheric $CO_2$ depends on the type of plant metabolism. Generally, plants are classified as $C_3$ and $C_4$ based on their metabolism. For detail explanation about the type of plants the reader is directed to (Ehleringer and Monson, 1993;Ehleringer
and Cerling, 2002;Ubierna et al., 2018;Ubierna et al., 2019;Cousins et al., 2020). In $C_3$ plants, the $CO_2$ diffuses through the boundary layer surrounding the leaf, the stomata, the intercellular air space, cell wall, plasma membrane, cytosol, chloroplast envelope and stroma where it is carboxylated into $C_3$ acid by Ribulose-1,5-bisphosphate carboxylase oxygenase RubisCO (Farquhar et al., 1982;Evans et al., 2009;Ubierna et al., 2019;Cousins et al., 2020) (Figure 1). For $C_4$ plants, $CO_2$ is initially carboxylated
into $C_4$ acid by phosphoenolpyruvate carboxylase (PEPC) in the mesophyll cytosol (Sage and Monson, 1998;Caemmerer et al., 2014;Cousins et al., 2020) (Figure 1). The mole fraction of $CO_2$ at the $CO_2$-$H_2O$ exchange site ($c_m$) is an important parameter to determine the effect of photosynthesis on the triple oxygen isotope composition of atmospheric $CO_2$. In $C_3$ plants, CA is found in the chloroplast, cytosol, mitochondria and plasma membrane (Fabre et al., 2007;DiMario et al., 2016), the $CO_2$-$H_2O$ exchange can
occur anywhere between the plasma membrane and within the chloroplast. For $C_4$ plants, CA is mainly found in the cytosol, the $CO_2$-$H_2O$ exchange occurs in the cytosol (Badger and Price, 1994).

In this study we report the effect of photosynthesis on the $\Delta^{17}O$ of $CO_2$ in the surrounding air at the leaf level, using three species that are representative for three different biomes. The fast-growing annual
herbaceous $C_3$ species *Helianthus annuus* (sunflower) has a high photosynthetic capacity ($A_{max}$) and high stomatal conductance ($g_s$) and is representative for temperate and tropical crops (Fredeen et al., 1991). The slower growing perennial evergreen $C_3$ species *Hedera hybernica* (ivy) is representative of forests and other woody vegetation and stress subjected habitats (Pons et al., 2009). The fast-growing, agronomically important crop *Zea mays* (maize) is an herbaceous annual $C_4$ species with a high $A_{max}$ and
a low $g_s$, typical for savanna type vegetation (Weijde et al., 2013). Sunflower and ivy are used to cover the $c_m/c_a$ ratio among $C_3$ plants and maize represents $c_m/c_a$ ratio for the $C_4$ plants. We measured the triple oxygen isotopic composition of $CO_2$ entering and leaving the leaf cuvette to calculate the isotopic fractionation associated with photosynthesis. Using these results, we estimated the effect of terrestrial vegetation on $\Delta^{17}O$ of $CO_2$ in the global atmosphere.




## 2. Theory

### 2.1. Notation and definition of δ values

Isotopic composition is expressed as the relative deviation of the heavy to light isotope ratio in a sample relative to reference material and it is denoted as δ (McKinney et al., 1950), expressed in per mill (‰).
In the case of oxygen isotopes, the deviation of the two isotope ratios $^{18}R = [^{18}O]/[^{16}O]$ and $^{17}R = [^{17}O]/[^{16}O]$ from an international reference ratio (Vienna Standard Mean Ocean Water, VSMOW) is quantified in δ notation as:

$$\delta^{18}O = \frac{^{18}R_{sample}}{^{18}R_{VSMOW}} - 1 \qquad (1)$$

$$\delta^{17}O = \frac{^{17}R_{sample}}{^{17}R_{VSMOW}} - 1 \qquad (2)$$


For most processes, isotope fractionation depends on mass, and therefore the fractionation against $^{17}O$ is approximately half of the fractionation against $^{18}O$ (equation 3).

$$\ln(\delta^{17}O + 1) = \lambda \times \ln(\delta^{18}O + 1) \qquad (3)$$

The factor $\lambda$ ranges from 0.5 to 0.5305 for different molecules and process (Matsuhisa et al., 1978;Young
et al., 2002;Thiemens, 1999;Cao and Liu, 2011). This relation is generally referred to as mass dependent fractionation and can also be expressed as $\left[ \frac{^{17}R}{^{17}R_{ref}} = (\frac{^{18}R}{^{18}R_{ref}})^{\lambda}, \text{or } \alpha^{17} = (\alpha^{18})^{\lambda} \right]$ where $\alpha^{17}$ and $\alpha^{18}$ are the fractionations of $^{17}O$ and $^{18}O$ relative to a reference material, respectively. For small $\delta^{17}$ and $\delta^{18}$ values, equation (3) can be linearized to $\delta^{17}O = \lambda \times \delta^{18}O$. $\Delta^{17}O$ is used to quantify the degree of deviation from equation (3) (see equation 4). Note that deviations from a chosen reference slope $\lambda$ are not only caused
by mass independent fractionation process, but can also be introduced by mass dependent process with a different three isotope slope relative to the chosen reference line (Barkan and Luz, 2005;Landais et al., 2006;2008;Luz and Barkan, 2010;Barkan and Luz, 2011;Pack and Herwartz, 2014).

$$\Delta^{17}O = \ln(\delta^{17}O + 1) - \lambda \times \ln(\delta^{18}O + 1) \qquad (4)$$

The choice of $\lambda$ is in principle arbitrary and in this study, we used $\lambda = 0.528$, the value associated with meteoric water (Meijer and Li, 1998;Landais et al., 2008;Brand et al., 2010;Luz and Barkan, 2010;Barkan and Luz, 2012;Sharp et al., 2018). Note that $\Delta^{17}O$ is not a measured quantity, it is inferred from measurements of $\delta^{17}O$ and $\delta^{18}O$.

### 2.2. Calculation of the discrimination in the oxygen isotope anomaly of CO₂



The overall isotope fractionation associated with the photosynthesis of $CO_2$ is commonly quantified using the term discrimination as described in (Farquhar and Richards, 1984;Farquhar et al., 1989a;Farquhar and Lloyd, 1993). We use the symbol $\Delta_A$ for discrimination due to assimilation in this manuscript since the commonly used $\Delta$ is already used for the definition of $^{17}O$-excess (see above). $\Delta_A$ quantifies the enrichment or depletion of carbon and oxygen isotopes of $CO_2$ in the surrounding atmosphere relative to the $CO_2$ that is assimilated (Farquhar and Richards, 1984). It can be calculated from the isotopic composition of the $CO_2$ entering and leaving the leaf cuvette (Evans et al., 1986;Gillon and Yakir, 2000b;Barbour et al., 2016), for instance for $^{18}O$-photosynthetic discrimination, as:

$$\Delta_A{}^{18}O_{obs} = \frac{^{18}R_a}{^{18}R_A} - 1 = \frac{\delta^{18}O_a - \delta^{18}O_A}{1 + \delta^{18}O_A} = \frac{\zeta \times (\delta^{18}O_a - \delta^{18}O_e)}{1 + \delta^{18}O_a - \zeta \times (\delta^{18}O_a - \delta^{18}O_e)} \quad (5)$$

where the indices $e$, $a$ and $A$ refer to $CO_2$ entering ($e$) and leaving ($a$) the cuvette and being assimilated ($A$), respectively. $\zeta = \frac{c_e}{c_e - c_a}$, where $c_e$ and $c_a$ are the mole fractions of $CO_2$ entering and leaving the leaf cuvette. The observed $^{17}O$-photosynthetic discrimination ($\Delta_A{}^{17}O_{obs}$) is calculated analogously. We note that the concept of discrimination associated with photosynthesis is more complicated for the oxygen isotopes compared to $^{13}C$. For $^{13}C$, the observed isotope change is directly associated with an isotope effect in assimilation due to RubisCO and PEPC. For the oxygen isotopes, the observed change in isotopic composition is caused by oxygen isotope exchange of $CO_2$ with leaf water rather than by fractionation in the assimilation process itself.

The discrimination against $\Delta^{17}O$ associated with assimilation in global models, assuming the degree of equilibration between $CO_2$ and $H_2O$ is unity, is calculated as shown in equation 6 (Hofmann et al., 2017;Liang et al., 2017b;Koren et al., 2019).

$$\Delta_A\Delta^{17}O = (\lambda_{diffusion} - \lambda_{RL}) \times \ln(\overline{a}_{18} + 1) + (\Delta^{17}O_m - \Delta^{17}O_a)\frac{c_m}{c_a - c_m} \quad (6)$$

$\overline{a}_{18}$, is the weighted mean of discrimination occurring during the diffusion of $^{12}C^{18}O^{16}O$ from the ambient air to the $CO_2$-$H_2O$ exchange site and it is estimated to be 7.4‰ (Farquhar et al., 1993). This value has been adopted in several global studies of $\delta^{18}O$ of atmospheric $CO_2$ (Ciais et al., 1997a;1997b;Cuntz et al., 2003a;2003b) and the global $\Delta^{17}O$ studies (Hofmann et al., 2017;Liang et al., 2017b;Koren et al., 2019). $\lambda_{diffusion}$=0.509 is the coefficient associated the fractionation of $C^{17}OO$ as it diffuses through air relative to $C^{18}OO$ (Young et al., 2002) and $\lambda_{RL}$ =0.528 (the reference slope used in this study). $\Delta^{17}O_m$ and $c_m$ are the oxygen isotope anomaly and mole fraction of $CO_2$ at the $CO_2$-$H_2O$ exchange site, respectively.

A good approximation for the observed $^{18}O$-discrimination can be derived from the leaf exchange parameters (Farquhar and Lloyd, 1993), see supplementary material for the derivation, as:





$$\Delta_A{}^{18}O_{FM} = \frac{\overline{a}_{18} + \frac{c_m}{c_a - c_m} \times \delta^{18}O_{ma}}{1 - \frac{c_m}{c_a - c_m} \times \delta^{18}O_{ma}} \approx \overline{a}_{18} + \frac{c_m}{c_a - c_m} \times \left(\delta^{18}O_m - \delta^{18}O_a\right) \tag{7}$$

The subscript *FM* stands for Farquhar model. $\delta^{18}O_{ma}$ is the enrichment in $\delta^{18}O$ of $CO_2$ in full isotopic equilibrium with water at the exchange site relative to the $CO_2$ in the surrounding air. $\delta^{18}O_{ma}$ is calculated as:

$$\delta^{18}O_{ma} = \frac{\delta^{18}O_m - \delta^{18}O_a}{1 + \delta^{18}O_a} \tag{8}$$

$\delta^{18}O_m$ is the isotope composition of $CO_2$ in equilibrium with leaf water at the $CO_2$-$H_2O$ exchange site (equation 16). Analogous to $\Delta_A{}^{18}O_{FM}$, a similar equation for $\Delta_A{}^{17}O_{FM}$ can be derived like equation 7 (see equation S12, supplementary material). In the global models (Hofmann et al., 2017;Liang et al., 2017b;Koren et al., 2019), $\Delta^{17}O$-photosynthetic discrimination shown in equation 6 is derived from $\Delta_A{}^{17}O_{FM}$ and $\Delta_A{}^{18}O_{FM}$ as shown from equation 9 to 11.

$$\Delta_A\Delta^{17}O = \left(\overline{a}_{17} + \frac{c_m}{c_a - c_m}\left(\delta^{17}O_m - \delta^{17}O_a\right)\right) - \lambda_{RL} \times \left(\overline{a}_{18} + \frac{c_m}{c_a - c_m}\left(\delta^{18}O_m - \delta^{18}O_a\right)\right) \tag{9}$$

$$\Delta_A\Delta^{17}O = \left(\overline{a}_{17} - \lambda_{RL} \times \overline{a}_{18}\right) + \left[\left(\delta^{17}O_m - \lambda_{RL}\delta^{18}O_m\right) - \left(\delta^{17}O_a - \lambda_{RL}\delta^{18}O_a\right)\right]\frac{c_m}{c_a - c_m} \tag{10}$$

$$\Delta_A\Delta^{17}O = \left(\overline{a}_{17} - \lambda_{RL} \times \overline{a}_{18}\right) + \left[\Delta^{17}O_m - \Delta^{17}O_a\right]\frac{c_m}{c_a - c_m} \tag{11}$$

Note that, $\ln(\overline{a}_{18} + 1) \approx \overline{a}_{18}$ and $(\overline{a}_{17} - \lambda_{RL} \times \overline{a}_{18}) = (\lambda_{diffusion} - \lambda_{RL}) \times \ln(\overline{a}_{18} + 1)$, i.e. the left and right side of equation 11 is similar to the left and right side of equation 6, respectively. For the leaf cuvette experiments, the $\Delta_A \Delta^{17}O$ is calculated with $\lambda_{RL} = 0.528$ as:

$$\Delta_A\Delta^{17}O_{obs} = \ln\left(\Delta_A{}^{17}O_{obs} + 1\right) - 0.528 \times \ln\left(\Delta_A{}^{18}O_{obs} + 1\right) \tag{12}$$

$\Delta^{17}O$ calculated using equation 4 (logarithmic definition) is not a conserved quantity. Adding or subtracting $\Delta^{17}O$ results calculated using equation 4 (logarithmic definition) results in an error that gets larger when the relative difference in $\delta^{18}O$ between the two $CO_2$ gases increases regardless of the $\Delta^{17}O$ of the individual $CO_2$ gases (Figure 1). The discrepancy of adding and subtracting $\Delta^{17}O$ with a logarithmic definition is the largest when the two $CO_2$ gases are mixed in equal proportions (50:50). To avoid this error either the subtraction and addition should be done in small $\delta$'s (see below) and $\Delta^{17}O$ value should





210  be calculated from the small δ's differences or use the linear definition of the anomaly ($\Delta^{17}O = \delta^{17}O - \lambda \times \delta^{18}O$) (Liang et al., 2017b). (Liang et al., 2017b), in their global mass balance budget calculation, reported adding and subtracting the anomaly with logarithmic definition results in a 10% error in each reservoir.

### 2.3. Isoflux calculation for $\Delta_A\Delta^{17}O$

An isoflux is the product of isotope composition and gross mass flux of the molecule. In the case of assimilation, which is a net sink, the net flux $F_A = F_{AL} - F_{LA}$ is multiplied with the discrimination associated with assimilation. $F_{LA}$ and $F_{AL}$ are total $CO_2$ fluxes from leaf to the atmosphere and from atmosphere to leaf, respectively. The global $\Delta_A^{18}O$-isoflux is $F_A \times \Delta_A^{18}O$ (Farquhar et al., 1993;Ciais et al., 1997a;1997b;Gillon and Yakir, 2001;Cuntz et al., 2003a;2003b).

The leaf scale discrimination against $\Delta^{17}O$ is then extrapolated to global vegetation using representative values for $\Delta_A\Delta^{17}O_{C4}$ and $\Delta_A\Delta^{17}O_{C3}$, considering the observed discriminations as a function of the $c_m/c_a$ ratio and the global average values for $\Delta^{17}O$ of leaf water and atmospheric $CO_2$, and the relative fractions of photosynthesis by $C_4$ and $C_3$ plants, respectively as:

$$\Delta_A\Delta^{17}O_{global} = f_{C4} \times \Delta_A\Delta^{17}O_{C4} + f_{C3} \times \Delta_A\Delta^{17}O_{C3} \qquad (13)$$

where $f_{C4}$ and $f_{C3}$ are the photosynthesis weighted global coverage of $C_4$ and $C_3$ vegetation. $\Delta_A\Delta^{17}O_{C4}$ and $\Delta_A\Delta^{17}O_{C3}$ quantify the discrimination against $\Delta^{17}O$ by $C_4$ and $C_3$ plants, which are calculated using estimated values of $c_m/c_a$ from a model. The global scale $\Delta^{17}O_A$ isoflux is calculated by multiplying the discrimination with the assimilation flux.

$$F_A \times \Delta_A\Delta^{17}O = A \times (f_{c4} \times \Delta_A\Delta^{17}O_{c4} + f_{c3} \times \Delta_A\Delta^{17}O_{c3}) \qquad (14)$$

where, A=0.88×GPP is the terrestrial assimilation rate, the factor 0.88 accounts for the fraction of $CO_2$ released due to autotrophic respiration (Ciais et al., 1997a).

### 2.4. Mole fraction of $CO_2$ at the site of $CO_2$-$H_2O$ exchange

Following (Farquhar and Cernusak, 2012;Barbour et al., 2016;Osborn et al., 2017), the $CO_2$ mole fraction at the site of $CO_2$-$H_2O$ exchange is calculated as:

$$c_m = c_i \left( \frac{\delta^{18}O_i - a_{18w} - \delta^{18}O_A \times (1 + a_{18w})}{\delta^{18}O_m - a_{18w} - \delta^{18}O_A \times (1 + a_{18w})} \right) \qquad (15)$$

where $\delta^{18}O_i$ is $\delta^{18}O$ of $CO_2$ in the intercellular airspace (Farquhar and Cernusak, 2012), $a_{18w}$ is the fractionation of $\delta^{18}O$ of $CO_2$ during diffusion and dissolution in water (0.8‰) (Farquhar and Lloyd, 1993),





$\delta^{18}O_A$ is $\delta^{18}O$ of the assimilated $CO_2$ and $\delta^{18}O_m$ is the $\delta^{18}O$ of $CO_2$ in equilibrium with leaf water at the $CO_2$-$H_2O$ exchange site. Assuming the isotopic composition of leaf water at the $CO_2$-$H_2O$ exchange site is the same as the $\delta^{18}O$ of leaf water at the evaporation site, $\delta^{18}O_m$ can be calculated as:

$$\delta^{18}O_m = \left(\delta^{18}O_{wes} + 1\right) \times \left(1 + \varepsilon_w^{18}\right) - 1 \tag{16}$$

where $\delta^{18}O_{wes}$ is the $\delta^{18}O$ of $H_2O$ at the exchange site and $\varepsilon_w^{18}$ is the equilibrium fractionation between $CO_2$ and water (equation 18). The $\delta^{18}O_{wes}$ is calculated using the modified Craig and Gordon model (Farquhar et al., 1989b;Flanagan et al., 1991;Harwood et al., 1998) as:

$$\delta^{18}O_{wes} = \delta^{18}O_{trans} + \varepsilon^{18}_k + \varepsilon^{18}_{equ} + \frac{w_a}{w_i} \times \left(\delta^{18}O_{wa} - \varepsilon^{18}_k + \delta^{18}O_{trans}\right) \tag{17}$$

where $w_i$ and $w_a$ are the mole fraction of water vapor inside the leaf and in the air leaving the cuvette and $\varepsilon^{18}_k$ and $\varepsilon^{18}_{equ}$ are the kinetic fractionation of water vapor in the air and the equilibrium fractionation between liquid and gas phase water, respectively (see Appendix 2). The equilibrium fractionation between $CO_2$ and water ($\varepsilon_w^{18}$) is temperature dependent and is calculated after (Brenninkmeijer et al., 1983) as:

$$\varepsilon_w^{18} = \frac{17604}{T} - 17.93 \tag{18}$$

where $T$ is leaf temperature. Analogous to $\delta^{18}O$, the mole fraction of $CO_2$ in the mesophyll cell can be calculated using $\delta^{17}O$ values (Appendix 3, equation A3.6). The detailed derivation for the $c_m$ calculation is shown in appendix 3.

## 3. Materials and methods

### 3.1. Plant material and growth conditions

Sunflower (*Helianthus annuus* L. cv "sunny") was grown from seeds in 0.6 L pots with potting soil (Primasta, the Netherlands. The dwarf type sunflowers were grown until the first leaf pair that was used for the experiments reached the final size, which is about 4 weeks. All leaves appearing above the first leaf pair were removed to avoid shading. Established juvenile Ivy (*Hedera hybernica* L.) plants were pruned and planted in 6 L pots. After at least 6 weeks in the growth chamber, leaves that had developed and matured there were used for the experiments. Mays (*Z. mays L.* cv "saccharate") was grown from seed in 1.6 L pots for at least 7 weeks. The 4th or higher leaf number was used for the experiments when mature. A section of the leaf at about 1/3 from the tip was inserted in the leaf cuvette. The pots with the potting soil were soaked in a complete nutrient solution containing 6.6 mM nitrate (Millenaar et al., 2005) after planting and applied weekly during growth. They were placed on a sub-irrigation system that provided water during the growth period in a controlled environment growth chamber, air temperature 20°C, relative humidity 70% and $CO_2$ mole fraction of about 400 ppm. The photosynthetic photon flux density (PPFD) was about 300 µmol m$^{-2}$ s$^{-1}$ during a daily photoperiod of 16 hours measured with a PPFD meter (Licor LI-250A, Li-Cor Inc, Nebraska, USA).



### 3.2. Gas exchange experiments

Gas exchange experiments were performed in an open system where a controlled flow of air enters and leaves the leaf cuvette similar to the setup used by (Pons and Welschen, 2002). A schematic of the gas
exchange experimental setup is shown in Figure 2. The leaf cuvette had dimensions of 7 x 7 x 7 cm$^3$ (lxwxh) and the top part of the cuvette was transparent. The temperature of the leaf was measured with a K type thermocouple. The leaf chamber temperature was controlled by a temperature-controlled water bath kept at 20ºC (Tamson TLC 3, The Netherlands). A fan inside the chamber was used to mix the air inside the cuvette thoroughly and to create a high boundary layer conductance. A halogen lamp in a slide
projector was used as a light source. Infrared was excluded by reflection from a cold mirror. The light intensity was varied by with spectrally neutral filters.

The $CO_2$ mole fraction of the incoming and outgoing air was measured with an infrared gas analyzer (IRGA, model LI-6262, LI-COR Inc., Nebraska, USA). The isotopic composition and mole fraction of
the incoming and outgoing water vapor were measured with a triple water vapor isotope analyzer (WVIA, model 911-0034, Los Gatos Research, USA). Compressed air (ambient outside air without drying) was passed through soda lime to scrub the $CO_2$. The $CO_2$ free air could be humidified depending on the experiment conditions (see Figure 2). The humidity of the inlet air was monitored continuously with a dewpoint meter (General Eastern, Watertown, MA, USA). Pure $CO_2$ (either normal $CO_2$ or isotopically
enriched $CO_2$) was mixed with the incoming air to produce a $CO_2$ mole fraction of 500 ppm. The isotopically enriched $CO_2$ was prepared by photochemical isotope exchange between $CO_2$ and $O_2$ under UV irradiation, as described in detail in (Adnew et al., 2019).

An attached leaf or part of it was inserted in the cuvette., the composition of the inlet air was measured,
and both IRGA and WVIA were switched to measure the outlet air. Based on the $CO_2$ mole fraction of the outgoing air the flow rate of the incoming air to the cuvette was adjusted to establish a drawdown of 100 ppm $CO_2$ due to photosynthesis in the plant chamber. The vapor pressure of the water vapor entering the cuvette is adjusted to the transpiration rate relative to $CO_2$ uptake (Figure 2). The outgoing air was measured continuously until a steady state was reached for $CO_2$ and water mole fractions and δD and
$δ^{18}O$ of the water vapor. After a steady state was established, the air was directed to the sampling flask while the IGRA and WVIA were switched back to measure the inlet air. The air passed through a $Mg(ClO_4)_2$ dryer before entering the sampling flask.

After sampling, the leaf area inside the cuvette was measured with a LI-3100C area meter (Li-COR, Inc.
USA). Immediately afterward, the leaf was placed in a leak tight 9 mL glass vial and kept in a freezer at -20ºC until leaf water extraction.

### 3.3. Calibration of the Water Vapor Isotope Analyzer (WVIA)

The WVIA was calibrated using five water standards provided by IAEA (Wassenaar et al., 2018) for both
$δ^{18}O$ and δD. We did not calibrate the WVIA for $δ^{17}O$, so the $δ^{17}O$ data are not used in the quantitative





evaluation. The results are shown in supplementary material Figure S1. The isotopic composition of the water standards ranged from -50.93 to 3.64‰ and -396.98 to 25.44‰ for δD and $δ^{18}O$, respectively. Based on the calibration using the five standards, a working standard was prepared to correct for short-term variability and to determine the non-linearity (dependency of δD and $δ^{18}O$ on the water vapor mole

fraction). Each day the LGR was calibrated with 3 standards that cover the isotopic composition of the samples measured ($δ^{18}O$ value of -24.777‰, -8.640‰ and 0.11‰, provided by IAEA (Wassenaar et al., 2018)).

Supplementary Figure S2, shows the results of the non-linearity tests. All three isotope signatures of water vapor showed relatively different dependence on the mole fraction of water vapor measured. The $δ^{18}O$ is

independent of the mole fraction above 11000 ppm but decreases at lower mole fraction until 4000 ppm, and then increases again. $δ^{17}O$ is relatively stable for mole fractions higher than 17000 ppm, but increases strongly and in a non-linear manner below. Similarly, δD is independent of the mole fraction of water vapor above 10000 ppm but increases non-linearly below. $δ^{18}O$, $δ^{17}O$ and δD values measured with the WVIS are dependent on the type of carrier gas used when measuring liquid samples as shown for pure

$N_2$ and zero air used as a carrier gas, Figure S2 (Johnson and Rella, 2017). To investigate how the precision of the isotope values depends on the averaging time, Allan deviation (square root of Allan variance) curves are shown in supplementary material Figure 3. All three isotope signatures of water vapor show a similar pattern. The optimum precision is reached at averaging times of 16.7 minutes for $δ^{18}O$ and δD and 15 minutes for $δ^{17}O$ (supplementary material Figure S3). Note that the $δ^{17}O$

measurements of water vapor are not calibrated to an international isotope scale for our experiments.

### 3.4. Leaf water extraction and isotope analysis

Leaf water was extracted by cryogenic vacuum distillation for 4 h at 60ºC following a well-established procedure (Wang and Yakir, 2000;Landais et al., 2006;West et al., 2006). The vial with the leaf was frozen using a liquid nitrogen bath and connected to another empty vial by glass tubing. The system was

then evacuated using a membrane pump (KNF Neuberger, Germany), (Supplementary Figure S4). The pressure was monitored with a Dual pressure sensor (DualTrans transducer, MKS, USA). After the target vacuum was reached (1mbar or below) the extraction system was isolated from the pump. The vial containing the leaf was placed into a heater block (ORI BLOCK DB-1, Techne, England) while the empty vial was kept at liquid nitrogen temperature for 4 hr (Supplementary Figure S4). The extracted leaf water,

~ 0.7 ml (determined based on weight by measuring the leaf weight before and after extraction), was collected in a 2 ml vial (Autosampler vials, National Scientific, the Netherlands) using a pipette and kept in the freezer at -20ºC before isotopic analysis. The $δ^{17}O$ and $δ^{18}O$ of leaf water was determined at the Laboratoire des Sciences du Climat et de l'Environnement laboratory using a fluorination technique as described in (Barkan and Luz, 2005;Landais et al., 2006;2008). Water was converted to $H_2$ and $O_2$ using

$CoF_3$ as fluorinating reagent and the $O_2$ was collected in a sample tube immersed in liquid Helium (-270ºC). Finally, $δ^{17}O$ and $δ^{18}O$ of $O_2$ were measured with an isotope ratio mass spectrometer



(ThermoQuest MAT 253 Finnigan, Germany) in dual inlet mode. The measurement reproducibility for two replicates is 0.015‰, 0.010‰ and 0.005‰ for $\delta^{17}O$, $\delta^{18}O$ and $\Delta^{17}O$, respectively.

### 3.5. Carbon dioxide extraction and isotope analysis

$CO_2$ was extracted from the air samples in a system made from electropolished stainless steel (Supplementary Figure S5). Our system used four commercial traps (MassTech, Bremen, Germany) which consist of a 1/8''inlet tube inserted within a ¼'' tube that is closed at the bottom. The first two traps were operated at dry ice temperature (-78°C) to remove moisture and some organics. The other two traps were operated at liquid nitrogen temperature (-196°C) to trap $CO_2$. The flow rate during extraction was 55 mL min$^{-1}$, controlled by a mass flow controller (Brooks Instruments, Holland). After processing usually about 2L of air, the remaining air was evacuated and the extracted $CO_2$ (together with $N_2O$ and potentially other condensable gases) was cryogenically transferred into a break seal tube from which it could later be liberated for isotopic measurement. The reproducibility of the extraction system was 30 parts per million (ppm) for $\delta^{18}O$ and 7 ppm for $\delta^{13}C$ determined on 14 extractions (1σ standard deviation for the 14 extractions, Supplementary Table S1).

The extracted $CO_2$ was first measured for $\delta^{13}C$ and $\delta^{18}O$ with a Delta$^{Plus}$XL isotope ratio mass spectrometer (IRMS) (Thermo Finnigan, Germany) in dual inlet mode. $N_2O$ was not separated from $CO_2$ in our system and we apply constant corrections of 0.2‰ for $\delta^{13}C$ and 0.3‰ for $\delta^{18}O$ to correct for the $N_2O$ interference as suggested by (Mook and Hoek, 1983). After the isotope measurement, the remaining gas in the bellow of the IRMS was frozen back into the break seal tube for the measurement of $\Delta^{17}O$. The $\Delta^{17}O$ of $CO_2$ was determined using the $CO_2$-$O_2$ exchange method (Mahata et al., 2013;Barkan et al., 2015;Adnew et al., 2019). A detailed description of the $CO_2$-$O_2$ exchange system at Utrecht University is given in (Adnew et al., 2019) and the method is only described here briefly. Equal amounts of $CO_2$ and $O_2$ were mixed in a quartz reactor containing a platinum sponge catalyst at the bottom and heated at 750°C for 2hrs. After isotope equilibration, the $CO_2$ was trapped at liquid nitrogen temperature, while the $O_2$ was collected with 1 pellet of 5Å molecular sieve (1.6 mm, Sigma Aldrich, USA) at liquid nitrogen temperature. The isotopic composition of the isotopically equilibrated $O_2$ was measured with a Delta$^{Plus}$XL isotope ratio mass spectrometer in dual inlet mode with reference to a pure $O_2$ calibration gas that has been assigned values of $\delta^{17}O$ = 9.254‰ and $\delta^{18}O$ = 18.542‰ by measurements of multiple aliquots by E. Barkan at the Hebrew University of Jerusalem. The reproducibility of the $\Delta^{17}O$ measurement was better than 10 ppm (Supplementary Table S1).

### 3.6. Leaf cuvette model

We used a simple leaf cuvette model to evaluate the dependence of $\Delta_A\Delta^{17}O$ on key parameters. In this model, the leaf is partitioned into three different compartments: the intercellular air space, the mesophyll cell, and the chloroplast, as shown in supplementary material Figure S6. For the calculations with this model, we assumed an infinite boundary layer conductance. The detailed description of the model and the python code is given in the supplementary material.



In the leaf cuvette model, we used a 100 ppm downdraw of $CO_2$, similar to the leaf exchange experiments, i.e., the $CO_2$ mole fraction decreases from 500 ppm in the entering air ($c_e$) to 400 ppm in the outgoing air ($c_o$), which is identical to the air surrounding the leaf (ca) as a result of thorough mixing in the cuvette. The assimilation rate is set to 20.0 μmol m$^{-2}$s$^{-1}$. The leaf area and flowrate of air are set to 30 cm$^2$ and 0.7 L min$^{-1}$, respectively. The isotope composition of leaf water at the site where the $H_2O$-$CO_2$ exchange occurs is $\delta^{17}O$ = 5.39‰ and $\delta^{18}O$ = 10.648‰, which is the mean of the measured $\delta^{17}O$ and $\delta^{18}O$ values of bulk leaf water in our experiments. The leaf water temperature is set to 22°C (similar to the experiment). In the model, the $\delta^{18}O$ of the $CO_2$ entering the cuvette is set to 30.47‰ for all the simulations, as in the normal $CO_2$ experiments, but the assigned $\Delta^{17}O$ values ranges from -0.5‰ to 0.5‰ which encompasses both the stratospheric intrusion and combustion components. The corresponding $\delta^{17}O$ of the $CO_2$ entering the cuvette is calculated from the assigned $\delta^{18}O$ value (30.47‰) and $\Delta^{17}O$ values (-0.5‰ to 0.5‰). The schematic of the leaf cuvette model is shown in Figure S6 (supplementary material).

## 4. Results

### 4.1. Discrimination against $^{18}O$ and $^{17}O$ of $CO_2$

$^{17}O$ and $^{18}O$-photosynthetic discrimination ($\Delta_A^{17}O$ and $\Delta_A^{18}O$) for sunflower, ivy, and maize as a function of the $c_m/c_a$ ratio is shown in Figure 4. $\Delta_A^{17}O$ and $\Delta_A^{18}O$ vary with $c_m/c_a$ (the $c_m$ is calculated using $^{18}O$ isotope measurement of $CO_2$, see section 2.4) for all plant species investigated. For sunflower, we observe $\Delta_A^{18}O$ values between 29‰ and 64‰ for $c_m/c_a$ between 0.54 and 0.86. Ivy shows a relatively little variation of $\Delta_A^{18}O$ around a mean of 22‰ for $c_m/c_a$ between 0.48 and 0.58. For maize, $\Delta_A^{18}O$ is lower than for the C$_3$ plants measured in this study, with values between 10‰ and 20‰ for $c_m/c_a$ between 0.15 and 0.37. As expected, for all species the behavior for $\Delta_A^{17}O$ is very similar to the one for $\Delta_A^{18}O$ (Figure 4b).

For sunflower changing the irradiance from 300 μmol m$^{-2}$s$^{-1}$ (low light, hereafter LL) to 1200 μmol m$^{-2}$s$^{-1}$(high light, hereafter HL) causes average decreases of 12‰ for $\Delta_A^{17}O$ and 22‰ for $\Delta_A^{18}O$. For maize, the changes are only 2.2‰ for $\Delta_A^{17}O$ and 4.4‰ for $\Delta_A^{18}O$. For ivy, changing the light intensity does not significantly change the observed $\Delta_A^{17}O$ and $\Delta_A^{18}O$. The blue diamond points in Fig. 4 show results for $\Delta_A^{18}O$ and $\Delta_A^{17}O$ calculated using Farquhar model (Farquhar and Lloyd, 1993) (equation 7 for $\Delta_A^{18}O$). Overall, there is a good agreement between the calculated and the measured discrimination, but for the highest discriminations (high $c_m/c_a$ ratios, LL experiments of sunflower) the calculations slightly underestimate the measured values. The Farquhar model fits well for both $\Delta_A^{17}O$ and $\Delta_A^{18}O$ with ($R^2$, root mean square error (RMSE)) values of (0.993,0.6‰) for $\Delta_A^{17}O$ and (0.994, 1.2‰), for $\Delta_A^{18}O$, respectively. The solid lines in Figure 4 show results of leaf cuvette model calculations, where the dependence of $\Delta_A^{17}O$ and $\Delta_A^{18}O$ on $c_m/c_a$ is explored for a set of calculations with otherwise fixed parameters. The model shows a good agreement with the experimental results except for ivy, where the model overestimates the discrimination.

### 4.2. Discrimination against $^{17}O$-excess of $CO_2$



The discrimination of photosynthesis against the $^{17}O$-excess ($\Delta_A\Delta^{17}O$) of $CO_2$ is shown in Figure 5. $\Delta_A\Delta^{17}O$ was negative for all experiments and it depends strongly on the $c_m/c_a$ ratio. For sunflower and ivy, $\Delta_A\Delta^{17}O$ is also strongly dependent on the $\Delta^{17}O$ of $CO_2$ supplied to the cuvette, whereas no significant dependence is found for maize. The leaf cuvette model results illustrate the shape of the dependence on the $c_m/c_a$ ratio and agree well with the experiments. For the leaf cuvette model, the $\Delta^{17}O$ value of the water is assigned a constant value of -0.122‰ (average $\Delta^{17}O$ value for the bulk leaf water). Results from the Farquhar model (equation 7 for $\Delta_A^{18}O_{FM}$ and analogous equation for $\Delta_A^{17}O_{FM}$) fit well with the measurements ($R^2 = 0.959$, RMSE = 0.1‰) (Figure 5a, Figure S7 (supplementary material)). The RMSE is lower than the measurement error for the $\Delta_A\Delta^{17}O$ in our experimental setup. Based on our measurement, the error introduced in $\Delta_A\Delta^{17}O$ for the individual experiment is 0.25‰ (SD) calculated from the individual errors of $\Delta_A^{17}O$ and $\Delta_A^{18}O$.

Figure 5b shows the same values of $\Delta_A\Delta^{17}O$ as a function of the difference between $\Delta^{17}O$ of $CO_2$ entering the leaf and $\Delta^{17}O$ of leaf water at the evaporation site where $CO_2$-$H_2O$ exchange takes place ($\Delta^{17}O_a$ - $\Delta^{17}O_{wes}$), for different $c_m/c_a$ ratios. The leaf cuvette model results (solid lines in Figure 5b) suggest a linear dependence between $\Delta_A\Delta^{17}O$ and ($\Delta^{17}O_a$ - $\Delta^{17}O_{wes}$). The experimental results agree with the hypothesis that $\Delta_A\Delta^{17}O$ is linearly dependent on $\Delta^{17}O_a$ - $\Delta^{17}O_{wes}$ at a certain $c_m/c_a$ ratio. Figure 6 shows the corresponding relation where $\Delta_A\Delta^{17}O$ is divided by $\Delta^{17}O_a$-$\Delta^{17}O_m$. All the values follow the same relationship with $c_m/c_a$ ratio with an exponential function (equation 19). This function quantifies the dependence of $\Delta_A\Delta^{17}O$ on $c_m/c_a$, and thus the effect of the diffusion of isotopically exchanged $CO_2$ back to the atmosphere, which increased with increasing $c_m/c_a$ ratio.

$$\frac{\Delta_A\Delta^{17}O}{\Delta^{17}O_a - \Delta^{17}O_m} = -0.150 \times \exp(3.707 \times c_m/c_a) + 0.028 \qquad (19)$$

Figure 7 shows results from the leaf cuvette model that illustrates in more detail how $\Delta^{17}O_e$ and $\Delta^{17}O_{wes}$ affect $\Delta^{17}O_a$ and $\Delta_A\Delta^{17}O$ and their dependence on $c_m/c_a$. At lower $c_m/c_a$, only a very small fraction of $CO_2$ that has undergone isotopic equilibration in the mesophyll diffuses back to the atmosphere, and therefore $\Delta^{17}O_a$ stays close to the incoming $\Delta^{17}O_e$, modified by the fractionation during $CO_2$ diffusion through the stomata. Figure 7c confirms that indeed at low $c_m/c_a$, $\Delta_A\Delta^{17}O$ approaches the fractionation constant expected for diffusion, -0.170‰. This diffusional fractionation is independent of the isotopic composition of the $CO_2$ entering the leaf, and therefore at low $c_m/c_a$, the $\Delta_A\Delta^{17}O$ curves for the different values of the anomaly of the $CO_2$ entering the leaf converge. For a high $c_m/c_a$ ratio, the back-diffusion of $CO_2$ that has equilibrated with water becomes the dominant factor, and in this case, the isotopic composition of the outgoing $CO_2$ converges towards this isotope value, independent of the isotopic composition of the incoming $CO_2$ (Figure 7a). This can lead to a very wide range of values for the discrimination against $\Delta^{17}O$, because now the effect on $\Delta^{17}O$ of the ambient $CO_2$ depends strongly on the difference in isotopic composition between incoming $CO_2$ and $CO_2$ in isotopic equilibrium with the leaf water.





In the model calculations shown in Fig. 7b and d, the isotopic composition of the water was changed in the model from $\Delta^{17}O_{wes}$ = -0.122‰ to 0.300‰, whereas all other parameters were kept the same. The value of $\Delta^{17}O_e$ for which $\Delta^{17}O_a$ does not depend on $c_m/c_a$ is shifted accordingly, again being similar to $\Delta^{17}O_m$. At low $c_m/c_a$ $\Delta_A\Delta^{17}O$ converges to the same value as in Fig 7 c), confirming the role of diffusion
into the stomata as discussed above.

Figure 8 shows how $\delta^{18}O$ and $\Delta^{17}O$ varied in key compartments of the leaf cuvette system that determine the oxygen isotope effects associated with photosynthesis. The irrigation water has a $\Delta^{17}O$ value of 0.017. The measured bulk leaf water is 6-16‰ enriched in $^{18}O$ and its $\Delta^{17}O$ value is lower by -0.075 to -0.200‰
(mean value -0.121‰) than the irrigation water, calculated using a three-isotope slope of $\lambda_{trans}$ = 0.516 at 80% humidity (Landais et al., 2006). $\Delta^{17}O$ of leaf water at the evaporation site, calculated from the transpired water, has slightly lower $^{17}O$-excess, with values between -0.119‰ and -0.237 (average -0.184‰). Note that the bulk leaf water was not measured for all the experiments. For the experiments where the bulk leaf water is measured, $\Delta^{17}O$ of leaf water at the evaporation site ranges from -0.160‰ to
-0.231 with an average value of -0.190 ± 0.020 ‰. The calculated isotopic composition of water at the exchange site was thus similar, but slightly lower in $\Delta^{17}O$ than the values measured for bulk leaf water. $CO_2$ exchanges with the water in the leaf with a well-established fractionation constant (equation 18) and a three-isotope slope of $\lambda_{CO2-H2O}$ = 0.5229 (Barkan and Luz, 2012), leading to the lower $\Delta^{17}O$ values of the equilibrated $CO_2$. In our experiments, the $\Delta^{17}O$ value of $CO_2$ in equilibrium with leaf water is lower
than the $\Delta^{17}O$ value of $CO_2$ entering the leaf. The $\Delta^{17}O$ of the $CO_2$ in the intercellular air space between the two end members (the $\Delta^{17}O$ of the $CO_2$ entering the leaf and $\Delta^{17}O$ of the $CO_2$ in equilibrium with leaf water). This explains why the observed values of $\Delta_A\Delta^{17}O$ are negative for the experiments performed in this study.

**5.   Discussion**

5.1.  **Discrimination against $^{17}O$ and $^{18}O$ of $CO_2$**

The higher $\Delta_A^{18}O_{obs}$ and $\Delta_A^{17}O_{obs}$ values for sunflower compared to maize and ivy (Figure 4) is mainly due to a higher back-diffusion flux ($c_m/(c_a-c_m)$). The back-diffusion flux is higher for sunflower and ivy
($C_3$ plants) than for maize ($C_4$ plant), a consequence of the lower stomatal conductance (Gillon and Yakir, 2000b;Barbour et al., 2016) and higher assimilation rate of $C_4$ plants. In $C_4$ plants most of the $CO_2$ entering the stomata is carboxylated by PEPC resulting in a lower $CO_2$ mixing ratio in the mesophyll which results in a lower back-diffusion flux. The increase of assimilation rate with higher light intensity also explains the decreases of $\Delta_A^{18}O_{obs}$ and $\Delta_A^{17}O_{obs}$ with decreasing $c_m/c_a$ ratio for maize and sunflower, which is
observed most clearly for sunflower. A similar trend of increase in $\Delta_A^{18}O_{obs}$ with an increase in $c_m/c_a$ ratio has been reported in previous studies (Gillon and Yakir, 2000a, b;Osborn et al., 2017). For ivy, $\Delta_A^{18}O_{obs}$ and $\Delta_A^{17}O_{obs}$ do not decrease with an increase in irradiance. The change in assimilation rate with irradiance is small, thus $CO_2$ mole fraction in the mesophyll cell will not decrease strongly and the effect on the back diffusion is smaller than the variability in $\Delta_A^{18}O_{obs}$ and $\Delta_A^{17}O_{obs}$ of different leaves of the
same plant.



The leaf cuvette model results shown in Figure 4 agree well with the measurements for sunflower and maize, but overestimate $\Delta_A{}^{18}O_{obs}$ and $\Delta_A{}^{17}O_{obs}$ for ivy. This is due to relatively higher $\delta^{17}O$ and $\delta^{18}O$ values of leaf water used in the leaf cuvette model calculations than the $\delta^{17}O$ and $\delta^{18}O$ values at the evaporation site. The Farquhar model (Equation 7) uses the individual values for each experiment, and agrees well with the experimental results for ivy, confirming that this is the cause of the discrepancy in the cuvette model.

In our experiments, photosynthesis causes enrichment in $\delta^{17}O$ and $\delta^{18}O$ of atmospheric $CO_2$ for both $C_3$ and $C_4$ plants, i.e. positive values of $\Delta_A{}^{17}O$ and $\Delta_A{}^{18}O$. In principle, $\Delta_A{}^{17}O$ and $\Delta_A{}^{18}O$ can also be negative if the $\delta^{17}O_m$ and $\delta^{18}O_m$ are depleted relative to the ambient $CO_2$. This is in contrast to $\Delta_A{}^{13}C$, which will always be positive since it is determined by the fractionation due to the PEPC and RuBisCO enzyme activity (Figure S8 and S9, supplementary material). In general, in our experiments, the values for $\Delta_A{}^{17}O$ and $\Delta_A{}^{18}O$ are about five times larger than the relative difference between the $\delta^{17}O$ and $\delta^{18}O$ of the $CO_2$ entering and leaving the cuvette, respectively (Figure S10 and S11, supplementary material). This is easy to understand from the definition of $\Delta_A$. Taking $\Delta_A{}^{18}O$ as an example, $\Delta_A{}^{18}O_{obs} = \frac{\zeta(\delta^{18}O_a - \delta^{18}O_e)}{1 + \delta^{18}O_a - \zeta(\delta^{18}O_a - \delta^{18}O_e)} \approx \zeta(\delta^{18}O_a - \delta^{18}O_e)$ and in our experiments, $\zeta = c_e / (c_e - c_a) \approx 500 / (500-400) = 5$.

### 5.2. Discrimination against the $^{17}O$-excess of $CO_2$

Unlike ivy and sunflower, maize does not show a significant change in $\Delta_A\Delta^{17}O$ when $CO_2$ gases with different $\Delta^{17}O$ are supplied to the plant. The $C_4$ plant maize has a small back-diffusion flux due to its high assimilation rate and low stomatal conductance, leading to a low $c_m/c_a$ ratio. At these low $c_m/c_a$ ratios, $\Delta_A{}^{18}O$ and $\Delta_A{}^{17}O$ (equation 7 for $\Delta_A{}^{18}O$) are close to the weighted fractionation due to diffusion through boundary layer and stomata, $\Delta_A{}^{18}O = a_{18bs}$ and $\Delta_A{}^{17}O = a_{17bs}$ (Appendix 3 equation A3.3 and A3.9, respectively). As a result, $\Delta_A\Delta^{17}O$ of $CO_2$ is dominated by the fractionation due to diffusion (Figure 7 c and d, inset). In general, the effect of diffusion on $\Delta^{17}O$ of atmospheric $CO_2$ can be expressed as follows:

$$\Delta^{17}O_{Modified} = \Delta^{17}O_a + \left(\lambda_{ref} - \lambda_{diffusion}\right) \times \ln\alpha_{diffusion} \qquad (20)$$

where $\Delta^{17}O_a$ is the $\Delta^{17}O$ of the $CO_2$ surrounding the leaf, $\Delta^{17}O_{modified}$ is the $\Delta^{17}O$ of the $CO_2$ modified due to diffusional fractionation and $\lambda_{diffusion}$, $\lambda_{ref}$ and $\alpha_{diffusion}$ are the oxygen three-isotope relationships during diffusion from the $CO_2$-$H_2O$ exchange site to the atmosphere, the reference slope used and the fractionation against $^{18}O$ for $CO_2$ during diffusion through the stomata. Using the values $\lambda_{RL} = 0.528$, $\lambda_{diffusion} = 0.509$ (Young et al., 2002) and $\alpha_{diffusion}=0.9912$ (Farquhar and Lloyd, 1993), the effect of diffusional fractionation on the $\Delta^{17}O$ of atmospheric $CO_2$ is -168 ppm regardless of the anomaly of the $CO_2$ entering the leaf, and the model results confirm this at low $c_m/c_a$ ratio (Figure 7 c and d, inset).

At a high $c_m/c_a$ ratio, $\Delta^{17}O_a$ is dominated by the back diffusion of $CO_2$ that has equilibrated with water to the cuvette. As a consequence, $\Delta^{17}O_a$ converges to a common value that is independent of the anomaly of





the $CO_2$ entering the cuvette and is determined by the isotopic composition of leaf water. The end member appears to be equal to the $\Delta^{17}O$ of $CO_2$ in equilibrium with leaf water, $\Delta^{17}O_m$ (Fig. 7). When $\Delta^{17}O_a$ = $\Delta^{17}O_m$, $\Delta^{17}O_a$ does not change with $c_m/c_a$, indicating that in this case the $\Delta^{17}O$ of the $CO_2$ diffusing back
from the leaf is the same as the $\Delta^{17}O$ $CO_2$ entering the leaf.

$\overline{a}_{18}$ is the overall discrimination occurring during the diffusion of $^{12}C^{18}O^{16}O$ from the ambient air to the $CO_2$-$H_2O$ exchange site. In our study $\overline{a}_{18}$ ranges from 5‰ to 7.2‰, lower than the literature estimate of 7.4‰ (Farquhar et al., 1993). $\overline{a}_{18}$ depends on the ratio of stomatal conductance, which is associated with a strong fractionation of 8.8‰ to mesophyll conductance with an associated fractionation of only 0.8‰.
Therefore, the higher the ratio ($g_s/g_{m18}$) the lower the $\overline{a}_{18}$ (Table S2, supplementary material). The difference in $\overline{a}_{18}$ of 2.4‰ between the literature value of 7.4‰ and the lowest $\overline{a}_{18}$ estimate in this study will introduce an error of only 46 ppm in the $\Delta^{17}O$ value (see equation 19). The uncertainty $\overline{a}_{18}$ has lower influence on the $\Delta_A\Delta^{17}O$ of $C_3$ plants compared to $C_4$ plants since the diffusional fractionation is less important at the higher $c_m/c_a$ ratio where $C_3$ plants operate.

### 5.3. Global average value of $\Delta_A\Delta^{17}O$ and $\Delta^{17}O$ isoflux

We can use the established relationship between $\Delta_A\Delta^{17}O$ and $\Delta^{17}O_a$ - $\Delta^{17}O_{wes}$ for a certain $c_m/c_a$ ratio to provide a leaf-scale based bottom-up estimate for the global effect of photosynthesis on $\Delta^{17}O$ in atmospheric $CO_2$. For this, we use results from a recent modeling study, which provides global average
values for $CO_2$ and leaf water ($\Delta^{17}O(CO_2)$ = -0.168‰, $\Delta^{17}O(H_2O_{-leaf})$ = -0.067‰; (Koren et al., 2019); Figure S12 and 13, supplementary material). The $\Delta^{17}O(CO_2)$ values agree well with the limited amount of available measurements (Table 1).

To extrapolate $\Delta_A\Delta^{17}O$ determined in the leaf scale experiments to the global scale, $c_m/c_a$ ratios of 0.7 and
0.3 are used for $C_3$ and $C_4$ plants, respectively, similar to previous studies (Hoag et al., 2005). From SIBCASA model results we obtained an annual variability of $c_i/c_a$ values with a standard deviation of 0.12 and 0.17 for $C_4$ and $C_3$ plants respectively (Figure S14, supplementary material) (Schaefer et al., 2008;Koren et al., 2019). We assigned this variability worst case estimates for the error in $c_m/c_a$ as shown in the light orange and light pink shaded areas in figure 5b. Based on the linear dependency of $\Delta_A\Delta^{17}O$
and $\Delta^{17}O_a$-$\Delta^{17}O_{wes}$, we estimated the $\Delta_A\Delta^{17}O$ for tropospheric $CO_2$ based on the $\Delta^{17}O$ of leaf water and $c_m/c_a$ ratio. In Figure 5b, the dashed black vertical line indicates $\Delta^{17}O_a$-$\Delta^{17}O_{wes}$ obtained from the 3D global model (Koren et al., 2019). The results of the global estimate and parameters used for the extrapolation of leaf scale study to the global scale are summarized in Table 1.

The $\delta^{17}O$ value of atmospheric $CO_2$ (21.53‰) is calculated from the global $\delta^{18}O$ and $\Delta^{17}O$ values ( 41.5‰ and -0.168‰, respectively) (Koren et al., 2019). The $\delta^{17}O$ and $\delta^{18}O$ values of global mean leaf water are calculated from the soil water. A global mean $\delta^{18}O$ value of soil water is -8.4‰ assuming soil water is similar to precipitation (Bowen and Revenaugh, 2003;Koren et al., 2019). The $\delta^{17}O$ value of soil water is -4.4‰, calculated using equation 20 (Luz and Barkan, 2010).



$$\ln(\delta^{17}O_{soil} + 1) = 0.528 \times \ln(\delta^{18}O_{soil} + 1) + 0.033 \qquad (21)$$

$\delta^{17}O$ and $\delta^{18}O$ of leaf water are calculated from $\delta^{17}O$ and $\delta^{18}O$ of soil water with fractionation factors of 1.0043 and 1.0084 and, respectively (Hofmann et al., 2017;Koren et al., 2019). The fractionation factor for $\delta^{17}O$ is calculated using $\alpha^{17} = (\alpha^{18})^{\lambda_{trans}}$ with a three-isotope exponent for transpiration of $\lambda_{trans}$ =0.516, assuming relative humidity to be 80% (Landais et al., 2006). The $\delta^{17}O$ and $\delta^{18}O$ values of global
mean leaf water are then -0.136‰ and -0.131‰, respectively. Thus, the difference between global atmospheric $CO_2$ and leaf water is $\delta^{17}O_{CO2 - water} = 21.666‰$ and $\delta^{18}O_{CO2 - water} = 41.631‰$. This yields $\Delta^{17}O_{CO2 - water} = -0.101‰$, and this value is indicated as dashed black line in figure 4. The grey shaded area indicates the propagated error using the standard deviation of the relevant parameters in 180 x 360 grid boxes for 12 months of leaf water and 45 x 60 grid boxes for 24 months for $CO_2$ (Koren et al., 2019).
In Fig. 5b, the intersection between the dashed black vertical line and the discrimination lines for the representative $c_m/c_a$ ratios of $C_3$ and $C_4$ plants correspond to the $\Delta_A\Delta^{17}O$ value of $C_3$ and $C_4$ plants. For $C_4$ plants ($c_m/c_a = 0.3$) this yields $\Delta_A\Delta^{17}O = -0.3‰$ (gray dashed line in Figure 5b) and for $C_3$ plants ($c_m/c_a = 0.7$), $\Delta_A\Delta^{17}O = -0.65‰$ (black dashed line in Figure 5b).

Three main factors contribute to the uncertainty of the extrapolated $\Delta_A\Delta^{17}O$ value. The first is due to measurement error which contributes 0.25‰ (standard error for individual experiments). The second factor is the uncertainty in the difference between $\Delta^{17}O$ of atmospheric $CO_2$ and leaf water. Statistics for all 45 x 60 grid boxes for 24 months (2012-2013) show a range of -0.218‰ to -0.151‰ for $\Delta^{17}O$ of atmospheric $CO_2$ with a mean of -0.168‰ and a standard deviation of 0.013‰ (Figure S12,
supplementary material). Statistics for all 180 x 360 grid boxes for 12 months show a range of -0.236‰ and -0.027‰ for $\Delta^{17}O$ of the leaf water (Figure S13, supplementary material). The mean is -0.067‰ with a standard deviation of 0.041‰. The third uncertainty in the extrapolation of $\Delta^{17}O$ comes from the uncertainty in the $c_m/c_a$ ratio. For $C_3$ and $C_4$ plants, these errors are indicated by the light orange and light blue shadings in Figure 5b.
Taking these uncertainties into account leads to a mean value of $\Delta_A\Delta^{17}O = -0.3\pm0.18‰$ for $C_4$ plants and. $\Delta_A\Delta^{17}O = -0.65\pm0.18‰$ for $C_3$ plants. Using assimilation weighted fractions of 23% for $C_4$ and 77% for $C_3$ vegetation (Still et al., 2003), the global mean value of $\Delta_A\Delta^{17}O$ obtained from equation 14 is -0.57±0.14‰. The $\Delta_A\Delta^{17}O$ isoflux due to photosynthesis is calculated using a GPP value of 120 PgCyr$^{-1}$
(Beer et al., 2010) and A=0.88×GPP, resulting in an isoflux of -60±15‰ PgCyr$^{-1}$ globally. This is the first global estimate of $\Delta_A\Delta^{17}O$ based on direct measurements of the discrimination during assimilation. Our value is in good agreement with the previous model estimates. (Hofmann et al., 2017) estimated an isoflux ranging from -42 to -92‰PgCyr$^{-1}$ (converted to a reference line with $\lambda$=0.528) using an average $c_m/c_a$ ratio of 0.7 for both $C_4$ and $C_3$ plants and $\Delta^{17}O$ of -0.147‰ for atmospheric $CO_2$. A previous model-
estimated value (Hoag et al., 2005) is -47‰PgCyr$^{-1}$(with $\lambda$=0.528), derived with a more simple model and using $\Delta^{17}O$ of -0.146‰ with $c_m/c_a$ ratio of 0.33 and 0.66 for $C_4$ and $C_3$ plants, respectively.

The main uncertainty in the extrapolation of $\Delta_A\Delta^{17}O$ from the leaf experiments to the global scale is the uncertainty in the $c_m/c_a$ ratio. The error from the uncertainty in $c_m/c_a$ ratio increases when the relative





difference in $\Delta^{17}O$ between $CO_2$ and leaf water increases (Figure 5b). It is difficult to determine a single
        representative $c_m$ value for different plants because this value would need to be properly weighted with
        temperature, irradiance, $CO_2$ mole fraction and other environmental factors (Flexas et al.,
        2008;2012;Shrestha et al., 2019). Recent developments in laser spectroscopy techniques (McManus et
        al., 2005;Nelson et al., 2008;Tuzson et al., 2008;Kammer et al., 2011) might enable more and easier
measurements of $c_m/c_a$ both in the laboratory and under field conditions. This could lead to a better
        understanding of variations in the $c_m/c_a$ ratio among plant species and, temporally, spatially and
        environmentally.

## 6. Conclusions

In order to investigate the effect of photosynthetic gas exchange on the $\Delta^{17}O$ of atmospheric $CO_2$ at the
        leaf level, gas exchange experiments were carried out with isotopically normal and slightly anomalous
        ($^{17}O$-enriched) $CO_2$ in leaf cuvettes using two $C_3$ plants (sunflower and ivy) and one $C_4$ plant (maize).
        Results for $^{13}C$, $^{17}O$ and $^{18}O$ agree with results reported in the literature previously. For $\Delta^{17}O$, our
        experiments confirm that two parameters determine the effect of photosynthesis on $CO_2$: 1) the $\Delta^{17}O$
difference between the incoming $CO_2$ and $CO_2$ in equilibrium with leaf water and 2) the $c_m/c_a$ ratio, which
        determines the degree of back-flux of isotopically exchanged $CO_2$ from the mesophyll to the atmosphere.
        In addition, at low $c_m/c_a$ ratios, $\Delta_A\Delta^{17}O$ is mainly influenced by the diffusional fractionation. Under our
        experimental conditions, the isotopic effect increased with $c_m/c_a$, e.g. $\Delta_A\Delta^{17}O$ was -0.3‰ and -0.65‰ for
        maize and sunflower with $c_m/c_a$ ratios of 0.3 and 0.7, respectively. However, experiments with mass
independently fractionated $CO_2$ demonstrate that the results depend strongly on the $\Delta^{17}O$ difference
        between the incoming $CO_2$ and $CO_2$ in equilibrium with leaf water. This is supported by calculations with
        a leaf cuvette model. Our results confirm that the formalism developed by Farquhar and others is also
        applicable to the evaluation of $\Delta^{17}O$.

Results from the leaf exchange experiments were upscaled to the global atmosphere using modeled values
        for $\Delta^{17}O$ of leaf water and $CO_2$, which results in $\Delta_A\Delta^{17}O$ = -0.57± 0.14‰ and a value for the $\Delta^{17}O$ isoflux
        of -60± 15‰ PgCyr$^{-1}$. This is the first study that provides such an estimated based on direct leaf chamber
        measurements, and the results agree with previous $\Delta^{17}O$ calculations. The largest contribution to the
        uncertainty originates from uncertainty in the $c_m/c_a$ ratio and the largest contributions to the isoflux come
from $C_3$ plants, which have both a higher share of the total assimilation and higher discrimination. $\Delta_A\Delta^{17}O$
        is less sensitive to $c_m/c_a$ ratios at lower values of $c_m/c_a$, for instance for $C_4$ plants, maize.

        $\Delta^{17}O$ of tropospheric $CO_2$ is controlled by photosynthetic gas exchange, respiration, soil invasion, and
        stratospheric influx. The stratospheric flux is well established and the effect of photosynthetic gas
exchange can now be quantified more precisely. To untangle the contribution of each component to the
        $\Delta^{17}O$ atmospheric $CO_2$ we recommend measuring the effects of foliage respiration and soil invasion both
        in the laboratory and at the ecosystem scale.

Code and data availability.



The data used in this study are included in the paper either with figures or tables. The python code for the cuvette model is provided as supplementary material.

Author contributions.

GAA and TR designed the main idea of the study. GAA and TP designed the leaf cuvette setup. TP monitors plant growth. GAA and TR designed the $CO_2$ extraction and $CO_2$-$H_2O$ exchange system. GAA conducted all the measurements. GK provided the leaf cuvette model. WP enabled the work within the ASICA project. All authors discussed the results at different steps of the project. GAA and TR prepared the manuscript with contributions from all the co-authors.

Competing interests.

The authors declare that they have no conflict of interest

**Acknowledgments**

The authors thank Leonard I. Wassenaar and Stefan Terzer-Wassmuth from the International Atomic and Energy Agency, Vienna for supplying water standards. The authors thank Eugeni Barkan and Rolf Vieten from the Hebrew University of Jerusalem for calibration of our $O_2$ and $CO_2$ working gases. We are grateful to Amaelle Landais from Laboratoire des Sciences Du Climat et de l'Environnement Université Paris-Saclay for measuring the $\Delta^{17}O$ of leaf water samples for our study. The authors thank Amzad Laskar for useful discussion during the design of the experiment. This work is funded by the EU ERC project ASICA.

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

## Appendix A1

Leaf exchange parameters are calculated following (Caemmerer and Farquhar, 1981). The transpiration
rate ($E$) is calculated from the air flowrate, leaf area and concentration of water vapor entering and leaving the cuvette as:

$$E = \frac{u_e}{s} \times \left(\frac{w_a - w_e}{1 - w_a}\right) \qquad (A1.1)$$

where $w_a$, $w_e$, are the mole fractions of water leaving ($a$) or entering ($e$) the cuvette, $u_e$ is flowrate of air entering the cuvette and s is the leaf surface area. The assimilation rate ($A$) is calculated as:



$$A = \frac{u_e}{s} \times \left( c_e - c_a \times \left( \frac{1 - w_e}{1 - w_a} \right) \right) \tag{A1.2}$$

where $c_e$ and $c_a$ are the mole fractions of $CO_2$ leaving and entering the cuvette. The total conductance for water vapor ($g_{wa}{}^t$) is calculated as:

$$g_{wa}^t = E \times \left( \frac{1 - \left( \frac{w_i + w_a}{2} \right)}{w_i - w_a} \right) \tag{A1.3}$$

where $w_i$ and $w_a$ are the water vapor mole fraction in the intercellular air space (calculated assuming saturation at ambient temperature) and the mole fraction of water vapor leaving the cuvette. The mole fraction of $CO_2$ in the intercellular air space is calculated as:


$$c_i = \frac{\left( g_{ac}^t - \frac{E}{2} \right) \times c_a - A}{\left( g_{ac}^t - \frac{E}{2} \right)} \tag{A1.4}$$

where $g_{ca}{}^t$ is the total conductance for $CO_2$. For a detailed derivation of the leaf exchange parameters, the reader is referred to (Caemmerer and Farquhar, 1981).

## Appendix A2

### Isotopic composition of water at the evaporation site


Using mass balance between the air entering and leaving the cuvette, the $\delta^{18}O$ of the transpired ($\delta^{18}O_{trans}$) water is calculated according to (Harwood et al., 1998):

$$\delta^{18}O_{trans} = \left( \frac{w_a}{w_a - w_e} \right) \times \left( \delta^{18}O_{wa} - \delta^{18}O_{we} \right) + \delta^{18}O_{we} \tag{A2.1}$$

where $\delta^{18}O_{we}$ and $\delta^{18}O_{wa}$ are $\delta^{18}O$ values of water vapor entering and leaving the cuvette and $w_a$ and $w_e$ are the mole fractions of water vapor entering and leaving the cuvette. $\delta^{17}O$ is calculated based on the triple isotope relationship for transpiration, $\alpha^{17} = (\alpha^{18})^{\lambda_{trans}}$ where $\lambda_{trans} = 0.522 - 0.008 \times h$ (Landais et al., 2006). $h$ is relative humidity, $0.3 \leq h \leq 1$, which is calculated as $h = \frac{w_a}{w_i}$, $w_i$ is the saturation mole fraction of water vapor in the intercellular air space.


Leaf water at the site of evaporation is enriched during evaporation and/or transpiration since the heavier isotopologues diffuse slower than the lighter ones (Flanagan et al., 1991;Farquhar et al., 1993;Flanagan, 1993;Yakir and Sternberg, 2000). The degree isotopic enrichment due to the phase change from water to vapor (evaporation) and diffusion is described by the modified Craig and Gordon model (Craig and

Gordon, 1965) including resistance to boundary layer and stomata diffusion as described by (Farquhar



et al., 1989b;Flanagan et al., 1991;Flanagan, 1993). Measurement of the isotopic composition of air entering and leaving the cuvette allows determining the isotopic composition of water at the evaporation site even if it is not in steady state as described in (Farquhar et al., 1989b;Flanagan et al., 1991;Harwood et al., 1998). The $\delta^{18}O$ of leaf water at the site of evaporation ($\delta^{18}O_{wes}$) is:


$$\delta^{18}O_{wes} = \delta^{18}O_{trans} + \varepsilon^{18}_k + \varepsilon^{18}_{equ} + \frac{w_a}{w_i} \times (\delta^{18}O_{wa} - \varepsilon^{18}_k + \delta^{18}O_{trans}) \qquad (A2.2)$$

where $\varepsilon^{18}_k$ and $\varepsilon^{18}_{equ}$ are the kinetic fractionation of water vapor in air and the equilibrium fractionation between liquid and gas phase water, respectively. The equilibrium fractionation is temperature dependent (Bottinga and Craig, 1968) and calculated as:

$$\varepsilon^{18}_{equ} = 2.644 - 3.206 \times \left(\frac{10^3}{T}\right) + 1.534 \times \left(\frac{10^6}{T}\right) \qquad (A2.3)$$

where $T$ is the temperature in Kelvin. $H_2^{18}O$ has lower vapor pressure and diffuses slower than $H_2^{16}O$ (Farquhar and Lloyd, 1993). The kinetic isotope effect due to diffusion $\epsilon_k$, is the weighted sum of the fractionations of water isotopologues during diffusion through the stomata in the air ($\varepsilon_{ks}$) and through the boundary layer ($\varepsilon_{kb}$) (Farquhar and Lloyd, 1993). According to (Merlivat, 1978;Barkan and Luz, 2007), the fractionation factor for $H_2^{18}O$ as it diffuses through stomata is 28‰ ($\varepsilon^{18}_{ks}$). According to (Farquhar

and Lloyd, 1993) $\varepsilon_{kb} = (\varepsilon_{ks})^{\frac{2}{3}}$, i.e, the fractionation factor as $H_2^{18}O$ diffuses through the boundary layer is 19‰ ($\varepsilon^{18}_{kb}$). The fractionation factors for $H_2^{17}O$ for diffusion through stomata and boundary layer are 14.6‰ and 9.7‰, respectively (Barkan and Luz, 2007). The kinetic fractionation of water vapor as it diffuses through stomata and boundary layer is given by equation A2.4 (Farquhar and Lloyd, 1993)

$$\varepsilon^{18}_k = \frac{28 \times g_b + 19 \times g_s}{g_b + g_s} \qquad (A2.4)$$


where $g_b$ and $g_s$ are boundary layer conductance and stomatal conductance respectively. $\delta^{17}O_{wes}$ can be calculated using a similar equation as $\delta^{18}O_{wes}$ if $\delta^{17}O_{wa}$ and $\delta^{17}O_{we}$ are known, for this study we calculated $\delta^{17}O_{wes}$ assuming the irrigation water is the same with soil water.

$$\delta^{17}O_{wes} = \left(\frac{\delta^{18}O_{wes} + 1}{\delta^{18}O_{IRW} + 1}\right)^{\lambda_{trans}} \times (\delta^{17}O_{IRW} + 1) - 1 \qquad (A2.5)$$

**Appendix 3**

**Mole fraction of CO₂ at the CO₂-H₂O exchange site**

$\delta^{18}O_i$ is $\delta^{18}O$ of $CO_2$ in the intercellular airspace, calculated as (Farquhar and Cernusak, 2012):






$$\delta^{18}O_i = \frac{\delta^{18}O_{io} + t^{18} \times \left(\delta^{18}O_A \times \left(\frac{c_a}{c_i} + 1\right) - \delta^{18}O_a \times \frac{c_a}{c_i}\right)}{1 + t^{18}} \tag{A3.1}$$

where the ternary correction factor $t^{18}$ is calculated as:

$$t^{18} = \frac{(1 + a_{18bs}) \times E}{2g_{ac}} \tag{A3.2}$$

$g_{ac}$ is the conductance as $CO_2$ diffuses through the boundary layer and stomata, $a_{18bs}$ is the weighted $^{18}O$
fractionation for $CO_2$ diffusion across the boundary layer and stomata in series.

$$a_{18bs} = \frac{(c_a - c_s) \times a_{18b} + (c_s - c_i) \times a_{18s}}{c_a - c_i} \tag{A3.3}$$

The $\delta^{18}O_{io}$ is the $\delta^{18}O$ of $CO_2$ in the intercellular air spaces ignoring ternary correction and it is given by (Farquhar and Cernusak, 2012).

$$\delta^{18}O_{io} = \delta^{18}O_A \times \left(1 - \frac{c_a}{c_i}\right) \times (1 + a_{18bs}) + \frac{c_a}{c_i} \times \left(\delta^{18}O_a - a_{18bs}\right) + a_{18bs} \tag{A3.4}$$


where $a_{18w}$ is the $^{18}O$ fractionation of $CO_2$ for dissolution and diffusion in water (0.8‰) and $a_{18s}$ and
$a_{18b}$ are the $^{18}O$ fractionation of $CO_2$ as it diffuses through stomata (8.8‰) and the boundary layer (5.8 ‰), respectively (Farquhar et al., 1982;Farquhar and Lloyd, 1993). The oxygen isotope composition of the assimilated $CO_2$ is calculated from a mass balance using the mole fraction and isotope composition
of $CO_2$ entering and leaving the cuvette as:

$$\delta^{18}O_A = \frac{\delta^{18}O_a - \Delta_A{}^{18}O}{\Delta_A{}^{18}O + 1} \tag{A3.5}$$

Similar to the derivation shown in the main paper for $\delta^{18}O$, the mole fraction of $CO_2$ at the $CO_2$-$H_2O$ exchange site can be calculated from $\delta^{17}O$ as:

$$c_m = c_i \left(\frac{\delta^{17}O_i - a_{17w} - \delta^{17}O_A \times (1 + a_{17w})}{\delta^{17}O_m - a_{17w} - \delta^{17}O_A \times (1 + a_{17w})}\right) \tag{A3.6}$$

where $\delta^{17}O_{iCO2}$ is $\delta^{17}O$ of $CO_2$ in the intercellular airspace including ternary correction ($t^{17}$), calculated
as:

$$\delta^{17}O_i = \frac{\delta^{17}O_{io} + t^{17} \times \left(\delta^{17}O_A \times \left(\frac{c_a}{c_i} + 1\right) - \delta^{17}O_o \times \frac{c_a}{c_i}\right)}{1 + t^{17}} \tag{A3.7}$$





$$t^{17} = \frac{(1 + a_{17bs}) \times E}{2g_{ac}}$$ (A3.8)

Here $a_{17bs}$ is the weighted discrimination of $C^{16}O^{17}O$ diffusion across the boundary layer and stomata in
series respectively and is given by:

$$a_{17bs} = \frac{(c_a - c_s) \times a_{17b} + (c_s - c_i) \times a_{17s}}{c_a - c_i}$$ (A3.9)

$\delta^{17}O_{io}$ is the $\delta^{17}O$ value of the $CO_2$ in the intercellular air spaces ignoring ternary correction and it is
calculated as:

$$\delta^{17}O_{io} = \delta^{17}O_A \times \left(1 - \frac{c_a}{c_i}\right) \times (1 + \bar{a}_{17}) + \frac{c_a}{c_i} \times \left(\delta^{17}O_a - \bar{a}_{17}\right) + \bar{a}_{17}$$ (A3.10)


where $\bar{a}_{17}$ is the $^{17}O$ fractionation of $CO_2$ during diffusion across boundary layer, stomata, cell wall and
plasma membrane in series, similar to $\delta^{18}O$.

$$\bar{a}_{17} = \frac{(c_a - c_s) \times a_{17b} + (c_s - c_i) \times a_{17s} + (c_i - c_m) \times a_{17w}}{c_a - c_m}$$ (A3.11)

The $^{18}O$ fractionation ($\alpha^{18}$-1) for dissolution is -0.8‰ (Vogel.J.C. et al., 1970). The corresponding $^{17}O$
fractionation is -0.418‰, calculated from the $^{18}O$ fractionation due to equilibrium dissolution using $\lambda_{CO2\text{-}H2O}$ is 0.5229 (Barkan and Luz, 2012). Assuming the $^{17}O$ fractionation during diffusion in water is the
same as the fractionation in the $^{13}CO_2$ (Farquhar and Lloyd, 1993) and using the average fractionation
determined for $^{13}CO_2$ is 0.8‰ (0.7‰ (O'Leary, 1984) and 0.9‰ (Jähne et al., 1987). The $^{17}O$ fractionation
due to the sum of the equilibrium dissolution and diffusion in water is then $a_{17w} = 0.382‰$. Similar to
(Farquhar and Lloyd, 1993), using the principle of binary diffusivities (Mason and Marrero, 1970) , $a_{17s}$
and $a_{17b}$ are 4.4‰ and 2.9‰ using the power of 2/3 relationship between the boundary layer and stomatal
conductance fractionation ($\alpha_b = \alpha_s^{2/3}$) obtained by (Farquhar and Lloyd, 1993).

For calculating the isotopic composition at the site of oxygen isotope exchange, we assume that the
isotopic composition of $CO_2$ is fully equilibrated with water at the evaporation site. This includes the
implicit assumption that the isotopic composition of the leaf water at the $CO_2$-$H_2O$ exchange site is the
same as at the site of evaporation. The $\delta^{17}O$ of $CO_2$ at the $CO_2$-$H_2O$ exchange site ($\delta^{17}O_m$) is then
calculated using the triple oxygen isotope ratio relationship, $\alpha^{17} = (\alpha^{18})^{\lambda_{CO_2-H_2O}}$.

$$\delta^{17}O_m = \left(\frac{\delta^{18}O_m + 1}{\delta^{18}O_{wes} + 1}\right)^{\lambda_{CO_2-H_2O}} \times (\delta^{17}O_{wes} + 1) - 1$$ (A3.12)






where $\lambda_{CO2\text{-}H2O}$ is 0.5229 (Barkan and Luz, 2012). Similar to $\delta^{18}O$, the $\delta^{17}O$ value of the assimilated $CO_2$ is calculated from a mass balance using the mole fraction and isotope composition of $CO_2$ entering and leaving the cuvette as:

$$\delta^{17}O_A = \frac{\delta^{17}O_a - \Delta_A{}^{17}O}{\Delta_A{}^{17}O + 1} \tag{A3.13}$$



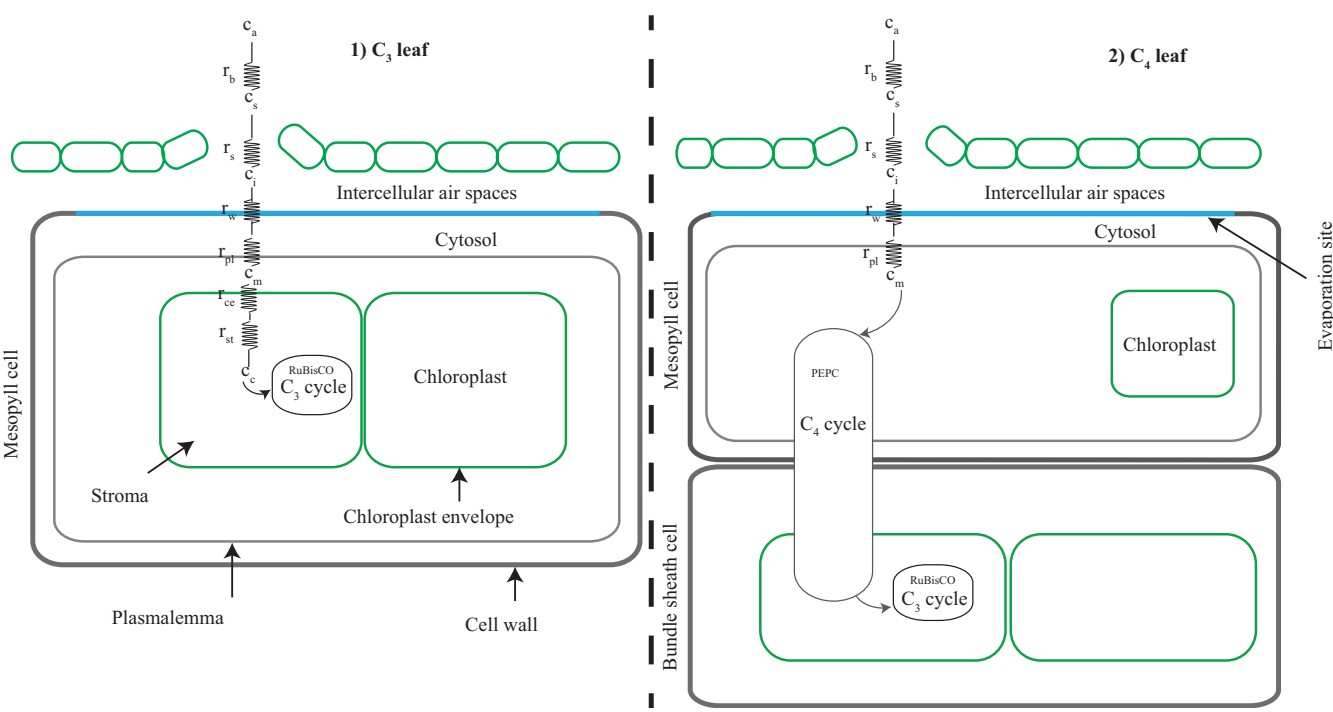

Figure 1 A simplistic 2-D schematic of $CO_2$ fluxes in a $C_3$ and $C_4$ leaf modified from (Cousins et al., 2020). During photosynthesis, ambient $CO_2$ ($c_a$) diffuses into the leaf intercellular spaces ($c_i$) through the
boundary layer ($r_b$) and stomata ($r_s$). $c_s$ is the mole fraction of $CO_2$ in the leaf surrounding. In a $C_3$ leaf (1) the resistances for $CO_2$ diffusion from the intercellular air space to RuBisCO in the chloroplast ($c_c$) is the wall ($r_{wall}$), the plasmalemma ($r_{pl}$), the chloroplast envelope ($r_{ce}$) and the stroma ($r_{st}$) resistance. In a $C_4$ leaf (2) the resistances for $CO_2$ diffusion to PEPC in the cytosol ($c_m$) is the wall ($r_{wall}$) and the plasmalemma ($r_{pl}$) resistances.




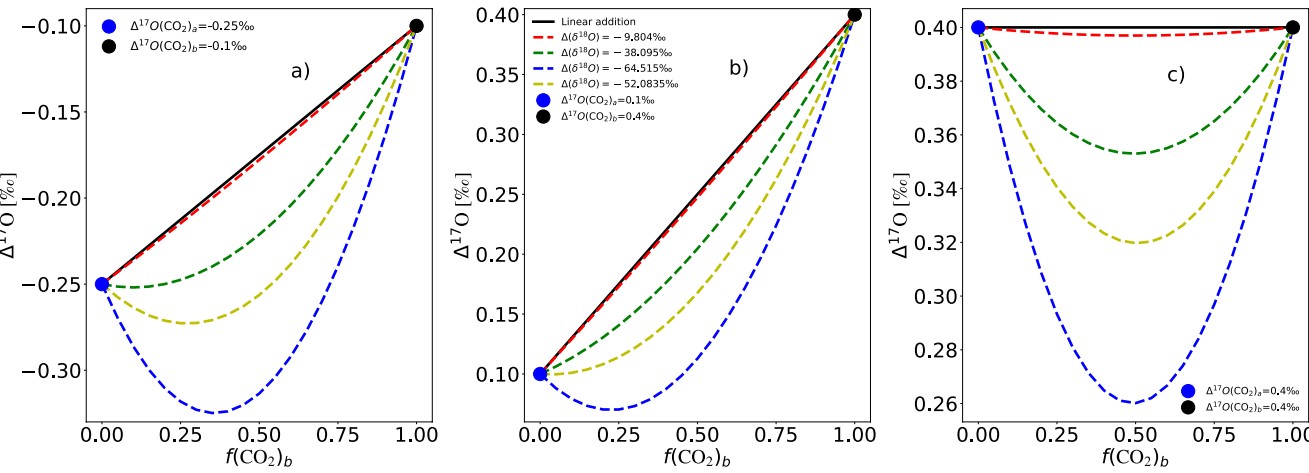

Figure 2 Illustration of the changes in $\Delta^{17}O$ for mixing of two different gases when the $\Delta^{17}O$ values are calculated in logarithmic form, as a function of the fraction of $CO_2$ gas b. The blue and black circles show the $\Delta^{17}O$ values of the mixing end members and the different colors show mixing lines for differences in $\delta^{18}O$.






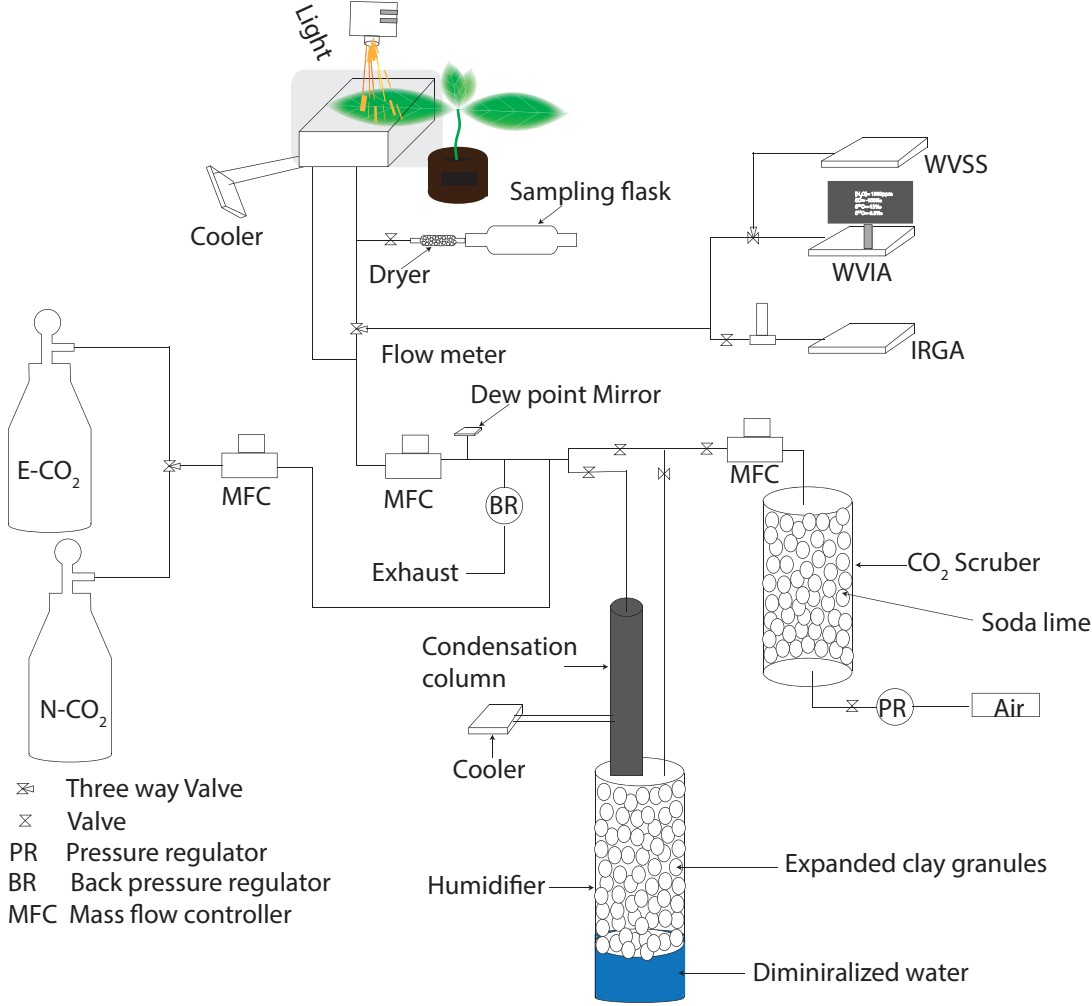


**Figure 3** Schematic diagram of the leaf cuvette experimental setup. IRGA stands for the infrared gas analyzer, WVSS is the water vapor standard source, WVIA is the water vapor isotope analyzer, N-CO$_2$ is normal CO$_2$, E-CO$_2$ is $^{17}$O-enriched CO$_2$.






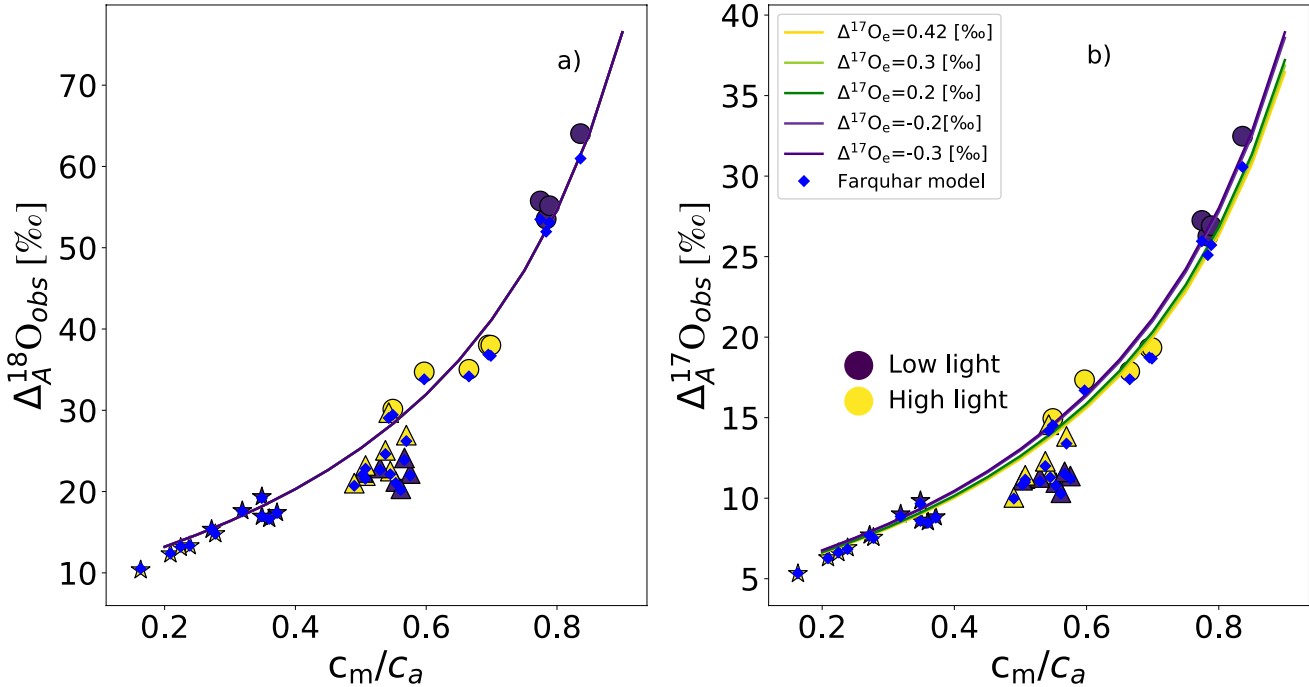

Figure 4 a) $\Delta_A{}^{18}O_{obs}$ and b) $\Delta_A{}^{17}O_{obs}$ during photosynthesis for two C$_3$ plants, sunflower (circles) and ivy
(triangles) and C$_4$ plant maize (stars) as a function of $c_m/c_a$. The solid lines show results from the leaf
cuvette model, where $\delta^{18}O$ of the CO$_2$ entering the cuvette is 30.47‰ while the $\delta^{17}O_e$ of the CO$_2$ is varied
based on the $\Delta^{17}O_e$ of the CO$_2$. The blue diamond dots are results from Farquhar model (Equation S12 in
the supplementary material for $\Delta_A{}^{17}O_{FM}$ and Equation 7 for $\Delta_A{}^{18}O_{FM}$).


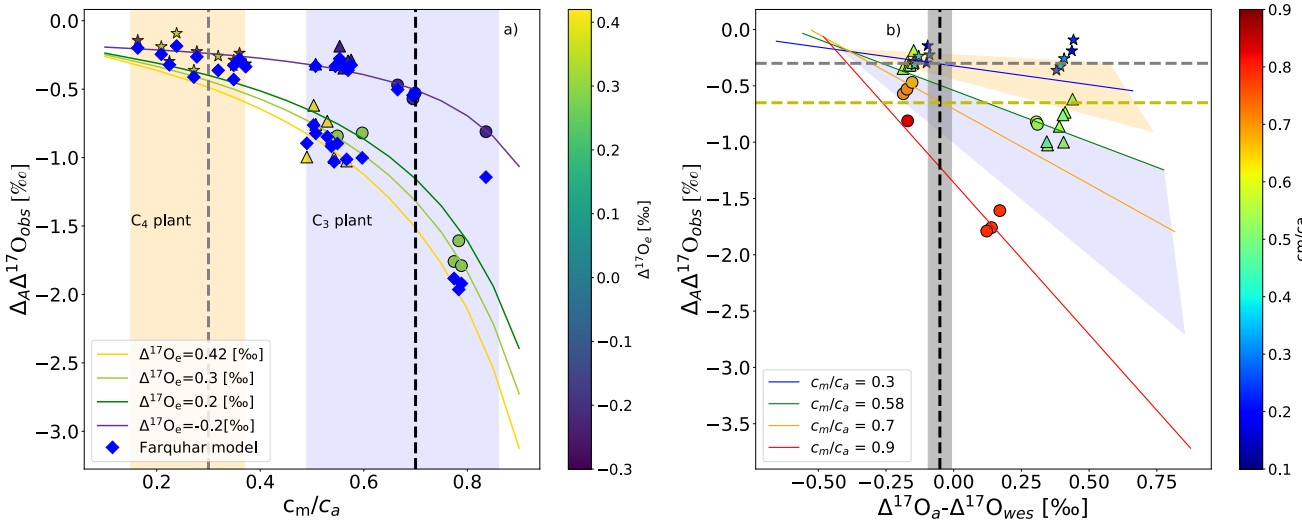

**Figure 5** a) $\Delta_A\Delta^{17}O$ of $CO_2$ as a function of $c_m/c_a$ for isotopically different $CO_2$ gases entering the cuvette
(color bar shows $\Delta^{17}O_e$) for sunflower (circles), ivy (triangles) and maize (stars). $\Delta_A\Delta^{17}O$ values calculated
using the leaf cuvette model are shown as solid lines in corresponding colors ($\Delta^{17}O_e$ values given in the
legend). The shaded areas indicate the $c_m/c_a$ ranges for $C_4$ and $C_3$ plants and the vertical dashed lines
indicate the mean $c_m/c_a$ ratio used for extrapolating from the leaf scale to the global scale. The blue
symbols represent results from the Farquhar model, calculated as $\Delta_A\Delta^{17}O = \ln(\Delta_A^{17}O_{FM}+1) - 0.528\times\ln$
$(\Delta_A^{18}O_{FM}+1)$. b) dependency of $\Delta_A\Delta^{17}O$ on the difference between the $\Delta^{17}O$ of $CO_2$ entering the cuvette
and the $\Delta^{17}O$ of leaf water at the evaporation site color coded for different $c_m/c_a$ ratios. The solid lines are
results of the leaf cuvette model for different $c_m/c_a$ ratios stated in the legend. The dashed vertical black
line indicates the difference between the global average $\Delta^{17}O$ value for $CO_2$ (-0.168‰) and leaf water (-
0.067‰) (Koren et al., 2019). The gray and yellow horizontal dashed lines indicate $\Delta_A\Delta^{17}O$ of $C_4$ and $C_3$
plants for $c_m/c_a$ ratio of 0.3 and 0.7, respectively globally.





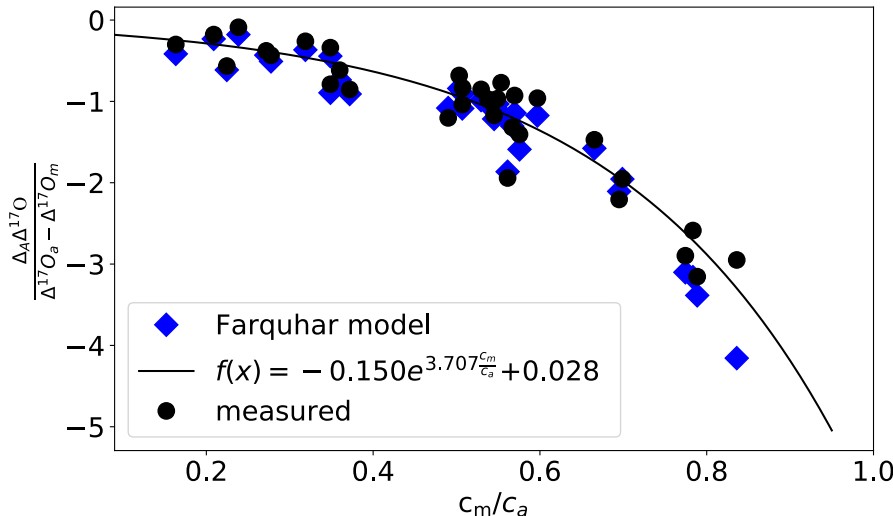


Figure 6 Dependency of $\Delta_A\Delta^{17}O$ on the relative difference on the $\Delta^{17}O$ $CO_2$ entering the leaf and the $\Delta^{17}O$ of $CO_2$ in equilibrium with leaf water against $c_m/c_a$ ratio.





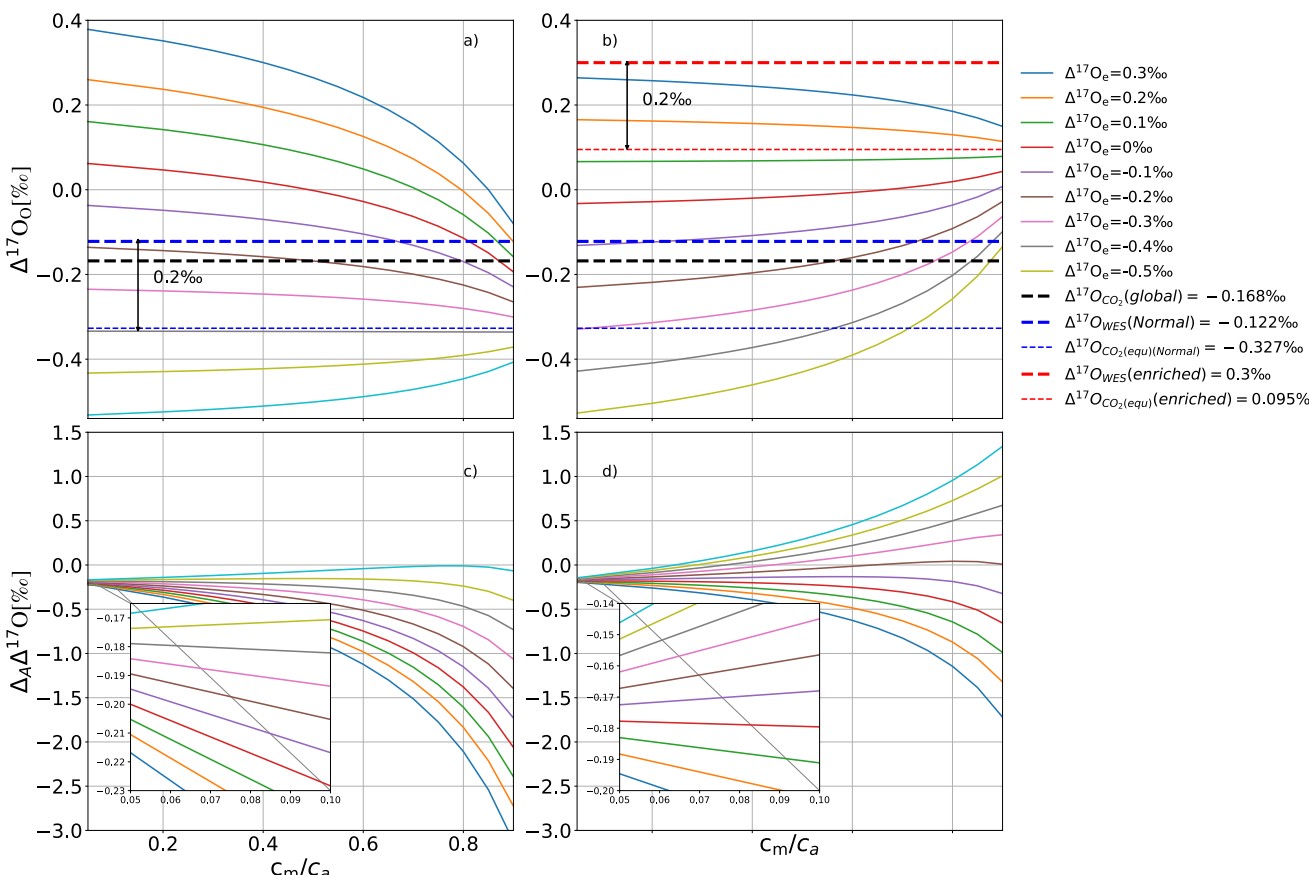

**Figure 7** a) and b) $\Delta^{17}O_a$ as a function of $c_m/c_a$ for various values of $\Delta^{17}O_e$ (see legend) for $\Delta^{17}O_{wes}$ = -0.122‰ in a) and $\Delta^{17}O_{wes}$ = 0.300‰ in b). c) and d) show the corresponding values for $\Delta_A\Delta^{17}O$. $\Delta^{17}O_{global}$ is the global average $\Delta^{17}O$ value for atmospheric $CO_2$ (Koren et al., 2019). When $\Delta^{17}O$ of $CO_2$ entering the cuvette is approximately 0.2‰ lower than the $\Delta^{17}O$ of leaf water at the $CO_2$-$H_2O$ exchange site, $\Delta^{17}O$ of the $CO_2$ leaving the cuvette does not change when $c_m/c_a$ vary.





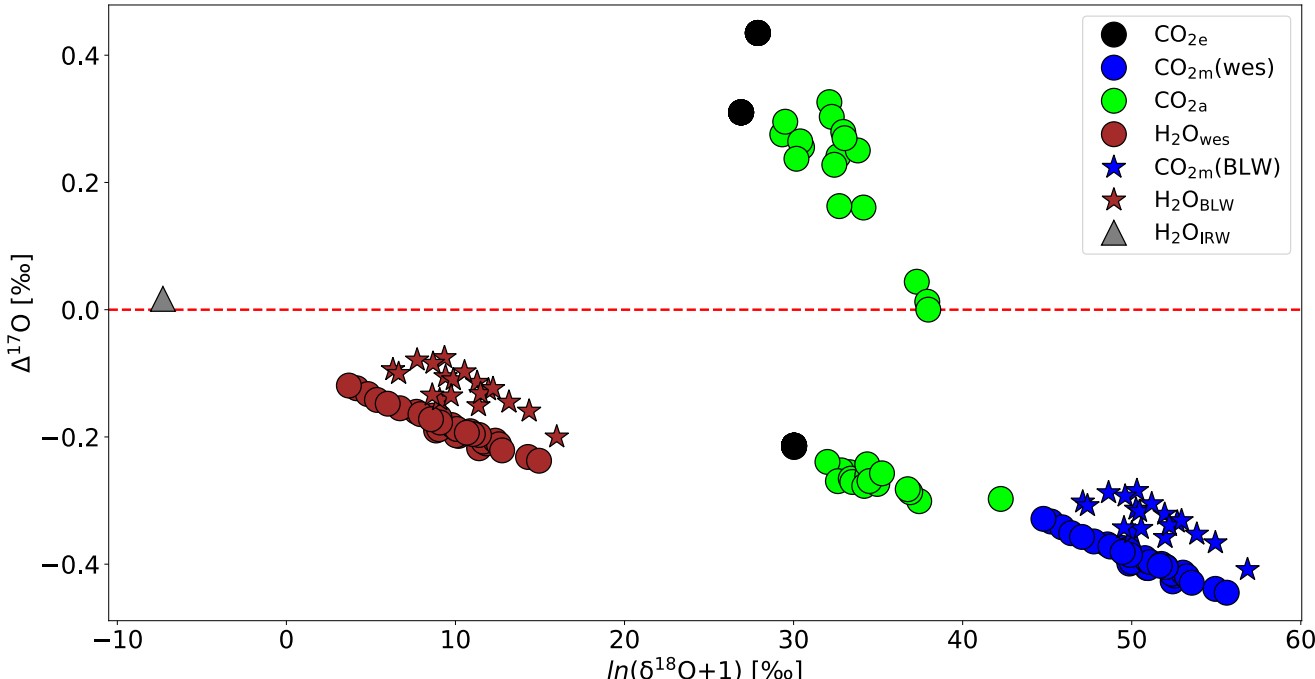

Figure 8 Isotopic composition of various relevant oxygen reservoirs that affect the $\Delta^{17}O$ of atmospheric
$CO_2$ during photosynthesis: irrigation water (grey triangle), calculated leaf water at the evaporation site
(brown circles), measured bulk leaf water (brown star), $CO_2$ entering the cuvette (black circles), $CO_2$
leaving the leaf cuvette (green circles), $CO_2$ equilibrated with leaf water at the evaporation site (blue
circles), $CO_2$ equilibrated with bulk leaf water (blue stars).

Table 1: Summary for the parameters used of the extrapolation of leaf scale experiments to the global
scale and the results obtained, and $\Delta^{17}O$ value of tropospheric $CO_2$ available measurements.

| Parameters and values used for global estimation | | |
|---|---|---|
| Parameter | Value | ref |
| GPP | 120 PgCyr$^{-1}$ | (Beer et al., 2010) |
| $f_{C4}$ | 23% | (Still et al., 2003) |
| $f_{C3}$ | 77% | (Still et al., 2003) |
| $c_m/c_a$ (C$_3$) | 0.7 | (Hoag et al., 2005) |
| $c_m/c_a$ (C$_4$) | 0.3 | (Hoag et al., 2005) |
| $\Delta^{17}O$ leaf water (global mean, modelled) | -0.067±0.04‰ | (Koren et al., 2019) |
| $\Delta^{17}O$ CO$_2$ (global mean, modelled) | -0.168±0.013‰ | (Koren et al., 2019) |
| $\Delta_A\Delta^{17}O$ (global mean for C$_4$) | -0.3±0.18‰ | (Figure 5b, for $c_m/c_a$ ratio of 0.3) |
| $\Delta_A\Delta^{17}O$ (global mean for C$_3$) | -0.65±0.18‰ | (Figure 5b, for $c_m/c_a$ ratio of 0.7) |
| $\Delta_A\Delta^{17}O$ (global mean for whole vegetation) | -0.57±0.14‰ | (Equation 13) |





| $\Delta_A\Delta^{17}O$-isoflux (global mean for $C_4$) | $-7.3\pm4‰PgCyr^{-1}$ | (Equation 14, only for $C_4$) |
|---|---|---|
| $\Delta_A\Delta^{17}O$-isoflux (global mean for $C_3$) | $-53\pm15‰PgCyr^{-1}$ | (Equation 14, only for $C_3$) |
| $\Delta_A\Delta^{17}O$-isoflux (global mean for whole vegetation) | $-60\pm15‰PgCyr^{-1}$ | (equation 14) |
| $\Delta_A\Delta^{17}O$-isoflux (global mean for whole vegetation) | $-47‰PgCyr^{-1}$ | (Hoag et al., 2005) |
| $\Delta_A\Delta^{17}O$-isoflux (global mean for whole vegetation) | $-42$ to $-92‰PgCyr^{-1}$ | (Hofmann et al., 2017) |
| **$\Delta^{17}O$ value of tropospheric $CO_2$** | | |
| $\Delta^{17}O(CO_2)$ for $CO_2$ samples collected in La Jolla-UCSD (California, USA) (1990 to 2000) | $-0.173\pm0.046‰$ | (Thiemens et al., 2014) |
| $\Delta^{17}O(CO_2)$ for $CO_2$ samples collected in Israel | $0.034\pm0.010‰$ | (Barkan and Luz, 2012) |
| $\Delta^{17}O(CO_2)$ for $CO_2$ samples collected in South china sea (2013-2014) | $-0.159\pm0.084‰$ | (Liang et al., 2017a;Liang et al., 2017b) |
| $\Delta^{17}O(CO_2)$ for $CO_2$ samples collected in Taiwan (2012-2015) | $-0.150\pm0.080‰$ | (Liang et al., 2017a;Liang et al., 2017b) |
| $\Delta^{17}O(CO_2)$ for $CO_2$ samples collected in California (USA) (2015) | $-0.177\pm0.029‰$ | (Liang et al., 2017a;Liang et al., 2017b) |
| $\Delta^{17}O(CO_2)$ for $CO_2$ samples collected in Göttingen (Germany) (2010-2012) | $-0.122\pm0.065‰$ | (Hofmann et al., 2017) |
