# Peer review of "Leaf-scale quantification of the effect of photosynthetic gas exchange on $\Delta^{17}O$ of atmospheric CO2"

_Biogeosciences, 2020_

## Referee Comment (RC1) · Anonymous Referee #1 · 25 Apr 2020

As far as I understand the justification for the great effort required in measuring 17O and its "access" (or anomaly), is the discovery of significant mass independent oxygen isotope effects in the stratosphere that is conserved to some extent in the troposphere (seems to be true both for atmospheric O2 and CO2). The extent to which this anomaly is conserved in the troposphere depends on the CO2 (or O2) cycling through the biosphere, which erases it by exchange with water. Thus, if the stratospheric production of the anomaly is known and it is relatively constant, the residual signal in the troposphere should reflect the biosphere productivity (GPP). This is exiting application considering the uncertainty around GPP.

[Figure]

However, ALL the processes associated with the Biosphere, including leaf gas exchange studied here, seems to be mass dependent and are FULLY covered by the conventional 18O studies. The only exception may be the small variations observed in the lambda factor that define the expected ratio of 18O to 17O mass dependent discrimination ($\sim$0.5), which is not studied here. And so, while the present paper goes through an impressive exercise of gas exchange and isotopic measurements and calculations, I fail to see the purpose and merit of this exercise, beyond a test that verifies that indeed the 17O measurements are consistent with the 18O studies. The occlusions as much as I can see are already fairly well-known form 18O studies and, in fact, much of the calculations here still depends on the 18O measurements.

For example, the key results indicated in the Abstract are: "Our results demonstrate that two key factors determine the effect of gas exchange on the D17O of atmospheric CO2. The relative difference between D17O of the CO2 entering the leaf and the CO2 in equilibrium with leaf water, and the back-diffusion flux of CO2 from the leaf to the atmosphere, which can be quantified by the Cm/Ca ratio". Isn't it that these 'basic principles' of leaf gas exchange are already fairly well known from previous CO2 and the 18O studies?

It seems also that the notion of "discrimination against D17O of atmospheric CO2" is not clear. If this is confused with D in leaf photosynthesis as for D18, then again 17O is predictable and has no clear additional information (other than perhaps the reflection of the possible variations in the lambda factor). The final estimate of global 17O discrimination anomaly is back of the envelope calculation based on these known principles and literature values. I am not sure what new insights are provided.

And so, while the experimental setup, measurements, and going through the isotopic theory are impressive and seems to be well done on first look, I think the authors have to re-think the presentation and provide a better justification of what in these measurements takes advantage of any mass independent effects (as declared), and in what ways this goes beyond a sophisticated confirmatory report.

---

## Referee Comment (RC2) · Anonymous Referee #2 · 26 Apr 2020

The triple oxygen isotopic composition of CO2 ( $\Delta^{17}O_{CO2}$ ) had been regarded as spatiotemporally constant in the troposphere because of its short residence time (e.g., Luz et al., 2000). Recently, significant seasonal and temporal variations of  $\Delta^{17}O_{CO2}$  were first revealed in the atmosphere near the surface by Hofmann et al. (2017) and Liang et al. (2017), respectively, both of which were mainly controlled by the interaction of CO2 between the atmosphere and biosphere. These studies were then followed by the threedimensional simulation study with an atmospheric physico-chemical model (Koren et al., 2019), to quantify the global CO2 budget. The next step, therefore, must be the process study involving oxygen isotope fractionations in association with individual CO2 fluxes.

This study by Adnew, Pons, Koren, Peters, Röckmann, aims to quantify the  $\Delta^{17}O_{CO2}$  change during photosynthetic CO2 removal from the atmosphere, caused by tiny difference of 17O-18O relationship between kinetic and equilibrium isotope fractionations inside the leaf.

To my knowledge, this is the first experimental study for  $\Delta^{17}O_{CO2}$  at the leaf-scale; thus, their results provided must be important. However, I am frustrated and feel difficult to plough through the manuscript because 1) the structure of the manuscript (context) seems scattered, 2) experimental results (raw data) were not shown although values in all graphs were processed, 3) there appears a lot of faults in equations or figure number in the main text, and 4) it's a mixture of lengthy and in-short explanations. I strongly recommend the authors to revise the manuscript more simply and concisely.

**General comments**

**It spent 11 of 18 pages (until conclusion) from the Introduction to "Materials and methods (M&M)." It seems too dominant; in other words, Results and Discussion seem too short. There appears a lengthy description in M&M, and the description for experimental results is too short.**

**L84-90: This block appears the center of your motivation; however, there is no specific description of what the problem or limitation exists currently. Until this block (and perhaps in previous studies), you mentioned the  $\Delta^{17}$ O is free from any terrestrial MDF processes and made readers believe that  $\Delta^{17}$ O be a more robust tracer for estimating GPP. You must describe what actual problems lying among previous studies such as inconsistency, uncertainty, speculation, assumption and so on. Without this explanation, readers could not have motivations to read the next pages. I strongly recommend adding descriptions for the different slopes of three-isotope plots due to the different MDF**

processes.

**I strongly recommend the authors to revise the Theory part completely. The structure is scattered and forces readers to jump frequently between the main text, Appendix and Supplementary Materials (SM). Appendix should be moved to SM.**

**The term "fractionation" should be replaced to "isotope fractionation" for all.**

**My major concern is the relation between dots of "Farquhar model" and curves in Figs 4 and 5a) and related description in Section 3.6. If I were not misunderstanding, both are results calculated from the "Farquhar model." Dots were obtained by giving several observed results and curves were simulated by giving similar boundary conditions to the experimental setting. Is the former necessary? This is very confusing.**

**I strongly recommend the authors to provide "List of symbols." for all parameters used and defined.**

**The parameter  $c_m$  seems one of the most important numbers in this study. For obtaining this, only  $\delta^{18}O$  and  $\alpha_{18}$  values were used concerning isotope ratio, though. Is it possible to use  $\Delta^{17}O$  and  $\lambda$  values to evaluate  $c_m$  instead? At least does it make sense to test its feasibility?**

**As shown in Figure 5, the discrimination of  $\Delta^{17}$ O of CO2 during photosynthesis varies widely, and controlled by the magnitude of oxygen isotope equilibration at the CO2-H2O site, that is to say, the relative contribution of kinetic (diffusion) and equilibrium isotope fractionation. This conclusion is almost identical to the knowledge using conventional  $\delta^{18}$ O results. Moreover, In the last paragraph of Discussion, authors mentioned that the main uncertainty is cm/ca ratio, which may be same as the main uncertainty of  $\delta^{18}$ O. My impression after reading this manuscript is that the intra-MDF variation dominate that of MIF signature on tropospheric CO2, which weakens the merit to study  $\Delta^{17}$ O of CO2. What is an advantage to use  $\Delta^{17}$ O instead of  $\delta^{18}$ O? Please provide suggestions or implications to general biogeochemists.**

**Specific comments**

L41: "replaced using..." What this means? Be more specific.

- L47: "see equation (1)" instead of "see below"
- L51: "the latter term" I guess it should be "the former term," which means photosynthetic CO2 uptake.
- L53: "variable  $\delta^{18}$ O gradient" I think "significant  $\delta^{18}$ O variation" is more appropriate.
- L56: Delete "the isotopically exchanged"
- L45-57: In this block, you should use the term "isotope fractionation" with its definition for the subsequent block. More desirably, the term "mass-dependent isotope

fractionation (MDF)" with its definition.

- L63: "mass-dependent fractionation" should be "mass-dependent isotope fractionation" with its definition in detail.
- L62-64: Need revision because the latter paragraph is just a refrain of the former.
- L65: Describe a specific value instead using "considerable"
- L60-71: In this block, you should use the term "mass-independent isotope fractionation (MIF)" with its definition, and associate it with "photochemical isotope exchange"
- L70-71: This is not sufficient because exchanges with soil and ocean water are also nonenzymatic processes.
- L78: "The  $\Delta^{17}$ O of CO2" instead of "The 17O-excess of CO2 ( $\Delta^{17}$ O) (equation 4)"
- L80: Clarify "well-known three-isotope slope." "Non three-isotope person" cannot understand what this means.
- L92-106 and Figure 1: The explanation is this block is too general, should reduce to a few sentences. Detail description may be required if you would like to discuss the difference of results due to the different types in the Discussion. As for Figure 1, not this scheme but simpler scheme in Figure S6 was actually used in this study. Therefore, it seems more appropriate to delete Figure 1 and insert S6 here.

L108-109: What is "leaf level"?

L116-117: " $\Delta^{17}$ O" instead of "triple oxygen isotopic composition"

Equations 1 and 2: Should be merged such as,

 $\delta^{n}O = {}^{n}R_{sample}/{}^{n}R_{VSMOW} - 1$ , n refers 17 or 18

or simpler,

 $\delta = R_{\text{sample}}/R_{\text{VSMOW}} - 1.$

- L134: I recommend "The MDF factor" instead of "The factor"
- L135-137: Delete "This relation..., respectively.
- L137: "variations" instead of "values." "Small delta value" is meaningless.
- L139-140: I recommend "Note that  $\Delta^{17}$ O changes not only by MIF processes, but also MDF processes with a different  $\lambda$  value from the definition,"
- L145-146: "which was obtained by the observation of" instead of "the value associated with"
- L147-148: Delete "Note that ...  $\delta^{18}$ O."
- L150-258 (Section 2.2-2.4): Revise completely.
- Equation 5: Use n (18 or 17) or simpler expression as above, then revise or delete L158 and L163.
- Equation 12: Move after equation 5 with related sentences.
- L163-168: Delete "We note that...itself."

- L170-200 and Section 2.4: Integrate and locate in new section such like "Extension of Farquhar-Lloyd model to oxygen triple isotopes. Eqs. 6 and 11 are almost identical so that they should be merged.
- Equation 15: Use n (18 or 17) or simpler expression, then revise or delete L256-257 and related sentences in Appendix A3. No definition of ci.

L208-213 and Figure 2: Move to SM.

- Section 2.3: I recommend moving this section to the Discussion.
- L217: Delete "which is a net sink,"
- L230: Specify which model is used.
- L241-259: Here detail but still insufficient description was made only for  $\delta_m$ , on the other hand, no description for  $c_i$  and  $\delta_i$  which were driven away to Appendix. This seems out of balance and forces readers to jump here and there. I recommend moving this block to SM.

L262-265: Could it be shorter?

- L268-269: "The 4th or higher..." Is this sentence an explanation for maize or all species?
- Section 3.2: Need the model and the manufacturer for halogen lamp, neutral filters, dewpoint meter (the model).
- Section 3.3: Could this section be shorter to several sentences? The description for  $\delta D$  and obtaining optimum setting seem appropriate in SM.
- L349: Water was converted to O2
- Section 3.5: In previous section, unit of  $\Delta^{17}$ O is ‰. Here ppm is used. Use a uniform manner.
- Section 3.6: See related general comment
- L403: The last sentence is a refrain.
- Results: Show experimental results (raw data) such as c,  $\delta$ ,  $\Delta$ , w, for entering and leaving from the cuvette, etc. Show table of them and describe them.
- L414-415: Delete this sentence
- Section 4.2: Avoid using "17O-excess" in the title and L433 for uniformity
- L477-493: I could not understand this block. If the authors applied different lambda values to individual results, the vertical axis in Figure 8 would be meaningless, and one could not evaluate the graph and related description at all.
- Section 5.2: Avoid using "17O-excess" for uniformity
- Figure 3: Add individual flow direction.
- Figure 4: Panel b seems unnecessary. Delete and insert Figure 5a here.
- Figure 5: Move Panel a to Figure 4 as above
- Figure 6: Is it important to plot both of blue diamonds and curve. Should the curve be

improved by blue diamonds?

**Typographic errors**

Space inserted after semicolon (e.g., L33)

L42: Welp et al. (2011)

L45: The concept of the latter study..

L60: equation 4))

L207: Figure 2

L237: "Following (Farquhar....)" Need grammatical correctness

L267: Maize

L279, L297: Need grammatical correctness.

- Section 3.2: "Figure 3" instead of "Figure 2" (If Figure 2 were moved to SM, they are accidentally correct, though)
- References: I found typo. in Barbour et al. (2016) and Caemmerer and Farquhar (1981). There may be more. Confirm all.

L950: "entering and leaving" instead of "leaving and entering"

Equation A1.4: If the referred article (Caemmerer and Farquhar, 1981) was correct, the denominator must be  $(g_{ac}^{t} + E/2)$ .

---

## Author Comment (AC1) · 13 May 2020

Referee: 1

The answers to the questions/ comments and suggestions are stated below each comment, but please note the added supplement where the responses are given with proper formatting and detailed caption of figure 1.

1) As far as I understand the justification for the great effort required in measuring 17O and its "access" (or anomaly), is the discovery of significant mass independent oxygen isotope effects in the stratosphere that is conserved to some extent in the troposphere

(seems to be true both for atmospheric O2 and CO2). The extent to which this anomaly is conserved in the troposphere depends on the CO2 (or O2) cycling through the biosphere, which erases it by exchange with water. Thus, if the stratospheric production of the anomaly is known and it is relatively constant, the residual signal in the troposphere should reflect the biosphere productivity (GPP). This is exiting application considering the uncertainty around GPP.

This summary by the reviewer is correct. We would like to emphasize that if we reliably want to estimate GPP from $\Delta 17O$, we need to know the precise effect of photosynthesis and respiration on $\Delta 17O$, in the words of the referee, how does the $\Delta 17O$ signature actually look after being "erased" by exchange with the biosphere.

2) ALL the processes associated with the Biosphere, including leaf gas exchange studied here, seems to be mass dependent and are FULLY covered by the conventional 18O studies.

The referee is correct that in principle 18O indeed cycles through the same biological system, and undergoes the same (bio)physical processes. However, we would like to nuance the idea that $\delta 18O$ can help us FULLY understand ALL processes of interest. This is because conventional $\delta 18O$ studies have a number of distinct disadvantages. Notably, the $\delta 18O$ signature of all water pools in the system must be known to use $\delta 18O$ as a carbon cycle tracer. In addition, significant changes in $\delta 18O$ can occur due to processes that are not of primary interest in understanding GPP, e.g., leaf evaporation, or soil equilibration.

$\Delta 17O$ variation due to kinetic and equilibrium fractionation effects is much smaller and is better defined. This is because conventional bio-geo-chemical processes that modify $\delta 17O$ and $\delta 18O$ follow a well-recognized isotope fractionation slope. In earlier studies, many assumptions had to be made, because the effect on $\Delta 17O$ had never been quantified precisely. This is now accomplished through our study.

3) The only exception may be the small variations observed in the lambda factor that

define the expected ratio of 18O to 17O mass dependent discrimination (âĹij0.5), which is not studied here.

The reviewer correctly identifies that the small variations in $\lambda$ values can impact the $\Delta$17O we measure. These effects have been studied previously and the three isotope slopes have been established and are used in our study. We have included an additional figure (reproduced below) in the revised manuscript, which shows conceptually how the three-isotope slopes differ between the various processes and how they affect the observed $\Delta$17O signals. In addition, we also include experiments and model studies that involve artificially 17O labeled CO2 for the first time. We demonstrate how the resulting differences in $\Delta$17O between CO2 and leaf water affect the results, and that experiments with 17O labeled CO2 actually increase the signal to (measurement) noise ratio.

4) And so, while the present paper goes through an impressive exercise of gas exchange and isotopic measurements and calculations, I fail to see the purpose and merit of this exercise, beyond a test that verifies that indeed the 17O measurements are consistent with the 18O studies. The occlusions as much as I can see are already fairly well-known form 18O studies and, in fact, much of the calculations here still depends on the 18O measurements.

We appreciate that the referee acknowledges the considerable analytical effort that was made to produce our results. As mentioned above, we think that $\delta$18O measurements alone are not sufficient to study all aspects related to gas exchange between plants and the atmosphere and to quantify GPP. Thus, we posit that an alternative independent tracer is still very useful, and in fact, $\Delta$17O has been repeatedly suggested and already used as an independent and potentially even superior tracer. We nevertheless realize from the comment that the merit of our study was not communicated well, and we have considerably strengthened the motivation. The key point is that so far, the three-isotope slope of each of the processes that participate in plant-atmosphere gas exchange has been studied individually in an idealized experiment. The overall effect of all processes,

which work together in complex interaction, on $\Delta 17O$ has never been evaluated in a real plant exchange experiment. This is what is achieved in the research described in this manuscript and it is explicitly stated in the revised version.

Specifically, the results communicated in our manuscript

a) demonstrate that the established theory is applicable to $\Delta 17O(CO2)$ exchange at leaf-level.

b) experimentally quantify for the first time the effect of photosynthesis on $\Delta 17O$ of atmospheric CO2

c) quantify of the dependence of this effect on critical parameters

d) provide an independent bottom-up $\Delta 17O$-isoflux estimate based on these lab experiments.

Furthermore, we have now demonstrated that such studies are possible with IRMS methods, with considerable effort, but they may actually become more widely accessible thanks to novel laser instrumentation in the near future (McManus et al., 2005).

5) For example, the key results indicated in the Abstract are: "Our results demonstrate that two key factors determine the effect of gas exchange on the $\Delta 17O$ of atmospheric CO2. The relative difference between $\Delta 17O$ of the CO2 entering the leaf and the CO2 in equilibrium with leaf water, and the back-diffusion flux of CO2 from the leaf to the atmosphere, which can be quantified by the Cm/Ca ratio". Isn't it that these 'basic principles' of leaf gas exchange are already fairly well known from previous CO2 and the 18O studies?

We clearly acknowledge in our paper that the processes affecting $\delta 18O$ and $\Delta 17O$ are indeed the same, and in fact, we use the established conceptual models, with appropriate references. Nevertheless, this is the first experimental leaf-scale study where the applicability of these theoretical concepts to $\Delta 17O$ is actually demonstrated.

6) It seems also that the notion of "discrimination against $\Delta 17O$ of atmospheric $CO2$" is not clear. If this is confused with D in leaf photosynthesis as for D18, then again 17O is predictable and has no clear additional information (other than perhaps the reflection of the possible variations in the lambda factor). The final estimate of global 17O discrimination anomaly is back of the envelope calculation based on these known principles and literature values. I am not sure what new insights are provided.

We realize from this comment that we have not explained clearly enough the difference of measuring $\delta 17O$ and $\Delta 17O$. What the referee calls "$\Delta A$ in leaf photosynthesis as for $\Delta A18O$" would be $\Delta A17O$. This was also shown in our original paper for consistency but does indeed not provide additional information. Only the combination of $\Delta A18O$ and $\Delta A17O$ to $\Delta A\Delta 17O$ provides independent information. In the revised manuscript, we only present the results for $\Delta A18O$ and $\Delta A\Delta 17O$. Some of the confusion may have to do with the notation because the plant communities and atmospheric communities have used the symbol $\Delta$ for different quantities that are both used here. Our final estimate of GPP is not dependent on the individual ឧ17O and ឧ18O values, but only on $\Delta A\Delta 17O$. It is indeed a box model calculation, but to incorporate more variability, the entire mechanism would need to be incorporated into a global model. We are considering implementing this in the future, but for the box model presentations in this paper, we have used a global estimate of $\Delta 17O$ of $CO2$ and leaf water from a recent 3D global $\Delta 17O$ study (Koren et al., 2019).

7) And so, while the experimental setup, measurements, and going through the isotopic theory are impressive and seems to be well done on first look, I think the authors have to re-think the presentation and provide a better justification of what in these measurements takes advantage of any mass-independent effects (as declared), and in what ways this goes beyond a sophisticated confirmatory report. We realized already in the preparation of the manuscript that the presentation was difficult, and the referee comment confirms this. Nevertheless, we still think that the four conclusions identified above (copied below) make this a valuable study, whereas the referee sees only point

1 as significant merit.

a) demonstrate that the established theory is applicable to $\Delta 17O(CO2)$ exchange at leaf-level

b) experimentally quantify for the first time the effect of photosynthesis on $\Delta 17O$ of atmospheric CO2

c) study of the dependence of this effect on critical parameters

d) provide an independent bottom-up $\Delta 17O$-isoflux estimate based on these lab experiments.

References

Barkan, E., and Luz, B.: High precision measurements of 17O/16O and 18O/16O ratios in H2O, Rapid Commun. Mass. Sp., 19, 3737-3742, 10.1002/rcm.2250, 2005. Barkan, E., and Luz, B.: Diffusivity fractionations of H216O/H217O and H216O/H218O in air and their implications for isotope hydrology, Rapid Commun. Mass. Sp., 21, 6, 2007. Barkan, E., and Luz, B.: High-precision measurements of 17O/16O and 18O/16O ratios in CO2, Rapid Commun. Mass. Sp., 26, 2733-2738, 10.1002/rcm.6400, 2012. Koren, G., Schneider, L., Velde, I. R. v. d., Schaik, E. v., Gromov, S. S., Adnew, G. A., D.J.Mrozek, Hofmann, M. E. D., Liang, M.-C., Mahata, S., Bergamaschi, P., Laan-Luijkx, I. T. v. d., Krol, M. C., Röckmann, T., and Peters, W.: Global 3‐D Simulations of the Triple Oxygen Isotope Signature $\Delta 17O$ in Atmospheric CO2 J. Geophys. Res-Atmos. , 124, 28, 2019. Landais, A., Barkan, E., Yakir, D., and Luz, B.: The triple isotopic composition of oxygen in leaf water, Geochim. Cosmochim. Ac., 70, 4105-4115, 10.1016/j.gca.2006.06.1545, 2006. McManus, J. B., Nelson, D. D., Shorter, J. H., Jimenez, R., Herndon, S., Saleska, S., and Zahniser, M.: A high precision pulsed quantum cascade laser spectrometer for measurements of stable isotopes of carbon dioxide, J. Mod. Optic., 52, 12, 2005. Young, E. D., Galy, A., and Nagahara, H.: Kinetic and equilibrium mass-dependent isotope fractionation

laws in nature and their geochemical and cosmochemical significance, Geochim. Cosmochim. Ac., 66, 9, 2002.

Please also note the supplement to this comment:
https://www.biogeosciences-discuss.net/bg-2020-91/bg-2020-91-AC1-supplement.pdf

[Figure]

[Figure]

**Fig. 1.** Schematic representation of the processes that affect the $\Delta 17O$ of CO2 and H2O during photosynthetic gas exchange (not to scale)

**Supplement:**

From: Getachew Agmuas Adnew and co-authors

Referee: 1

The answers to the questions/ comments and suggestions are stated below each comment.

As far as I understand the justification for the great effort required in measuring $^{17}O$ and its "access" (or anomaly), is the discovery of significant mass independent oxygen isotope effects in the stratosphere that is conserved to some extent in the troposphere (seems to be true both for atmospheric $O_2$ and $CO_2$). The extent to which this anomaly is conserved in the troposphere depends on the $CO_2$ (or $O_2$) cycling through the biosphere, which erases it by exchange with water. Thus, if the stratospheric production of the anomaly is known and it is relatively constant, the residual signal in the troposphere should reflect the biosphere productivity (GPP). This is exiting application considering the uncertainty around GPP.

This summary by the reviewer is correct. We would like to emphasize that if we reliably want to estimate GPP from $\Delta^{17}O$, we need to know the *precise* effect of photosynthesis and respiration on $\Delta^{17}O$, in the words of the referee, how does the $\Delta^{17}O$ signature actually look after being "erased" by exchange with the biosphere.

ALL the processes associated with the Biosphere, including leaf gas exchange studied here, seems to be mass dependent and are FULLY covered by the conventional $^{18}O$ studies.

The referee is correct that in principle $^{18}O$ indeed cycles through the same biological system, and undergoes the same (bio)physical processes. However, we would like to nuance the idea that $\delta^{18}O$ can help us FULLY understand ALL processes of interest. This is because conventional $\delta^{18}O$ studies have a number of distinct disadvantages. Notably, the $\delta^{18}O$ - signature of all water pools in the system must be known to use $\delta^{18}O$ as a carbon cycle tracer. In addition, significant changes in $\delta^{18}O$ can occur due to processes that are not of primary interest to understanding GPP, e.g., leaf evaporation, or soil equilibration.

$\Delta^{17}O$ variation due to kinetic and equilibrium fractionation effects is much smaller and is better defined. This is because conventional bio-geo-chemical processes that modify $\delta^{17}O$ and $\delta^{18}O$ follow a well-recognized isotope fractionation slope. In earlier studies many assumptions had to be made, because the effect on $\Delta^{17}O$ had never been quantified precisely. This is now accomplished through our study.

**The only exception may be the small variations observed in the lambda factor that define the expected ratio of $^{18}O$ to $^{17}O$ mass dependent discrimination (~0.5), which is not studied here**.

The reviewer correctly identifies that the small variations in $\lambda$ values can impact the $\Delta^{17}O$ we measure. These effects have been studied previously and the three isotope slopes have been established and are used in our study. We have included an additional figure (reproduced below) in the revised manuscript, which shows conceptually how the three-isotope slopes differ between the various processes and how they affect the observed $\Delta^{17}O$ signals.
In addition, we also include for the first-time experiments and model studies that involve artificially $^{17}O$ labeled $CO_2$. We demonstrate how the resulting differences in $\Delta^{17}O$ between

CO₂ and leaf water affect the results, and that experiments with $^{17}O$ labeled $CO_2$ actually increase the signal to (measurement) noise ratio.

[Figure]

Figure 1. Schematic representation of the processes that affect the $\Delta^{17}O$ of $CO_2$ and $H_2O$ during photosynthetic gas exchange (not to scale). The triple oxygen isotope slope for transpiration at h=75 % relative humidity is $\theta_{trans}$=0.522-0.008 ×h = 0.516 (Landais et al., 2006). The triple isotope slope of $CO_2$-$H_2O$ exchange is $\theta_{CO2-H2O}$ = 0.5229 (Barkan and Luz, 2012). The triple isotope slope for diffusion of $CO_2$ is $\theta_{CO2-diff}$ = 0.509 (Young et al., 2002) and for diffusion of water vapor $\theta_{H2O(v)-diff}$= 0.518 (Barkan and Luz, 2007). The three isotope slope for equilibration between liquid (l) and gaseous (v) water is $\theta_{H2O(v)-H2O(l)}$ = 0.529 (Barkan and Luz, 2005). $\varepsilon^{18}O$ is the enrichment or depletion in $^{18}O$ due to the corresponding isotope fractionation process.

And so, while the present paper goes through an impressive exercise of gas exchange and isotopic measurements and calculations, I fail to see the purpose and merit of this exercise, beyond a test that verifies that indeed the $^{17}O$ measurements are consistent with the $^{18}O$ studies. **The occlusions as much as I can see are already fairly well-known form $^{18}O$ studies and, in fact, much of the calculations here still depends on the $^{18}O$ measurements.**

We appreciate that the referee acknowledges the considerable analytical effort that was made to produce our results. As mentioned above, we think that $\delta^{18}O$ measurements alone are not sufficient to study all aspects related to gas exchange between plants and the atmosphere and to quantify GPP. Thus, we posit that an alternative independent tracer is still very useful, and in fact $\Delta^{17}O$ has been repeatedly suggested and already used as independent and potentially even superior tracer.
We nevertheless realize from the comment that the merit of our study was not communicated well, and we have considerably strengthened the motivation. The key point is that so far, the three-isotope slope of each of the processes that participate in plant-atmosphere gas exchange has been studied individually in an idealized experiment. The overall effect of all processes, which work together in complex interaction, on $\Delta^{17}O$ has never been evaluated in a real plant exchange experiment. This is what is achieved in the research described in out manuscript and it is explicitly stated in the revised version.

Specifically, the results communicated in our manuscript
a) demonstrate that the established theory is applicable to $\Delta^{17}O$-$CO_2$ exchange at leaf-level.

b) experimentally quantify for the first time the effect of photosynthesis on $\Delta^{17}O$ of atmospheric $CO_2$
c) quantify of the dependence of this effect on critical parameters
d) provide an independent bottom-up $\Delta^{17}O$-isoflux estimate based on these lab experiments.

Furthermore, we have now demonstrated that such studies are possible with IRMS methods, with considerable effort, but they may actually become more widely accessible thanks to novel laser instrumentation in the near future (McManus et al., 2005).

For example, the key results indicated in the Abstract are: "Our results demonstrate that two key factors determine the effect of gas exchange on the $\Delta17O$ of atmospheric CO2. The relative difference between $\Delta17O$ of the CO2 entering the leaf and the CO2 in equilibrium with leaf water, and the back-diffusion flux of CO2 from the leaf to the atmosphere, which can be quantified by the Cm/Ca ratio". Isn't it that these 'basic principles' of leaf gas exchange are already fairly well known from previous CO2 and the 18O studies?

We clearly acknowledge in our paper that the processes affecting $\delta^{18}O$ and $\Delta^{17}O$ are indeed the same, and in fact we use the established conceptual models, with appropriate references. Nevertheless, this is the first experimental leaf-scale study where the applicability of these theoretical concepts to $\Delta^{17}O$ is actually demonstrated.

It seems also that the notion of "discrimination against $\Delta17O$ of atmospheric CO2" is not clear. If this is confused with D in leaf photosynthesis as for D18, then again 17O is predictable and has no clear additional information (other than perhaps the reflection of the possible variations in the lambda factor). The final estimate of global 17O discrimination anomaly is back of the envelope calculation based on these known principles and literature values. I am not sure what new insights are provided.

We realize from this comment that we have not explained clearly enough the difference of measuring $\delta^{17}O$ and $\Delta^{17}O$. What the referee calls "$\Delta_A$ in leaf photosynthesis as for $\Delta_A{}^{18}O$" would be $\Delta_A{}^{17}O$. This was also shown in our original paper for consistency, but does indeed not provide additional information. Only the combination of $\Delta_A{}^{18}O$ and $\Delta_A{}^{17}O$ to $\Delta_A\Delta^{17}O$ provides the independent information. In the revised manuscript, we only present the results for $\Delta_A{}^{18}O$ and $\Delta_A\Delta^{17}O$. Some of the confusion may have to do with the notation, because the plant communities and atmospheric communities have used the symbol $\Delta$ for different quantities that are both used here.

Our final estimate of GPP is not dependent on the individual $\delta^{17}O$ and $\delta^{18}O$ values, but only on $\Delta_A\Delta^{17}O$. It is indeed a box model calculation, but to incorporate more variability, the entire mechanism would need to be incorporated in a global model. We are considering to implement this in the future, but for the box model presentations in this paper we have used global estimate of $\Delta^{17}O$ of $CO_2$ and leaf water from a recent 3D global $\Delta^{17}O$ study (Koren et al., 2019).

And so, while the experimental setup, measurements, and going through the isotopic theory are impressive and seems to be well done on first look, I think the authors have to re-think the presentation and provide a better justification of what in these measurements takes advantage of any mass independent effects (as declared), and in what ways this goes beyond a sophisticated confirmatory report.

We realized already in the preparation of the manuscript that the presentation was difficult, and the referee comment confirms this. Nevertheless, we still think that the four conclusions

identified above (copied below) make this a valuable study, whereas the referee sees only point 1 as significant merit.

a) demonstrate that the established theory is applicable to $\Delta^{17}O$-$CO_2$ exchange at leaf-level
b) experimentally quantify for the first time the effect of photosynthesis on $\Delta^{17}O$ of atmospheric $CO_2$
c) study of the dependence of this effect on critical parameters
d) provide an independent bottom-up $\Delta^{17}O$-isoflux estimate based on these lab experiments.

Barkan, E., and Luz, B.: High precision measurements of $^{17}O/^{16}O$ and $^{18}O/^{16}O$ ratios in $H_2O$, Rapid Commun. Mass. Sp., 19, 3737-3742, 10.1002/rcm.2250, 2005.

Barkan, E., and Luz, B.: Diffusivity fractionations of $H_2^{16}O/H_2^{17}O$ and $H_2^{16}O/H_2^{18}O$ in air and their implications for isotope hydrology, Rapid Commun. Mass. Sp., 21, 6, 2007.

Barkan, E., and Luz, B.: High-precision measurements of $^{17}O/^{16}O$ and $^{18}O/^{16}O$ ratios in $CO_2$, Rapid Commun. Mass. Sp., 26, 2733-2738, 10.1002/rcm.6400, 2012.

Koren, G., Schneider, L., Velde, I. R. v. d., Schaik, E. v., Gromov, S. S., Adnew, G. A., D.J.Mrozek, Hofmann, M. E. D., Liang, M.-C., Mahata, S., Bergamaschi, P., Laan-Luijkx, I. T. v. d., Krol, M. C., Röckmann, T., and Peters, W.: Global 3-D Simulations of the Triple Oxygen Isotope Signature $\Delta^{17}O$ in Atmospheric $CO_2$ J. Geophys. Res-Atmos.
, 124, 28, 2019.

Landais, A., Barkan, E., Yakir, D., and Luz, B.: The triple isotopic composition of oxygen in leaf water, Geochim. Cosmochim. Ac., 70, 4105-4115, 10.1016/j.gca.2006.06.1545, 2006.

McManus, J. B., Nelson, D. D., Shorter, J. H., Jimenez, R., Herndon, S., Saleska, S., and Zahniser, M.: A high precision pulsed quantum cascade laser spectrometer for measurements of stable isotopes of carbon dioxide, J. Mod. Optic., 52, 12, 2005.

Young, E. D., Galy, A., and Nagahara, H.: Kinetic and equilibrium mass-dependent isotope fractionation laws in nature and their geochemical and cosmochemical significance, Geochim. Cosmochim. Ac., 66, 9, 2002.

---

## Author Comment (AC2) · 13 May 2020

Referee 2

Thank you for your constructive comments. We carefully went through all the comments and suggestions and have adjusted the manuscript according to the comments made. The answers to the questions/ comments and suggestions are stated below each comment.

Please note the added supplement where the responses are given with proper formatting and detailed caption of figure 1 is provided.

[Figure]

The triple oxygen isotopic composition of CO2 ($\Delta$17OCO2) had been regarded as spatiotemporally constant in the troposphere because of its short residence time (e.g., Luz et al., 2000). Recently, significant seasonal and temporal variations of $\Delta$17OCO2 were first revealed in the atmosphere near the surface by Hofmann et al. (2017) and Liang et al. (2017), respectively, both of which were mainly controlled by the interaction of CO2 between the atmosphere and biosphere. These studies were then followed by the three dimensional simulation study with an atmospheric physico-chemical model (Koren et al., 2019), to quantify the global CO2 budget. The next step, therefore, must be the process study involving oxygen isotope fractionations in association with individual CO2 fluxes.

This study by Adnew, Pons, Koren, Peters, Röckmann, aims to quantify the $\Delta$17OCO2 change during photosynthetic CO2 removal from the atmosphere, caused by tiny difference of 17O-18O relationship between kinetic and equilibrium isotope fractionations inside the leaf.

To my knowledge, this is the first experimental study for $\Delta$17OCO2 at the leaf-scale; thus, their results provided must be important. However, I am frustrated and feel difficult to plough through the manuscript because 1) the structure of the manuscript (context) seems scattered, 2) experimental results (raw data) were not shown although values in all graphs were processed, 3) there appears a lot of faults in equations or figure number in the main text, and 4) it's a mixture of lengthy and in-short explanations. I strongly recommend the authors to revise the manuscript more simply and concisely.

We thank the referee for acknowledging the relevance of our study. We realize that the manuscript is quite difficult. We, therefore, thank the referee for the concrete suggestions below (including the suggesting for shortening), which helped us to improve the general storyline and readability.

General comments

1) # It spent 11 of 18 pages (until conclusion) from the Introduction to "Materials and

methods (M&M)." It seems too dominant; in other words, Results and Discussion seem too short. There appears a lengthy description in M&M, and the description for experimental results is too short.

In the revised manuscript, we reduced the description of the materials and methods section. The introduction, from line 91 to 106 was shortened and combined with the previous paragraph We shortened the materials and methods section and moved part of it to the supplementary material Section 2.1 was shortened based on the recommendation of the referee We shortened the theory part, line 170 to 201 in section 2.2 was moved to supplementary material. Section 2.3 was moved to the discussion section Section 2.4 was moved to the supplementary material We have also extended the results section following the concrete suggestions as described below.

2) # L84-90: This block appears the center of your motivation; however, there is no specific description of what the problem or limitation exists currently. Until this block (and perhaps in previous studies), you mentioned the $\Delta 17O$ is free from any terrestrial MDF processes and made readers believe that $\Delta 17O$ be a more robust tracer for estimating GPP. You must describe what actual problems lying among previous studies such as inconsistency, uncertainty, speculation, assumption and so on. Without this explanation, readers could not have motivations to read the next pages. I strongly recommend adding descriptions for the different slopes of three-isotope plots due to the different MDF processes.

Thank you very much for your suggestion. Indeed, different MDF processes with different three-isotope slopes are involved, and in the revised manuscript we incorporated the following schematic figure (Figure 1) to illustrate this point and to illustrate the objective of our study.

Furthermore, we have reformulated our motivation. The key point is that so far, the three-isotope slope of each of the processes that participate in plant-atmosphere gas exchange has been studied individually in an idealized experiment. The overall effect

of all processes, which work together in complex interaction, on $\triangle 17O$ has never been evaluated in a real plant exchange experiment. This is what is achieved in the research described in this manuscript and it is explicitly stated in the revised version.

3) # I strongly recommend the authors to revise the Theory part completely. The structure is scattered and forces readers to jump frequently between the main text, Appendix, and Supplementary Materials (SM).

We revised the theory part of the manuscript and incorporated your suggestions into the modified manuscript.

4) # Appendix should be moved to SM.

In the revised manuscript, the appendix is moved to the supplementary material.

5) # The term "fractionation" should be replaced to "isotope fractionation" for all.

We use isotope fractionation instead of fractionation alone throughout the revised manuscript.

6) # My major concern is the relation between dots of "Farquhar model" and curves in Figs 4 and 5a) and related description in Section 3.6. If I were not misunderstanding, both are results calculated from the "Farquhar model." Dots were obtained by giving several observed results and curves were simulated by giving similar boundary conditions to the experimental setting. Is the former necessary? This is very confusing.

We are sorry for the confusion, but the two are not the same. The curves are based on the leaf cuvette model which we implemented for this study and the blue diamonds were the results for the individual experiments using the Farquhar model. In the revised manuscript, we excluded the blue diamond points because this is not really necessary for our line of argumentation.

7) # I strongly recommend the authors to provide "List of symbols." for all parameters used and defined.

[Figure]

In the revised manuscript, the list of symbols for all parameters used in this study are provided

8) # The parameter cm seems one of the most important numbers in this study. For obtaining this, only $\delta 18O$ and $\alpha 18$ values were used concerning isotope ratio, though. Is it possible to use $\Delta 17O$ and $\lambda$ values to evaluate cm instead? At least does it make sense to test its feasibility?

Yes, it is possible to calculate the mole fraction of $CO_2$ at the $CO_2$-$H_2O$ exchange site (cm) using the $\Delta 17O$ and $\lambda$ values. Since this requires the development of yet another complicated set of equations and detailed discussion of the process of assimilation from a plant physiology point of view, it would make our paper even more complex and less focused. A companion manuscript with detailed description and derivation of the cm using $\Delta 17O$ and $\lambda$ values is under preparation.

9) # As shown in Figure 5, the discrimination of $\Delta 17O$ of $CO_2$ during photosynthesis varies widely, and controlled by the magnitude of oxygen isotope equilibration at the $CO_2$-$H_2O$ site, that is to say, the relative contribution of kinetic (diffusion) and equilibrium isotope fractionation. This conclusion is almost identical to the knowledge using conventional $\delta 18O$ results. Moreover, In the last paragraph of Discussion, authors mentioned that the main uncertainty is cm/ca ratio, which may be same as the main uncertainty of $\delta 18O$. My impression after reading this manuscript is that the intra-MDF variation dominate that of MIF signature on tropospheric $CO_2$, which weakens the merit to study $\Delta 17O$ of $CO_2$. What is an advantage to use $\Delta 17O$ instead of $\delta 18O$? Please provide suggestions or implications to general biogeochemists.

The referee is correct that the processes that affect $\delta 18O$ are the same that affect $\Delta 17O$. Nevertheless, the quantitative evaluation of $\Delta 17O$ is largely independent of $\delta 18O$. The limitation of using $\delta 18O$ of atmospheric $CO_2$ as a tracer is its dependency on the $\delta 18O$ value of different water reservoirs and fractionation processes in the hydrological cycle, water isotopic inhomogeneity, and dynamics, which are difficult to

ascertain (Hoag et al., 2005). Unlike $\delta$18O, $\Delta$17O variation is much smaller and is better defined (Miller, 2018). This is because conventional bio-geo-chemical processes that modify $\delta$17O and $\delta$18O follow well-defined three-isotope fractionation slope. Consequently, the formulation of the CO2 budget using $\Delta$17O is a lot simplified, compared to using $\delta$18O. Furthermore, unlike $\delta$'s, $\lambda$ is insensitive to temperature (Cao and Liu, 2011;Bao et al., 2016;Hofmann et al., 2012;Dauphas and Schauble, 2016;Miller, 2018).

Specific comments

10) L41: "replaced using. . ." What this means? Be more specific.

In the revised manuscript it is replaced with "replicated based on cross-consistency checks with atmospheric inversions, sun-induced fluorescence (SIF) and dynamic global vegetation models"

11) L47: "see equation (1)" instead of "see below"

In the revised manuscript we used "see equation (1)"

12) L51: "the latter term" I guess it should be "the former term," which means photosynthetic CO2 uptake.

Thank you, in the revised manuscript corrected it to "the former term"

13) L53: "variable $\delta$18O gradient" I think "significant $\delta$18O variation" is more appropriate.

In the revised manuscript we used significant $\delta$18O variation

14) L56: Delete "the isotopically exchanged"

Deleted

15) L45-57: In this block, you should use the term "isotope fractionation" with its definition for the subsequent block. More desirably, the term "mass-dependent isotope fractionation (MDF)" with its definition.

In the revised manuscript, we included "These physico-chemical processes change 17O/16O by approximately half the corresponding change in 18O/16O, a process called mass-dependent isotope fractionation (see equation 2). This is because the mass difference between 17O and 16O (1.0042 amu) is approximately half as large as the mass difference between 18O and 16O (2.0042 amu). "at the end of the paragraph.

16) L63: "mass-dependent fractionation" should be "mass-dependent isotope fraction-ation" with its definition in detail.

We excluded this paragraph, L62-64 in the revised manuscript since it does not add additional information to the paragraph mentioned above.

17) L62-64: Need revision because the latter paragraph is just a refrain of the former.

We excluded this paragraph, L62-64 in the revised manuscript since it does not add additional information.

18) L65: Describe a specific value instead using "considerable"

In the revised manuscript, instead of " considerable $\Delta17O$" we used "the $\delta17O$ of CO2 is 1.7 to 2.2 times $\delta18O$ of CO2 (Wiegel et al., 2013)"

19) L60-71: In this block, you should use the term "mass-independent isotope fraction-ation (MIF)" with its definition, and associate it with "photochemical isotope exchange"

In the revised manuscript, we included the following paragraph " In nature, it was be-lieved all process that modifies the oxygen isotope distribution is mass-dependent isotope fractionation until the discovery of the deviation from the assigned mass-dependent three-isotope fractionation line in meteorites (Clayton et al., 1973;Clayton et al., 1976) and ozone formation (Thiemens, 1983;Heidenreich and Thiemens, 1983, 1986), called mass-independent isotope fractionation (see equation 3). The $\Delta17O$ of ozone can be transferred to other oxygen-bearing molecules via a direct chemical reaction with ozone or via O(1D)."

and rearranged the whole paragraph

20) L70-71: This is not sufficient because exchanges with soil and ocean water are also nonenzymatic processes.

The isotope exchange in the atmosphere is negligible due to lower liquid water content, lower residence time and the absence of carbonic anhydrase (Mills and Urey, 1940;Johnson, 1982;Miller et al., 1971;Silverman, 1982). We incorporated this sentence in the updated section.

It is true that CO2-H2O exchange with ocean water is a non-enzymatic process, but CO2-H2O exchange with soil water is controlled by carbonic anhydrase (Wingate et al., 2009), similar to the exchange with leaf water.

21) L78: "The $\Delta$17O of CO2" instead of "The 17O-excess of CO2 ($\Delta$17O) (equation 4)"

In the revised manuscript, we only used $\Delta$17O

22) L80: Clarify "well-known three-isotope slope." "Non three-isotope person" cannot understand what this means.

In the revised manuscript, we included the three-isotope fractionation slope of 0.5229, and the figure above.

23) L92-106 and Figure 1: The explanation is this block is too general, should reduce to a few sentences. Detail description may be required if you would like to discuss the difference of results due to the different types in the Discussion. As for Figure 1, not this scheme but simpler scheme in Figure S6 was actually used in this study. Therefore, it seems more appropriate to delete Figure 1 and insert S6 here.

In the revised manuscript, we merged the necessary information with the other paragraphs and we agree that Figure 1 is not necessary, so it is left out. We excluded the general description of plant types. We only kept the following three sentences

"The mole fraction of $CO_2$ at the $CO_2$-$H_2O$ exchange site (cm) is an important parameter to determine the effect of photosynthesis on the triple oxygen isotope composition of atmospheric $CO_2$. In C3 plants, CA is found in the chloroplast, cytosol, mitochondria and plasma membrane (Fabre et al., 2007;DiMario et al., 2016) and the $CO_2$-$H_2O$ exchange can occur anywhere between the plasma membrane and the chloroplast. For C4 plants, CA is mainly found in the cytosol, the $CO_2$-$H_2O$ exchange occurs in the cytosol (Badger and Price, 1994)."

24) L108-109: What is "leaf level"?

In the revised manuscript we changed it from leaf level to leaf scale

25) L116-117: "$\Delta$17O" instead of "triple oxygen isotopic composition"

Changed accordingly

26) Equations 1 and 2: Should be merged such as, $\delta n$ O = n Rsample/ n RVSMOW – 1, n refers 17 or 18 or simpler, $\delta$ = Rsample/RVSMOW – 1.

Thank you, in the revised manuscript we used the first suggestion.

27) L134: I recommend "The MDF factor" instead of "The factor"

Changed accordingly

28) L135-137: Delete "This relation..., respectively.

In the revised manuscript we excluded the sentence, based on the suggestion above, we already defined mass-dependent isotope fractionation.

29) L137: "variations" instead of "values." "Small delta value" is meaningless.

In the revised manuscript we changed values to variations. And at the end of the paragraph, we introduced "Equation 4 can be linearized to $\Delta$17O=$\delta$17 O-$\lambda\times\delta$18 O (Miller, 2002), but this approximation causes an error that increases with $\delta$18 O." for more clarity.

30) L139-140: I recommend "Note that Δ17O changes not only by MIF processes, but also MDF processes with a different $\lambda$ value from the definition,"

Changed accordingly

31) L145-146: "which was obtained by the observation of" instead of "the value associated with"

Changed accordingly

32) L147-148: Delete "Note that ... $\delta$18O."

Changed accordingly

33) L150-258 (Section 2.2-2.4): Revise completely.

In the revised manuscript, we moved most of section 2.2 to the supplementary material we moved section 2.3 to the discussion we moved section 2.4 to the supplementary material

34) Equation 5: Use n (18 or 17) or simpler expression as above, then revise or delete

In the revised manuscript, we implemented the suggestion expression

35) L158 and L163. Equation 12: Move after equation 5 with related sentences.

Changed accordingly

36) L163-168: Delete "We note that...itself."

The sentence is excluded from the revised manuscript.

37) L170-200 and Section 2.4: Integrate and locate in new section such like "Extension of Farquhar-Lloyd model to oxygen triple isotopes. Eqs. 6 and 11 are almost identical so that they should be merged. Equation 15: Use n (18 or 17) or simpler expression, then revise or delete

This section is moved to the supplementary material and revised in the new version of

the manuscript

38) L208-213 and Figure 2: Move to SM.

Changed accordingly

39) Section 2.3: I recommend moving this section to the Discussion.

Changed accordingly

40) L217: Delete "which is a net sink,"

This section is moved to the discussion part, and "which is a net sink," is removed in the revised manuscript.

41) L230: Specify which model is used.

In the revised manuscript this section is moved to the discussion section and revised entirely.

42) L241-259: Here detail but still insufficient description was made only for $\delta m$, on the other hand, no description for ci and $\delta i$ which were driven away to Appendix. This seems out of balance and forces readers to jump here and there. I recommend moving this block to SM.

In the revised manuscript, we moved this part to the supplementary material

43) L256-257 and related sentences in Appendix A3. No definition of ci.

In the revised manuscript, the definition for all parameters is included as a table in the appendix

44) L262-265: Could it be shorter?

In the revised manuscript we shortened this part by excluding the sentence from line 263 to line 265, "The dwarf type sunflowers were grown until the first leaf pair that was used for the experiments reached the final size, which is about 4 weeks." We did the

same for line 267 to 268, i.e. "After at least 6 weeks in the growth chamber, leaves that had developed and matured there were used for the experiment" is excluded in the revised manuscript.

45) L268-269: "The 4th or higher..." Is this sentence an explanation for maize or all species?

In the revised manuscript, we write "For maize, the 4th or higher …. "

46) Section 3.2: Need the model and the manufacturer for halogen lamp, neutral filters, dewpoint meter (the model).

The models and manufacturers are included in the revised manuscript

47) Section 3.3: Could this section be shorter to several sentences? The description for $\delta$D and obtaining optimum setting seem appropriate in SM.

In the revised manuscript this section has been considerably shortened and part of it is moved to the supplementary material.

48) L349: Water was converted to O2

In the revised manuscript L349 - L 354 has been deleted to make the manuscript more concise.

49) Section 3.5: In previous section, unit of $\Delta$17O is ‰. Here ppm is used. Use a uniform manner.

In the revised manuscript, all numbers are given in ‰

50) Section 3.6: See related general comment

The leaf cuvette model is described here for the first time, as a result we cannot make it shorter than this.

51) L403: The last sentence is a refrain.

In the revised manuscript, we excluded the sentence.

52) Results: Show experimental results (raw data) such as c, $\delta$, $\Delta$, w, for entering and leaving from the cuvette, etc. Show a table of them and describe them.

In the revised manuscript, we provided the raw data for the gas exchange parameters at the beginning of the results section

53) L414-415: Delete this sentence

Changed accordingly

54) Section 4.2: Avoid using "17O-excess" in the title and L433 for uniformity

Changed accordingly

55) L477-493: I could not understand this block. If the authors applied different lambda values to individual results, the vertical axis in Figure 8 would be meaningless, and one could not evaluate the graph and related description at all.

In the revised manuscript we incorporated the reference triple oxygen isotope fractionation slope ($\lambda$), also in the caption. Sorry for the confusion we did not mention it clearly. When we described $\Delta$17O in the theory section, we clearly mentioned which lambda value we used ($\lambda$=0.528).

56) Section 5.2: Avoid using "17O-excess" for uniformity Changed accordingly

57) Figure 3: Add individual flow direction. Changed accordingly

58) Figure 4: Panel b seems unnecessary. Delete and insert Figure 5a here.

Changed accordingly

59) Figure 5: Move Panel a to Figure 4 as above

Changed accordingly

60) Figure 6: Is it important to plot both of blue diamonds and curve. Should the curve

be improved by blue diamonds?

The curves are based on the leaf cuvette model which we implemented for this study. The blue diamonds were the results for the individual experiments using the Farquhar model. In the revised manuscript, we excluded the blue diamond points.

Typographic errors

61) Space inserted after semicolon (e.g., L33) corrected 62) L42: Welp et al. (2011) Corrected

63) L45: The concept of the latter study.. Corrected

64) L60: equation 4)) Corrected, now it is "see equation 2"

65) L207: Figure 2 Now Figure S1

66) L237: "Following (Farquhar.…..)" Need grammatical correctness

Corrected to: "The CO2 mole fraction at the site of CO2-H2O exchange is calculated as shown in equation S10 following (Farquhar and Cernusak, 2012;Barbour et al., 2016;Osborn et al., 2017). " This section has also been moved to the supplementary material.

67) L267: Maize In the revised manuscript, "Mays" is corrected to "Maize"

68) L279, L297: Need grammatical correctness. Line 279

In the revised manuscript "of" is replaced by "for", now it reads as follow: A schematic for the gas exchange experimental setup is shown in Figure 2

L297 In the revised manuscript, we deleted the phrase "as described in detail in" since it does not change the meaning of the sentence. Now it reads as:

"The isotopically enriched CO2 was prepared by photochemical isotope exchange between CO2 and O2 under UV irradiation (Adnew et al., 2019). "

69) Section 3.2: "Figure 3" instead of "Figure 2" (If Figure 2 were moved to SM, they are accidentally correct, though)

Thank you, this is corrected in the revised manuscript

70) References: I found typo. in Barbour et al. (2016) and Caemmerer and Farquhar (1981). There may be more. Confirm all.

Thank you very much. In the revised manuscript we corrected all of them. All of them related to the name von Caemmerer.

73) L950: "entering and leaving" instead of "leaving and entering" In the revised manuscript, we re-ordered it chronologically. The appendix has also been moved to the supplementary material.

74) Equation A1.4: If the referred article (Caemmerer and Farquhar, 1981) was correct, the denominator must be (gt ac + E/2).

Corrected Thank you very much, all the Typographic errors are corrected in the revised manuscript.

References

Adnew, G. A., Hofmann, M. E. G., Paul, D., Laskar, A., Surma, J., Albrecht, N., Pack, A., Schwieters, J., Koren, G., Peters, W., and Röckmann, T.: Determination of the triple oxygen and carbon isotopic composition of CO2 from atomic ion fragments formed in the ion source of the 253 Ultra High‐Resolution Isotope Ratio Mass Spectrometer, Rapid Commun. Mass. Sp., 33, 17, 2019. Badger, M. R., and Price, G. D.: The role of carbonic anhydrase in photosynthesis, Annu. Rev. Plant Biol., 45, 23, 1994. Bao, H., Cao, X., and Hayles, J. A.: Triple oxygen isotopes: fundamental relationships and applications, Annual Review of Earth and Planetary Sciences, 44, 29, 2016. Barbour, M. M., Evans, J. R., Simonin, K. A., and Caemmerer, S. v.: Online CO2 and H2O oxygen isotope fractionation allows estimation of mesophyll conductance in C4 plants, and reveals that mesophyll conductance decreases as

leaves age in both C4 and C3 plants, New Phytol., 14, 2016. Barkan, E., and Luz, B.: High precision measurements of 17O/16O and 18O/16O ratios in H2O, Rapid Commun. Mass. Sp., 19, 3737-3742, 10.1002/rcm.2250, 2005. Barkan, E., and Luz, B.: Diffusivity fractionations of H216O/H217O and H216O/H218O in air and their implications for isotope hydrology, Rapid Commun. Mass. Sp., 21, 6, 2007. Barkan, E., and Luz, B.: High-precision measurements of 17O/16O and 18O/16O ratios in CO2, Rapid Commun. Mass. Sp., 26, 2733-2738, 10.1002/rcm.6400, 2012. Cao, X., and Liu, Y.: Equilibrium mass-dependent fractionation relationships for triple oxygen isotopes, Geochimica et Cosmochimica Acta, 75, 7435-7445, 10.1016/j.gca.2011.09.048, 2011. Clayton, R. N., Grossman, L., and Mayeda, T. K.: A component of primitive nuclear composition in carbonaceous meteorites, Science, 182, 3, 1973. Clayton, R. N., Onuma, N., and Mayeda, T. K.: A classification of meteorites based on oxygen isotopes, Earth and Planetary Science Letters, 30, 8, 1976. Dauphas, N., and Schauble, E. A.: Mass fractionation laws, mass-independent effects, and isotopic anomalies, Annu Rev Earth Pl Sc, 44, 74, 2016. DiMario, R. J., Quebedeaux, J. C., Longstreth, D. J., Dassanayake, M., Hartman, M. M., and Moroney, J. V.: The cytoplasmic carbonic anhydrases $\beta$CA2 and $\beta$CA4 are required for optimal plant growth at low CO2, Plant Physiol., 171, 13, 2016. Fabre, N., Reiter, I. M., Becuwe-Linka, N., Genty, B., and Rumeau, D.: Characterization and expression analysis of genes encoding alpha and beta carbonic anhydrases in Arabidopsis, Plant Cell Environ, 30, 617-629, 10.1111/j.1365-3040.2007.01651.x, 2007. Farquhar, G. D., and Cernusak, L. A.: Ternary effects on the gas exchange of isotopologues of carbon dioxide, Plant Cell Environ, 35, 1221-1231, 10.1111/j.1365-3040.2012.02484.x, 2012. Heidenreich, J. E., and Thiemens, M. H.: A non‐mass‐dependent isotope effect in the production of ozone from molecular oxygen The Journal of Chemical Physics, 78, 3, 1983. Heidenreich, J. E., and Thiemens, M. H.: A non‐mass‐dependent oxygen isotope effect in the production of ozone from molecular oxygen: The role of molecular symmetry in isotope chemistry The Journal of chemical physics, 84, 5, 1986. Hoag, K. J., Still, C. J., Fung, I. Y., and Boering, K. A.: Triple oxygen

isotope composition of tropospheric carbon dioxide as a tracer of terrestrial gross carbon fluxes, Geophys. Res. Lett., 32, 10.1029/2004gl021011, 2005. Hofmann, M. E. G., Horváth, B., and Pack, A.: Triple oxygen isotope equilibrium fractionation between carbon dioxide and water, Earth and Planetary Science Letters, 319, 5, 2012. Hofmann, M. E. G., Horváth, B., Schneider, L., Peters, W., Schützenmeister, K., and Pack, A.: Atmospheric measurements of $\Delta 17O$ in CO2 in Göttingen, Germany reveal a seasonal cycle driven by biospheric uptake, Geochim. Cosmochim. Ac., 199, 143-163, 10.1016/j.gca.2016.11.019, 2017. Johnson, K. S.: Carbon dioxide hydration and dehydration kinetics in seawater 1, Limnology and Oceanography, 27, 6, 1982. Koren, G., Schneider, L., Velde, I. R. v. d., Schaik, E. v., Gromov, S. S., Adnew, G. A., D.J.Mrozek, Hofmann, M. E. D., Liang, M.-C., Mahata, S., Bergamaschi, P., Laan-Luijkx, I. T. v. d., Krol, M. C., Röckmann, T., and Peters, W.: Global 3‐D Simulations of the Triple Oxygen Isotope Signature $\Delta 17O$ in Atmospheric CO2 J. Geophys. Res-Atmos., 124, 28, 2019. Landais, A., Barkan, E., Yakir, D., and Luz, B.: The triple isotopic composition of oxygen in leaf water, Geochim. Cosmochim. Ac., 70, 4105-4115, 10.1016/j.gca.2006.06.1545, 2006. Liang, M.-C., Mahata, S., Laskar, A. H., Thiemens, M. H., and Newman, S.: Oxygen isotope anomaly in tropospheric CO2 and implications for CO2 residence time in the atmosphere and gross primary productivity, Sci Rep, 7, 13180, 10.1038/s41598-017-12774-w, 2017. Miller, M. F.: Isotopic fractionation and the quantification of 17O anomalies in the oxygen three-isotope system: an appraisal and geochemical significance, Geochimica et Cosmochimica Acta, 66, 8, 2002. Miller, M. F.: Precipitation regime influence on oxygen triple-isotope distributions in Antarctic precipitation and ice cores, Earth and Planetary Science Letters, 481, 11, 2018. Miller, R. F., Berkshire, D. C., Kelley, J. J., and Hood, D. W.: Method for determination of reaction rates of carbon dioxide with water and hydroxyl ion in seawater, Environmental Science & Technology, 5, 6, 1971. Mills, G. A., and Urey, H. C.: The kinetics of isotopic exchange between carbon dioxide, bicarbonate ion, carbonate ion and water, Journal of the American Chemical Society, 62, 7, 1940. Osborn, H. L., Alonso-Cantabrana, H., Sharwood,

[Figure]

R. E., Covshoff, S., Evans, J. R., Furbank, R. T., and Caemmerer, S. v.: Effects of reduced carbonic anhydrase activity on CO2 assimilation rates in Setaria viridis: a transgenic analysis, J. Exp. Bot., 68, 11, 2017. Silverman, D. N.: Carbonic anhydrase: Oxygen-18 exchange catalyzed by an enzyme with rate-contributing Proton-transfer steps Methods in enzymology, Elsevier, 1982. Thiemens, M. H., and Heidenreich, J.E. III: Mass-independent fractionation of oxygen: a novel isotope effect and its possible cosmochemical implications, Science 10.1126/science.219.4588.1073, 1983. Wiegel, A. A., Cole, A. S., Hoag, K. J., Atlas, E. L., Schauffler, S. M., and Boering, K. A.: Unexpected variations in the triple oxygen isotope composition of stratospheric carbon dioxide, PNAS, 110, 17680-17685, 10.1073/pnas.1213082110, 2013. Wingate, L., Ogée, J., Cuntz, M., Genty, B., Reiter, I., Seibt, U., Yakir, D., Maseyk, K., Pendall, E. G., Barbour, M. M., Mortazavi, B., Burlett, R., Peylin, P., Miller, J., Mencuccini, M., Shim, J. H., Hunt, J., and Grace, J.: The impact of soil microorganisms on the global budget of $\delta$18O in atmospheric CO2, PNAS, 106, 4, 2009. Young, E. D., Galy, A., and Nagahara, H.: Kinetic and equilibrium mass-dependent isotope fractionation laws in nature and their geochemical and cosmochemical significance, Geochim. Cosmochim. Ac., 66, 9, 2002.

Please also note the supplement to this comment:
https://www.biogeosciences-discuss.net/bg-2020-91/bg-2020-91-AC2-supplement.pdf

―――――――――――――――――

[Figure]

**Fig. 1.** Schematic of the process that affect the Δ17O of CO2 and H2O during photosynthetic gas exchange (not to scale)

---

## Author Response (AR1)

From: Getachew Agmuas Adnew and co-authors

To: - Dear Dr. Aninda Mazumdar

5  Thank you for editing and facilitating the review process of our manuscript.

We thank both the anonymous reviewers for their valuable and constructive feedbacks. The answers for each reviewer's question are included under each question of the reviewer.

Referee: 1

The answers to the questions/ comments and suggestions are stated below each comment.

15  As far as I understand the justification for the great effort required in measuring $^{17}O$ and its "access" (or anomaly), is the discovery of significant mass independent oxygen isotope effects in the stratosphere that is conserved to some extent in the troposphere (seems to be true both for atmospheric $O_2$ and $CO_2$). The extent to which this anomaly is conserved in the troposphere depends on the $CO_2$ (or $O_2$) cycling through the biosphere, which erases it by exchange with water. Thus, if the stratospheric production of the
20  anomaly is known and it is relatively constant, the residual signal in the troposphere should reflect the biosphere productivity (GPP). This is exiting application considering the uncertainty around GPP.

This summary by the reviewer is correct. We would like to emphasize that if we reliably want to estimate GPP from $\Delta^{17}O$, we need to know the *precise* effect of photosynthesis and respiration on $\Delta^{17}O$, in the
25  words of the referee, how does the $\Delta^{17}O$ signature actually look after being "erased" by exchange with the biosphere.

ALL the processes associated with the Biosphere, including leaf gas exchange studied here, seems to be
30  mass dependent and are FULLY covered by the conventional $^{18}O$ studies.

The referee is correct that in principle $^{18}O$ indeed cycles through the same biological system, and undergoes the same (bio)physical processes. However, we would like to nuance the idea that $\delta^{18}O$ can help us FULLY understand ALL processes of interest. This is because conventional $\delta^{18}O$ studies have a
35  number of distinct disadvantages. Notably, the $\delta^{18}O$ -signature of all water pools in the system must be known to use $\delta^{18}O$ as a carbon cycle tracer. In addition, significant changes in $\delta^{18}O$ can occur due to processes that are not of primary interest to understanding GPP, e.g., leaf evaporation, or soil equilibration.

40    $\Delta^{17}O$ variation due to kinetic and equilibrium fractionation effects is much smaller and is better defined. This is because conventional bio-geo-chemical processes that modify $\delta^{17}O$ and $\delta^{18}O$ follow a well-recognized isotope fractionation slope. In earlier studies many assumptions had to be made, because the effect on $\Delta^{17}O$ had never been quantified precisely. This is now accomplished through our study.

45    **The only exception may be the small variations observed in the lambda factor that define the expected ratio of $^{18}O$ to $^{17}O$ mass dependent discrimination ($\sim$0.5), which is not studied here**.

   The reviewer correctly identifies that the small variations in $\lambda$ values can impact the $\Delta^{17}O$ we measure. These effects have been studied previously and the three isotope slopes have been established and are
50    used in our study. We have included an additional figure (reproduced below) in the revised manuscript, which shows conceptually how the three-isotope slopes differ between the various processes and how they affect the observed $\Delta^{17}O$ signals.
   In addition, we also include for the first-time experiments and model studies that involve artificially $^{17}O$ labeled $CO_2$. We demonstrate how the resulting differences in $\Delta^{17}O$ between $CO_2$ and leaf water affect
55    the results, and that experiments with $^{17}O$ labeled $CO_2$ actually increase the signal to (measurement) noise ratio.

[Figure]

Figure 1. Schematic representation of the processes that affect the $\Delta^{17}O$ of $CO_2$ and $H_2O$ during
60 photosynthetic gas exchange (not to scale). The triple oxygen isotope slope for transpiration at h=75 % relative humidity is $\theta_{trans}$=0.522-0.008 ×h = 0.516 (Landais et al., 2006). The triple isotope slope of $CO_2$-$H_2O$ exchange is $\theta_{CO2-H2O}$ = 0.5229 (Barkan and Luz, 2012). The triple isotope slope for diffusion of $CO_2$ is $\theta_{CO2-diff}$ = 0.509 (Young et al., 2002) and for diffusion of water vapor $\theta_{H2O(v)-diff}$ = 0.518 (Barkan and Luz, 2007). The three isotope slope for equilibration between liquid (l) and gaseous (v) water is $\theta_{H2O(v)-}$
65 $_{H2O(l)}$ = 0.529 (Barkan and Luz, 2005). $\varepsilon^{18}O$ is the enrichment or depletion in $^{18}O$ due to the corresponding isotope fractionation process.

And so, while the present paper goes through an impressive exercise of gas exchange and isotopic measurements and calculations, I fail to see the purpose and merit of this exercise, beyond a test that verifies that indeed the [17]O measurements are consistent with the [18]O studies.

**The occlusions as much as I can see are already fairly well-known form [18]O studies and, in fact, much of the calculations here still depends on the [18]O measurements.**

We appreciate that the referee acknowledges the considerable analytical effort that was made to produce our results. As mentioned above, we think that $\delta^{18}O$ measurements alone are not sufficient to study all aspects related to gas exchange between plants and the atmosphere and to quantify GPP. Thus, we posit that an alternative independent tracer is still very useful, and in fact $\Delta^{17}O$ has been repeatedly suggested and already used as independent and potentially even superior tracer.

We nevertheless realize from the comment that the merit of our study was not communicated well, and we have considerably strengthened the motivation. The key point is that so far, the three-isotope slope of each of the processes that participate in plant-atmosphere gas exchange has been studied individually in an idealized experiment. The overall effect of all processes, which work together in complex interaction, on $\Delta^{17}O$ has never been evaluated in a real plant exchange experiment. This is what is achieved in the research described in out manuscript and it is explicitly stated in the revised version.

Specifically, the results communicated in our manuscript

1) demonstrate that the established theory is applicable to $\Delta^{17}O$-$CO_2$ exchange at leaf-level.

2) experimentally quantify for the first time the effect of photosynthesis on $\Delta^{17}O$ of atmospheric $CO_2$

3) quantify of the dependence of this effect on critical parameters

4) provide an independent bottom-up $\Delta^{17}O$-isoflux estimate based on these lab experiments.

Furthermore, we have now demonstrated that such studies are possible with IRMS methods, with considerable effort, but they may actually become more widely accessible thanks to novel laser instrumentation in the near future (McManus et al., 2005).

For example, the key results indicated in the Abstract are: "Our results demonstrate that two key factors determine the effect of gas exchange on the $\Delta17O$ of atmospheric CO2. The relative difference between $\Delta17O$ of the CO2 entering the leaf and the CO2 in equilibrium with leaf water, and the back-diffusion flux of CO2 from the leaf to the atmosphere, which can be quantified by the Cm/Ca ratio". Isn't it that these 'basic principles' of leaf gas exchange are already fairly well known from previous CO2 and the 18O studies?

We clearly acknowledge in our paper that the processes affecting $\delta^{18}O$ and $\Delta^{17}O$ are indeed the same, and in fact we use the established conceptual models, with appropriate references. Nevertheless, this is the first experimental leaf-scale study where the applicability of these theoretical concepts to $\Delta^{17}O$ is actually demonstrated.

It seems also that the notion of "discrimination against Δ17O of atmospheric CO2" is not clear. If this is confused with D in leaf photosynthesis as for D18, then again 17O is predictable and has no clear additional information (other than perhaps the reflection of the possible variations in the lambda factor). The final estimate of global 17O discrimination anomaly is back of the envelope calculation based on these known principles and literature values. I am not sure what new insights are provided.

We realize from this comment that we have not explained clearly enough the difference of measuring $\delta^{17}O$ and $\Delta^{17}O$. What the referee calls "$\Delta_A$ in leaf photosynthesis as for $\Delta_A^{18}O$" would be $\Delta_A^{17}O$. This was also shown in our original paper for consistency, but does indeed not provide additional information. Only the combination of $\Delta_A^{18}O$ and $\Delta_A^{17}O$ to $\Delta_A\Delta^{17}O$ provides the independent information. In the revised manuscript, we only present the results for $\Delta_A^{18}O$ and $\Delta_A\Delta^{17}O$. Some of the confusion may have to do with the notation, because the plant communities and atmospheric communities have used the symbol $\Delta$ for different quantities that are both used here.

Our final estimate of GPP is not dependent on the individual $\delta^{17}O$ and $\delta^{18}O$ values, but only on $\Delta_A\Delta^{17}O$. It is indeed a box model calculation, but to incorporate more variability, the entire mechanism would need to be incorporated in a global model. We are considering to implement this in the future, but for the box model presentations in this paper we have used global estimate of $\Delta^{17}O$ of $CO_2$ and leaf water from a recent 3D global $\Delta^{17}O$ study (Koren et al., 2019).

And so, while the experimental setup, measurements, and going through the isotopic theory are impressive and seems to be well done on first look, I think the authors have to re-think the presentation and provide a better justification of what in these measurements takes advantage of any mass independent effects (as declared), and in what ways this goes beyond a sophisticated confirmatory report.

We realized already in the preparation of the manuscript that the presentation was difficult, and the referee comment confirms this. Nevertheless, we still think that the four conclusions identified above (copied below) make this a valuable study, whereas the referee sees only point 1 as significant merit.

1) demonstrate that the established theory is applicable to $\Delta^{17}O$-$CO_2$ exchange at leaf-level
2) experimentally quantify for the first time the effect of photosynthesis on $\Delta^{17}O$ of atmospheric $CO_2$
3) study of the dependence of this effect on critical parameters
4) provide an independent bottom-up $\Delta^{17}O$-isoflux estimate based on these lab experiments.

Referee 2

Thank you for your constructive comments. We carefully went through all the comments and suggestions and have adjusted the manuscript according to the comments made. The answers to the questions/ comments and suggestions are stated below each comment.

The triple oxygen isotopic composition of CO2 (Δ17OCO2) had been regarded as spatiotemporally constant in the troposphere because of its short residence time (e.g., Luz et al., 2000). Recently, significant

seasonal and temporal variations of $\Delta 17OCO2$ were first revealed in the atmosphere near the surface by Hofmann et al. (2017) and Liang et al. (2017), respectively, both of which were mainly controlled by the interaction of CO2 between the atmosphere and biosphere. These studies were then followed by the three dimensional simulation study with an atmospheric physico-chemical model (Koren et al., 2019), to quantify the global CO2 budget. The next step, therefore, must be the process study involving oxygen isotope fractionations in association with individual CO2 fluxes.

This study by Adnew, Pons, Koren, Peters, Röckmann, aims to quantify the $\Delta 17OCO2$ change during photosynthetic CO2 removal from the atmosphere, caused by tiny difference of 17O-18O relationship between kinetic and equilibrium isotope fractionations inside the leaf.

To my knowledge, this is the first experimental study for $\Delta 17OCO2$ at the leaf-scale; thus, their results provided must be important. However, I am frustrated and feel difficult to plough through the manuscript because 1) the structure of the manuscript (context) seems scattered, 2) experimental results (raw data) were not shown although values in all graphs were processed, 3) there appears a lot of faults in equations or figure number in the main text, and 4) it's a mixture of lengthy and in-short explanations. I strongly recommend the authors to revise the manuscript more simply and concisely.

We thank the referee for acknowledging the relevance of our study. We realize ourselves that the manuscript is quite difficult. We therefore thank the referee for the concrete suggestions below (including the suggesting for shortening), which helped us to improve the general storyline and readability.

**General comments**

\# It spent 11 of 18 pages (until conclusion) from the Introduction to "Materials and methods (M&M)." It seems too dominant; in other words, Results and Discussion seem too short. There appears a lengthy description in M&M, and the description for experimental results is too short.

In the revised manuscript, we reduced the description of materials and methods section.
- The introduction, from line 91 to 106 was shortened and combined with the previous paragraph
- We shortened the materials and methods section and moved part of it to the supplementary material
- Section 2.1 was shortened based on the recommendation of the referee
- We shortened the theory part, line 170 to 201 in section 2.2 was moved to supplementary material.
- Section 2.3 was moved to the discussion section
- Section 2.4 was moved to the supplementary material

- We have also extended the results section following the concrete suggestions as described below.

\# L84-90: This block appears the center of your motivation; however, there is no specific description of what the problem or limitation exists currently. Until this block (and perhaps in previous studies), you mentioned the $\Delta 17O$ is free from any terrestrial MDF processes and made readers believe that $\Delta 17O$ be

a more robust tracer for estimating GPP. You must describe what actual problems lying among previous studies such as inconsistency, uncertainty, speculation, assumption and so on. Without this explanation, readers could not have motivations to read the next pages. I strongly recommend adding descriptions for
190  the different slopes of three-isotope plots due to the different MDF processes.

Thank you very much for the suggestion. Indeed, different MDF processes with different three-isotope slopes are involved, and in the revised manuscript we incorporated the following schematic figure to illustrate this point and to illustrate the objective of our study.

195

[Figure]

Figure 1 Schematic of the process that affects the $\Delta^{17}O$ of CO2 and H2O during the photosynthetic gas
200  exchange (not to scale). The three-isotope slopes $\theta$ for the individual isotope fractionation processes (both kinetic and equilibrium fractionation) are $\theta_{trans}=0.522-0.008 \times h =0.516$ for transpiration at h = 75 % relative humidity (Landais et al., 2006), $\theta_{CO2-H2O} =0.5229$ for isotope exchange between CO2 and H2O (Barkan and Luz, 2012), $\theta_{CO2-diff} = 0.509$ for the diffusion of CO2 (Young et al., 2002), $\theta_{H2O(v)-H2O(l)} = 0.529$ for the equilibrium between gas phase (v) and liquid (l) water, (Barkan and Luz, 2005)
205  and $\theta_{H2O(v)-diff}$ for the diffusion of water vapor (Barkan and Luz, 2007). $\varepsilon^{18}O$, indicated along the x-axis, is the isotope fractionation in 18O due to the responding process.

Furthermore, we have reformulated our motivation. The key point is that so far, the three-isotope slope of each of the processes that participate in plant-atmosphere gas exchange has been studied individually
210  in an idealized experiment. The overall effect of all processes, which work together in complex interaction, on $\Delta^{17}O$ has never been evaluated in a real plant exchange experiment. This is what is achieved in the research described in out manuscript and it is explicitly stated in the revised version.

**I strongly recommend the authors to revise the Theory part completely. The structure is scattered and forces readers to jump frequently between the main text, Appendix and Supplementary Materials (SM).**

We revised the theory part of the manuscript and incorporated your suggestions in the modified manuscript.

Appendix should be moved to SM.

In the revised manuscript, the appendix is moved to the supplementary material.

**The term "fractionation" should be replaced to "isotope fractionation" for all.**

We use isotope fractionation instead of fractionation alone throughout the revised manuscript.

**My major concern is the relation between dots of "Farquhar model" and curves in Figs 4 and 5a) and related description in Section 3.6. If I were not misunderstanding, both are results calculated from the "Farquhar model." Dots were obtained by giving several observed results and curves were simulated by giving similar boundary conditions to the experimental setting. Is the former necessary? This is very confusing.**

We are sorry for the confusion, but the two are not the same. The curves are based on the leaf cuvette model which we implemented for this study and the blue diamonds were the results for the individual experiments using the Farquhar model. In the revised manuscript, we excluded the blue diamond points because this is not really necessary for our line of argumentation.

**I strongly recommend the authors to provide "List of symbols." for all parameters used and defined.**

In the revised manuscript, the list of symbols for all parameters used in this study is provided

**The parameter cm seems one of the most important numbers in this study. For obtaining this, only $\delta 18O$ and $\alpha 18$ values were used concerning isotope ratio, though. Is it possible to use $\Delta 17O$ and $\lambda$ values to evaluate cm instead? At least does it make sense to test its feasibility?**

Yes, it is possible to calculate the mole fraction of $CO_2$ at the $CO_2$-$H_2O$ exchange site ($c_m$) using the $\Delta^{17}O$ and $\lambda$ values. Since this requires development of yet another complicated set of equations and detailed discussion of the process of assimilation from a plant physiology point of view, it would make our paper even more complex and less focused. A companion manuscript with detailed description and derivation of the $c_m$ using $\Delta^{17}O$ and $\lambda$ values is under preparation.

**As shown in Figure 5, the discrimination of $\Delta 17O$ of CO2 during photosynthesis varies widely, and controlled by the magnitude of oxygen isotope equilibration at the CO2-H2O site, that is to say, the**

relative contribution of kinetic (diffusion) and equilibrium isotope fractionation. This conclusion is almost identical to the knowledge using conventional δ18O results. Moreover, In the last paragraph of Discussion, authors mentioned that the main uncertainty is cm/ca ratio, which may be same as the main uncertainty of δ18O. My impression after reading this manuscript is that the intra-MDF variation dominate that of MIF signature on tropospheric CO2, which weakens the merit to study Δ17O of CO2. What is an advantage to use Δ17O instead of δ18O? Please provide suggestions or implications to general biogeochemists.

The referee is correct that the processes that affect $\delta^{18}O$ are the same that affect $\Delta^{17}O$. Nevertheless, the quantitative evaluation of $\Delta^{17}O$ is largely independent of $\delta^{18}O$. The limitation of using $\delta^{18}O$ of atmospheric $CO_2$ as a tracer is its dependency on the $\delta^{18}O$ value of different water reservoirs and fractionation processes in the hydrological cycle, water isotopic inhomogeneity and dynamics, which are difficult to ascertain (Hoag et al., 2005). Unlike $\delta^{18}O$, $\Delta^{17}O$ variation is much smaller and is better defined (Miller, 2018). This is because conventional bio-geo-chemical processes that modify $\delta^{17}O$ and $\delta^{18}O$ follow well-defined three-isotope fractionation slope. Consequently, the formulation of the $CO_2$ budget using $\Delta^{17}O$ is a lot simplified, compared to using $\delta^{18}O$. Furthermore, unlike $\delta$'s, $\lambda$ is insensitive to temperature (Cao and Liu, 2011;Bao et al., 2016;Hofmann et al., 2012;Dauphas and Schauble, 2016;Miller, 2018).

**Specific comments**

L41: "replaced using…" What this means? Be more specific.

In the revised manuscript it is replaced with "replicated based on cross-consistency checks with atmospheric inversions, sun-induced fluorescence (SIF) and dynamic global vegetation models"

L47: "see equation (1)" instead of "see below"

In the revised manuscript we used "see equation (1)"

L51: "the latter term" I guess it should be "the former term," which means photosynthetic CO2 uptake.

Thank you, in the revised manuscript corrected it to "the former term"

L53: "variable δ18O gradient" I think "significant δ18O variation" is more appropriate.

In the revised manuscript we used significant $\delta^{18}O$ variation

L56: Delete "the isotopically exchanged"

Deleted

L45-57: In this block, you should use the term "isotope fractionation" with its definition for the subsequent block. More desirably, the term "mass-dependent isotope fractionation (MDF)" with its definition.

In the revised manuscript, we included "These physico-chemical processes change $^{17}O/^{16}O$ by approximately half the corresponding change in $^{18}O/^{16}O$, a process called mass-dependent isotope fractionation (see equation 2). This is because the mass difference between $^{17}O$ and $^{16}O$ (1.0042 amu) is approximately half as large as the mass difference between $^{18}O$ and $^{16}O$ (2.0042 amu). "at the end of the paragraph.

L63: "mass-dependent fractionation" should be "mass-dependent isotope fractionation" with its definition in detail.

We excluded this paragraph, L62-64 in the revised manuscript since it does not add additional information to the paragraph mentioned above.

L62-64: Need revision because the latter paragraph is just a refrain of the former.

We excluded this paragraph, L62-64 in the revised manuscript since it does not add additional information.

L65: Describe a specific value instead using "considerable"

In the revised manuscript, instead of " considerable $\Delta^{17}O$" we used "the $\delta^{17}O$ of $CO_2$ is 1.7 to 2.2 times $\delta^{18}O$ of $CO_2$ (Wiegel et al., 2013)"

L60-71: In this block, you should use the term "mass-independent isotope fractionation (MIF)" with its definition, and associate it with "photochemical isotope exchange"

In the revised manuscript, we included the following paragraph
"
In nature, it was believed all process that modify the oxygen isotope distribution are mass dependent isotope fractionation until the discovery of the a deviation from the assigned mass dependent three-isotope fractionation line in meteorites (Clayton et al., 1973;Clayton et al., 1976) and ozone formation (Thiemens, 1983;Heidenreich and Thiemens, 1983, 1986), called mass-independent isotope fractionation (see

equation 3). The $\Delta^{17}O$ of ozone can be transferred to other oxygen bearing molecules via direct chemical reaction with ozone or via $O(^1D)$."

and rearranged the whole paragraph

340

L70-71: This is not sufficient because exchanges with soil and ocean water are also nonenzymatic processes.

345 The isotope exchange in the atmosphere is negligible due to lower liquid water content, lower residence time and the absence of carbonic anhydrase (Mills and Urey, 1940;Johnson, 1982;Miller et al., 1971;Silverman, 1982). We incorporated this sentence in the updated section.

It is true that $CO_2$-$H_2O$ exchange with ocean water is a non-enzymatic processes, but $CO_2$-$H_2O$ exchange
350 with soil water is controlled by carbonic anhydrase (Wingate et al., 2009), similar to the exchange with leaf water.

L78: "The $\Delta17O$ of CO2" instead of "The 17O-excess of CO2 ($\Delta17O$) (equation 4)"
355
In the revised manuscript, we only used $\Delta17O$

L80: Clarify "well-known three-isotope slope." "Non three-isotope person" cannot understand what this means.
360
In the revised manuscript, we included the three-isotope fractionation slope of 0.5229, and the figure above.

L92-106 and Figure 1: The explanation is this block is too general, should reduce to a few sentences.
365 Detail description may be required if you would like to discuss the difference of results due to the different types in the Discussion. As for Figure 1, not this scheme but simpler scheme in Figure S6 was actually used in this study. Therefore, it seems more appropriate to delete Figure 1 and insert S6 here.

In the revised manuscript, we merged the necessary information with the other paragraphs and we agree
370 that Figure 1 is not necessary, so it is left out. We excluded the general description of plant types. We only kept the following three sentences

"The mole fraction of $CO_2$ at the $CO_2$-$H_2O$ exchange site ($c_m$) is an important parameter to determine the effect of photosynthesis on the triple oxygen isotope composition of atmospheric $CO_2$. In $C_3$ plants, CA
375 is found in the chloroplast, cytosol, mitochondria and plasma membrane (Fabre et al., 2007;DiMario et al., 2016) and the $CO_2$-$H_2O$ exchange can occur anywhere between the plasma membrane and the

chloroplast. For $C_4$ plants, CA is mainly found in the cytosol, the $CO_2$-$H_2O$ exchange occurs in the cytosol (Badger and Price, 1994)."

380

L108-109: What is "leaf level"?

In the revised manuscript we changed it from leaf level to leaf scale

385

L116-117: "$\Delta 17O$" instead of "triple oxygen isotopic composition"

Changed accordingly

390   Equations 1 and 2: Should be merged such as, $\delta n O$ = n Rsample/ n RVSMOW – 1, n refers 17 or 18 or simpler, $\delta$ = Rsample/RVSMOW – 1.

Thank you, in the revised manuscript we used the first suggestion.

395

L134: I recommend "The MDF factor" instead of "The factor"

Changed accordingly

400   L135-137: Delete "This relation…, respectively.

In the revised manuscript we excluded the sentence, based on the suggestion above, we already defined mass dependent isotope fractionation.

405   L137: "variations" instead of "values." "Small delta value" is meaningless.

In the revised manuscript we changed values to variations. And at the end of the paragraph we introduced "Equation 4 can be linearized to $\Delta^{17}O = \delta^{17}O - \lambda \times \delta^{18}O$ (Miller, 2002), but this approximation causes an error that increases with $\delta^{18}O$." for more clarity.

410

L139-140: I recommend "Note that $\Delta 17O$ changes not only by MIF processes, but also MDF processes with a different $\lambda$ value from the definition,"

415

Changed accordingly

L145-146: "which was obtained by the observation of" instead of "the value associated with"

420

Changed accordingly

L147-148: Delete "Note that ... δ18O."

425 Changed accordingly

L150-258 (Section 2.2-2.4): Revise completely.

430 In the revised manuscript,
- we moved most of section 2.2 to the supplementary material
- we moved section 2.3 to the discussion
- we moved section 2.4 to the supplementary material

435

Equation 5: Use n (18 or 17) or simpler expression as above, then revise or delete

In the revised manuscript, we implemented the suggestion expression

440 L158 and L163. Equation 12: Move after equation 5 with related sentences.

Changed accordingly

L163-168: Delete "We note that...itself."
445
The sentence is excluded from the revised manuscript.

L170-200 and Section 2.4: Integrate and locate in new section such like "Extension of Farquhar-Lloyd model to oxygen triple isotopes. Eqs. 6 and 11 are almost identical so that they should be merged.
450 Equation 15: Use n (18 or 17) or simpler expression, then revise or delete

This section is moved to the supplementary material and revised in the new version of the manuscript

L208-213 and Figure 2: Move to SM.
455
Changed accordingly

Section 2.3: I recommend moving this section to the Discussion.

460  Changed accordingly

L217: Delete "which is a net sink,"

This section is moved to the discussion part, and "which is a net sink," is removed in the revised
465  manuscript.

L230: Specify which model is used.

In the revised manuscript this section is moved to the discussion section and revised entirely.
470
  L241-259: Here detail but still insufficient description was made only for $\delta m$, on the other hand, no
description for ci and $\delta i$ which were driven away to Appendix. This seems out of balance and forces
readers to jump here and there. I recommend moving this block to SM.

475  In the revised manuscript, we moved this part to the supplementary material

L256-257 and related sentences in Appendix A3. No definition of ci.

480
In the revised manuscript, definition for all parameters is included as a table in the appendix

L262-265: Could it be shorter?

In the revised manuscript we shortened this part by excluding the sentence from line 263 to line 265,
485  "*The dwarf type sunflowers were grown until the first leaf pair that was used for the experiments
reached the final size, which is about 4 weeks.*" We did the same for line 267 to 268, i.e. "*After at least
6 weeks in the growth chamber, leaves that had developed and matured there were used for the
experiment*" is excluded in the revised manuscript.

490  L268-269: "The 4th or higher…" Is this sentence an explanation for maize or all species?

In the revised manuscript, we write "For maize, the 4th or higher …. "

Section 3.2: Need the model and the manufacturer for halogen lamp, neutral filters, dewpoint meter (the
495  model).

The models and manufacturers are included in the revised manuscript

Section 3.3: Could this section be shorter to several sentences? The description for $\delta D$ and obtaining
optimum setting seem appropriate in SM.

In the revised manuscript this section has been considerably shortened and part of it is moved to the
supplementary material.

L349: Water was converted to O2

In the revised manuscript L349 - L 354 has been deleted to make the manuscript more concise.

Section 3.5: In previous section, unit of $\Delta 17O$ is ‰. Here ppm is used. Use a uniform manner.

In the revised manuscript, all numbers are given in ‰

Section 3.6: See related general comment

The leaf cuvette model is described here for the first time, as a result we cannot make it shorter than this.

L403: The last sentence is a refrain.

In the revised manuscript, we excluded the sentence.

Results: Show experimental results (raw data) such as c, $\delta$, $\Delta$, w, for entering and leaving from the cuvette,
etc. Show table of them and describe them.

In the revised manuscript, we provided the raw data for the gas exchange parameters in the beginning of
the results section

L414-415: Delete this sentence

Changed accordingly

Section 4.2: Avoid using "17O-excess" in the title and L433 for uniformity

Changed accordingly

L477-493: I could not understand this block. If the authors applied different lambda values to individual results, the vertical axis in Figure 8 would be meaningless, and one could not evaluate the graph and related description at all.

In the revised manuscript we incorporated the reference triple oxygen isotope fractionation slope ($\lambda$), also in the caption. Sorry for the confusion we did not mention it clearly. When we described $\Delta^{17}O$ in the theory section, we clearly mentioned which lambda value we used ($\lambda$=0.528).

Section 5.2: Avoid using "17O-excess" for uniformity
Changed accordingly

Figure 3: Add individual flow direction.
Changed accordingly

Figure 4: Panel b seems unnecessary. Delete and insert Figure 5a here.

Changed accordingly

Figure 5: Move Panel a to Figure 4 as above

Changed accordingly

Figure 6: Is it important to plot both of blue diamonds and curve. Should the curve be improved by blue diamonds?

The curves are based on the leaf cuvette model which we implemented for this study. The blue diamonds were the results for the individual experiments using the Farquhar model. In the revised manuscript, we excluded the blue diamond points.

**Typographic errors**

Space inserted after semicolon (e.g., L33)

L42: Welp et al. (2011)
Corrected

L45: The concept of the latter study..
 Corrected

580 L60: equation 4))
Corrected, now it is "see equation 2"

L207: Figure 2
Now Figure S1

585
 L237: "Following (Farquhar…..)" Need grammatical correctness

Corrected to: "The $CO_2$ mole fraction at the site of $CO_2$-$H_2O$ exchange is calculated as shown in equation
S10 following (Farquhar and Cernusak, 2012;Barbour et al., 2016;Osborn et al., 2017).
590 "
This section has also been moved to the supplementary material.

L267: Maize
595 In the revised manuscript, "Mays" is corrected to "Maize"

L279, L297: Need grammatical correctness.
**Line 279**

600 In the revised manuscript "of" is replaced by "for", now it reads as follow:
A schematic for the gas exchange experimental setup is shown in Figure 2

**L297**
**In the revised manuscript, we deleted the phrase "as described in detail in" since it does not change**
605 **the meaning of the sentence.  Now it reads as:**

**"The isotopically enriched $CO_2$ was prepared by photochemical isotope exchange between $CO_2$ and $O_2$**
**under UV irradiation (Adnew et al., 2019).**
**"**
610

Section 3.2: "Figure 3" instead of "Figure 2" (If Figure 2 were moved to SM, they are accidentally correct,
though)

615 Thank you, this is corrected in the revised manuscript
References: I found typo. in Barbour et al. (2016) and Caemmerer and Farquhar (1981). There may be
more. Confirm all.
Thank you very much. In the revised manuscript we corrected all of them. All of them related to the name
von Caemmerer.

620

L950: "entering and leaving" instead of "leaving and entering"
In the revised manuscript, we re-ordered it chronologically. The appendix has also been moved to the supplementary material.

625

Equation A1.4: If the referred article (Caemmerer and Farquhar, 1981) was correct, the denominator must be (gt ac + E/2).

630 Corrected
Thank you very much, all the **Typographic errors** are corrected in the revised manuscript.

References

[revised manuscript text omitted]

Outline numbered + Level: 2 + Numbering Style: 1, 2, 3, … + Start at: 1 + Alignment: Left + Aligned at:  0.65 cm + Indent at:  1.29 cm

| Page 23: [3] Deleted | Microsoft Office User | 5/22/20 12:01:00 PM |
|---|---|---|

| Page 23: [4] Deleted | Microsoft Office User | 5/22/20 12:01:00 PM |
|---|---|---|

| Page 23: [5] Deleted | Microsoft Office User | 5/22/20 12:01:00 PM |
|---|---|---|

| Page 24: [6] Deleted | Microsoft Office User | 5/22/20 12:01:00 PM |
|---|---|---|

1.1.

| Page 24: [7] Deleted | Microsoft Office User | 5/22/20 12:01:00 PM |
|---|---|---|

| Page 24: [8] Deleted | Microsoft Office User | 5/22/20 12:01:00 PM |
|---|---|---|

| Page 24: [9] Deleted | Microsoft Office User | 5/22/20 12:01:00 PM |
|---|---|---|

| Page 24: [10] Deleted | Microsoft Office User | 5/22/20 12:01:00 PM |
|---|---|---|

| Page 24: [11] Deleted | Microsoft Office User | 5/22/20 12:01:00 PM |
|---|---|---|

| Page 24: [12] Deleted | Microsoft Office User | 5/22/20 12:01:00 PM |
|---|---|---|

| Page 24: [13] Deleted | Microsoft Office User | 5/22/20 12:01:00 PM |
|---|---|---|

| Page 24: [14] Deleted | Microsoft Office User | 5/22/20 12:01:00 PM |
|---|---|---|

| Page 24: [15] Deleted | Microsoft Office User | 5/22/20 12:01:00 PM |
|---|---|---|

| Page 24: [16] Deleted | Microsoft Office User | 5/22/20 12:01:00 PM |
|---|---|---|

| Page 24: [17] Formatted | Microsoft Office User | 5/22/20 12:01:00 PM |
|---|---|---|

Outline numbered + Level: 1 + Numbering Style: 1, 2, 3, … + Start at: 1 + Alignment: Left +
Aligned at:  0.65 cm + Indent at:  1.29 cm

Outline numbered + Level: 2 + Numbering Style: 1, 2, 3, … + Start at: 1 + Alignment: Left + Aligned at:  0.65 cm + Indent at:  1.29 cm

| Page 26: [19] Deleted | Microsoft Office User | 5/22/20 12:01:00 PM |
|---|---|---|

| Page 26: [20] Formatted | Microsoft Office User | 5/22/20 12:01:00 PM |
|---|---|---|

Heading 2, Left, Adjust space between Latin and Asian text, Adjust space between Asian text and numbers

| Page 26: [21] Deleted | Microsoft Office User | 5/22/20 12:01:00 PM |
|---|---|---|

| Page 26: [22] Deleted | Microsoft Office User | 5/22/20 12:01:00 PM |
|---|---|---|

| Page 26: [23] Formatted | Microsoft Office User | 5/22/20 12:01:00 PM |
|---|---|---|

Font: 12 pt, Not Bold, Check spelling and grammar

| Page 26: [24] Deleted | Microsoft Office User | 5/22/20 12:01:00 PM |
|---|---|---|

| Page 26: [25] Formatted | Microsoft Office User | 5/22/20 12:01:00 PM |
|---|---|---|

Outline numbered + Level: 2 + Numbering Style: 1, 2, 3, … + Start at: 1 + Alignment: Left + Aligned at:  0.65 cm + Indent at:  1.29 cm

| Page 26: [26] Deleted | Microsoft Office User | 5/22/20 12:01:00 PM |
|---|---|---|

| Page 26: [27] Deleted | Microsoft Office User | 5/22/20 12:01:00 PM |
|---|---|---|

| Page 26: [28] Deleted | Microsoft Office User | 5/22/20 12:01:00 PM |
|---|---|---|

| Page 28: [29] Deleted | Microsoft Office User | 5/22/20 12:01:00 PM |
|---|---|---|

| Page 30: [30] Formatted | Microsoft Office User | 5/22/20 12:01:00 PM |
|---|---|---|

Outline numbered + Level: 2 + Numbering Style: 1, 2, 3, … + Start at: 1 + Alignment: Left + Aligned at:  0.65 cm + Indent at:  1.29 cm

| Page 30: [31] Deleted | Microsoft Office User | 5/22/20 12:01:00 PM |
|---|---|---|

| Page 40: [32] Deleted | Microsoft Office User | 5/22/20 12:01:00 PM |
|---|---|---|

**Page 43: [34] Deleted**        **Microsoft Office User**        **5/22/20 12:01:00 PM**

**Page 43: [35] Deleted**        **Microsoft Office User**        **5/22/20 12:01:00 PM**